# MreB filaments align along greatest principal membrane curvature to orient cell wall synthesis

Saman Hussain[1†], Carl N Wivagg[1†], Piotr Szwedziak[2], Felix Wong[3], Kaitlin Schaefer[4], Thierry Izoré[2], Lars D Renner[5], Matthew J Holmes[1], Yingjie Sun[1], Alexandre W Bisson-Filho[1], Suzanne Walker[6], Ariel Amir[3], Jan Löwe[2], Ethan C Garner[1]*

[1]Department of Molecular and Cellular Biology, Harvard University, Cambridge, United States; [2]MRC Laboratory of Molecular Biology, Cambridge, United Kingdom; [3]Harvard John A. Paulson School of Engineering and Applied Sciences, Cambridge, United States; [4]Department of Microbiology and Immunology, Harvard University, Cambridge, United States; [5]Leibniz Institute of Polymer Research, Dresden, Germany; [6]Department of Chemistry and Chemical Biology, Harvard University, Cambridge, United States

**Abstract** MreB is essential for rod shape in many bacteria. Membrane-associated MreB filaments move around the rod circumference, helping to insert cell wall in the radial direction to reinforce rod shape. To understand how oriented MreB motion arises, we altered the shape of *Bacillus subtilis*. MreB motion is isotropic in round cells, and orientation is restored when rod shape is externally imposed. Stationary filaments orient within protoplasts, and purified MreB tubulates liposomes *in vitro*, orienting within tubes. Together, this demonstrates MreB orients along the greatest principal membrane curvature, a conclusion supported with biophysical modeling. We observed that spherical cells regenerate into rods in a local, self-reinforcing manner: rapidly propagating rods emerge from small bulges, exhibiting oriented MreB motion. We propose that the coupling of MreB filament alignment to shape-reinforcing peptidoglycan synthesis creates a locally-acting, self-organizing mechanism allowing the rapid establishment and stable maintenance of emergent rod shape.

DOI: https://doi.org/10.7554/eLife.32471.001

*For correspondence:
egarner@g.harvard.edu

†These authors contributed equally to this work

Competing interests: The authors declare that no competing interests exist.

## Introduction

Although many bacteria are rod shaped, the cellular mechanisms that construct and replicate this geometry have remained largely unknown. Bacterial shape is determined by the cell wall sacculus, a giant, encapsulating macromolecule that serves to resist internal osmotic pressure. One of the primary components of the cell wall is peptidoglycan (PG), which is created by the polymerization of single glycan strands linked by peptide crossbridges. Studies of isolated cell walls from rod-shaped bacteria suggest material is generally oriented circumferentially around the rod, perpendicular to the long axis of the cell (*Gan et al., 2008*; *Hayhurst et al., 2008*; *Verwer et al., 1980*), or in thick cables in others (*Hayhurst et al., 2008*). This mostly circumferential, hoop-like organization of cell wall material allows the cell wall to better resist the internal osmotic pressure, as this pressure causes a stress twice as large in the circumferential direction (on the rod sidewalls) than in the axial direction (on the poles) (*Amir and Nelson, 2012*; *Chang and Huang, 2014*). This organization of material confers a mechanical anisotropy to the cell wall, causing it to stretch more along its length than across its width for a given stress; this anisotropy may assist rod-shaped cells in preferentially elongating

**eLife digest** Many bacteria are surrounded by both a cell membrane and a cell wall – a rigid outer covering made of sugars and short protein chains. The cell wall often determines which of a variety of shapes – such as rods or spheres – the bacteria grow into. One protein required to form the rod shape is called MreB. This protein forms filaments that bind to the bacteria's cell membrane and associate with the enzymes that build the cell wall. Together, these filament-enzyme complexes rotate around the cell to build and reinforce the cell wall in a hoop-like manner. But how do the MreB filaments know how to move around the circumference of the rod, instead of moving in any other direction?

Using a technique called total internal reflection microscopy to study how MreB filaments move across bacteria cells, Hussain, Wivagg et al. show that the filaments sense the shape of a bacterium by orienting along the direction of greatest curvature. As a result, the filaments in rod-shaped cells orient and move around the rod, while in spherical bacteria they move in all directions. However, spherical bacteria can regenerate into rods from small surface 'bulges'. The MreB filaments in the bulges move in an oriented way, helping them to generate the rod shape.

Hussain, Wivagg et al. also found that forcing cells that lack a cell wall into a rod shape caused the MreB filaments bound to the cell membrane to orient and circle around the rod. This shows that the organization of the filaments is sufficient to shape the cell wall.

In the future, determining what factors control the activity of the MreB filaments and the enzymes they associate with might reveal new targets for antibiotics that disrupt the cell wall and so kill the bacteria. This will require higher resolution microscopes to be used to examine the cell wall in more detail. The activity of all the proteins involved in building cell walls will also need to be extensively characterized.

DOI: https://doi.org/10.7554/eLife.32471.002

along their length (*Baskin, 2005*; *Chang and Huang, 2014*). Concordantly, atomic force microscopy (AFM) has shown that *Escherichia coli* sacculi are 2–3 times more elastic along their length than across their width (*Yao et al., 1999*). This rod-reinforcing circumferential organization is also observed in the cell walls of plants; hypocotyl and root axis cells rapidly elongate as rods by depositing cellulose fibrils in circumferential bands around their width, resulting not only in a similar dispersive rod-like growth, but also a similar anisotropic response to stress (*Baskin, 2005*). The organized deposition of cellulose arises from cortical microtubules self-organizing into a radial array oriented around the rod width, and this orients the directional motions of the cellulose synthases to insert material in circumferential bands (*Paredez et al., 2006*).

In contrast to our understanding of the self-organization underlying rod-shaped growth in plants, how bacteria construct a circumferential organization of glycan strands is not known. This organization may arise via the actions of a small number of genes essential for the formation and maintenance of rod shape. Collectively termed the Rod complex, (or elongasome) these include MreB, MreC, MreD (encoded by the mreBCD operon) (*Wachi et al., 1989*), RodZ (*Alyahya et al., 2009*; *Bendezú et al., 2009*), and the glycosyltransferase/transpeptidase enzyme pair RodA/Pbp2 (*Cho et al., 2016*). These components are conserved across a wide range of rod shaped bacteria, and mostly absent in cocci (*Alyahya et al., 2009*; *Chastanet and Carballido-Lopez, 2012*), leading many to speculate they function as the central determinants of rod shape (*Carballido-Lopez, 2006*; *Jones et al., 2001*).

The spatial coordination of RodA/Pbp2-mediated PG synthesis is conferred by MreB, an actin homolog (*Jones et al., 2001*; *van den Ent et al., 2001*). MreB polymerizes onto membranes as antiparallel double filaments, which have been observed to bend liposome membranes inward (*Figure 1A*) (*Salje et al., 2011*; *van den Ent et al., 2014*). Loss or depolymerization of MreB causes rod-shaped cells to grow as spheres (*Gitai et al., 2005*; *Jones et al., 2001*; *Bendezú et al., 2009*). *B. subtilis* contains 3 MreB paralogs (MreB, Mbl, and MreBH) that have been shown to co-polymerize into mixed filaments in vitro, and always colocalize in vivo (*Defeu Soufo and Graumann, 2004*; *Soufo and Graumann, 2010*; *Dempwolff et al., 2011*).

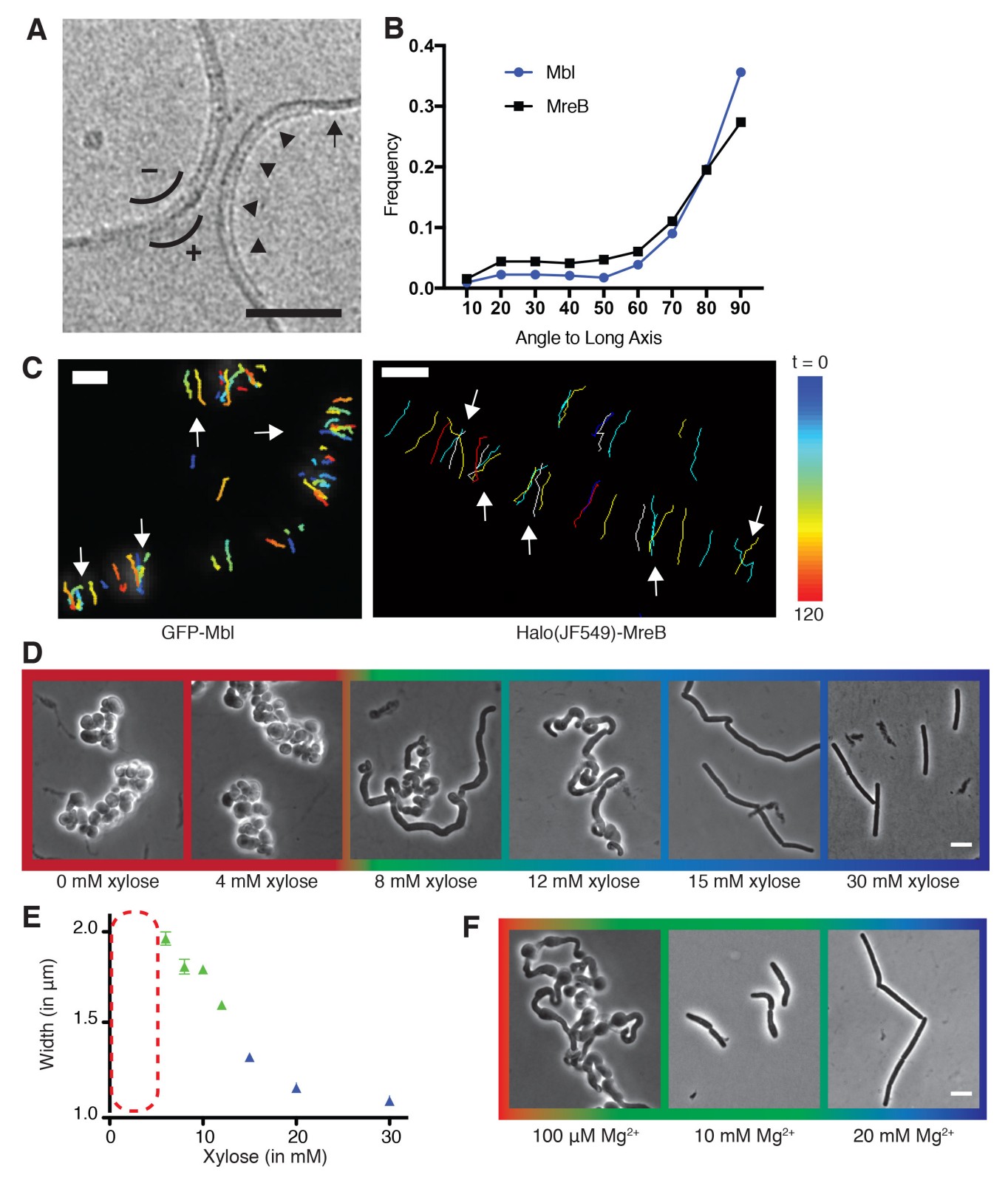

**Figure 1.** Curved MreB filament motions do not follow an ordered template (A–C). (**A**) The negative curvature of MreB filaments (arrowheads) aligns with the negative principal curvature of the liposome surface (arrow). Scale bar is 50 nm. (**B**) Angular distribution of GFP-MreB and GFP-Mbl trajectories relative to the long axis of *B subtilis* cells indicates that while the distributions have a mode of 90°, they are broad (Mean Deviation = 34°, n = 1041 for GFP-MreB and Mean Deviation = 26°, n = 1772 for GFP-Mbl). (**C**) Particle tracking of GFP-Mbl (left) and Halo-JF549-MreB (right) during 120 s (~1

*Figure 1 continued on next page*

*Figure 1 continued*

rotation) indicates trajectories close in time frequently cross paths (white arrows). Scale bar is 1 μm. See corresponding *Figure 1—video 1*. (D) Strains with *tagO* under inducible control display a teichoic acid-dependent decrease in width. Strain BEG300 was grown in LB supplemented with 20 mM Mg$^{2+}$. (E) Plot of cell width as a function of *tagO* induction in LB supplemented with 20 mM Mg$^{2+}$, calculated from rod-shaped cells (error bars are Standard Error of the Mean (SEM), n = 33, 56, 104, 175, 228, 489, 119). Areas not plotted at lower xylose levels (red dashed rectangle) are regions where cells are round (no width axis). Color scheme for D-F: red indicates round cells (no width axis), blue indicates rods (measurable width axis), and green indicates intermediate regimes where both rods and round cells are observed. (F) BEG300 at an intermediate level of *tagO* induction (15 mM xylose) shows a Mg$^{2+}$ dependent decrease in width. All scale bars are 5 μm. See also *Figure 1—figure supplement 1*.

DOI: https://doi.org/10.7554/eLife.32471.003

The following video, source data, and figure supplements are available for figure 1:

**Source data 1.** *Figure 1B* – Raw GFP-MreB and GFP-Mbl track angle values from the cell midline and their associated frequency distributions.
DOI: https://doi.org/10.7554/eLife.32471.006
**Source data 2.** *Figure 1—figure supplement 1B*-left – Mean cell width, Standard Deviation (SD) and number of cells (N) analyzed at various magnesium and xylose concentrations.
DOI: https://doi.org/10.7554/eLife.32471.007
**Figure supplement 1.** Varying magnesium levels in the growth medium changes cell shape.
DOI: https://doi.org/10.7554/eLife.32471.004
**Figure supplement 2.** Schematic of cell contours and correlation.
DOI: https://doi.org/10.7554/eLife.32471.005
**Figure 1—video 1.** Movie showing the trajectories taken by MreB and Mbl filaments frequently cross each other close in time.
DOI: https://doi.org/10.7554/eLife.32471.008

MreB filaments move circumferentially around the width of rod-shaped cells (*Domínguez-Escobar et al., 2011*; *Garner et al., 2011*; *van Teeffelen et al., 2011*).Super-resolution imaging has demonstrated that MreB filaments always translocate along their length, moving in the direction of their orientation (*Olshausen et al., 2013*). MreB filaments move in concert with MreC, MreD, and RodA/Pbp2 (*Domínguez-Escobar et al., 2011*; *Garner et al., 2011*), and loss of any one component stops the motion of the others. The directional motion of MreB filaments and associated Rod complexes depends on, and thus likely reflects, the insertion of new cell wall, as this motion halts upon the addition of cell wall synthesis-inhibiting antibiotics (*Domínguez-Escobar et al., 2011*; *Garner et al., 2011*; *van Teeffelen et al., 2011*), or specific inactivation or depletion of Pbp2 (*Garner et al., 2011*; *van Teeffelen et al., 2011*) or RodA (*Cho et al., 2016*).

It is not known how MreB and the rest of the Rod complex construct rod-shaped cells. As the motions of the Rod complexes reflect the insertion of new cell wall, their circumferential motions could deposit glycans in the hoop-like organization required to both build and reinforce rod shape. Therefore, we worked to understand the origin of this circumferential organization, seeking to determine what orients the motions of MreB and associated enzymes around the rod width in *Bacillus subtilis*.

## Results

### Oriented MreB motion is unlikely to arise from an ordered cell wall template

The mechanism by which MreB filaments and associated PG synthases orient their motion around the rod circumference is not known. Each filament-synthase complex is disconnected from the others, moving independently of proximal neighbors (*Garner et al., 2011*). The organized, circumferential motion of these independent filament-synthase complexes could arise in two ways: (1) A templated organization, where cell wall synthetic complexes move along an existing pattern of ordered glycan strands in the cell wall as they insert new material into it (*Höltje, 1998*), or (2) A template-independent organization, where each synthetic complex has an intrinsic mechanism that orients its motion and resultant PG synthesis around the rod circumference.

To explore the extent of order within the motions of the Rod complex, we analyzed the trajectories of GFP-Mbl and GFP-MreB with respect to the cell body using total internal reflection fluorescence microscopy (TIRFM) (*Figure 1B*). Overall, these motions are circumferentially oriented, but not perfectly aligned, a characteristic reflected by the broad distribution of angles that GFP-MreB, its

homologs, and the other components of the Rod complex move relative to the long axis of the cell (*Domínguez-Escobar et al., 2011*; *Garner et al., 2011*). However, examination of TIRFM time lapse movies revealed that both MreB and Mbl trajectories close in time (within the period of one revolution) frequently cross (*Figure 1C*, *Figure 1—video 1*), making it unlikely that MreB filaments move along a perfectly ordered template. As MreB movement reflects the insertion of new glycan strands, these motions indicate that the siacculus is built from somewhat disorganized, yet predominantly circumferential strands. This conclusion is in agreement with X-ray diffraction (*Balyuzi et al., 1972*; *Labischinski et al., 1979*) and cryoelectron microscopy studies of *E. coli* sacculi (*Gan et al., 2008*) which found that, while glycans are oriented circumferentially around the rod width on average, they are not ordered, running at variable angles to each other. Similarly, atomic force microscopy of *B. subtilis* sacculi has observed a generally oriented, but unaligned arrangement of 50 nm thick cables oriented roughly perpendicular to the long axis (*Hayhurst et al., 2008*). Thus, both the motions of MreB and ultrastructural studies indicate the sacculus is not highly ordered, making it unlikely that it can serve as a self-propagating spatial template for rod shape. Furthermore, given that preexisting cell wall is not necessary for the regeneration of rod shape from wall-less *B. subtilis* L-forms (*Kawai et al., 2014*), it is likely that both oriented MreB motion and rod shape can arise without an ordered template.

## MreB motions become isotropic in the absence of rod shape

As it appeared that organized MreB motion does not arise from patterns in the cell wall, we hypothesized there was an intrinsic mechanism orienting the motion of each MreB filament-cell wall synthetic complex. To test this hypothesis, we examined MreB motions as we changed the shape of cells from rods to spheres. As the internal osmotic pressure and stiffness of *B. subtilis* resists external mechanical perturbations to its shape (*Renner et al., 2013*), we first altered the shape of cells by controlling the level of wall teichoic acids (WTAs). WTAs are negatively charged cell wall polymers believed to increase the rigidity of the sacculus (*Matias and Beveridge, 2005*), a process that could occur via their coordination of extracellular $Mg^{2+}$ (*Thomas and Rice, 2014*; *Kern et al., 2010*), or modulation of hydrolase activity (*Atilano et al., 2010*). Knockouts of *tagO*, the first gene in the WTA synthesis pathway, create large, slow-growing, round cells that still synthesize PG, building extremely thick and irregular cell walls (*D'Elia et al., 2006*). We placed *tagO* under xylose-inducible control and grew cells at different induction levels. As expected, at high TagO inductions, cells displayed normal widths. As we reduced TagO levels, rods became gradually wider (*Figure 1D–E*) until, beneath a given induction, cells were no longer able to maintain rod shape, growing as spheres (or clumps of spheres) with no identifiable long axis. At intermediate induction levels, we observed a transition region between the two states, with cells growing as steady state populations of interconnected rods and spheres (*Figure 1D*). In agreement with models that (A) WTAs work with PG to bind $Mg^{2+}$ (*Thomas and Rice, 2014*; *Kern et al., 2010*), and (B) are required for cell wall rigidity (*Matias and Beveridge, 2005*), both the cell width and the amount of TagO induction determining the rod/sphere transition could be modulated by $Mg^{2+}$ levels (*Figure 1F*, *Figure 1—figure supplement 1B*). Likewise, decreasing extracellular $Mg^{2+}$ or *tagO* induction resulted in increasingly curved cell contours (*Figure 1—figure supplement 1B*), suggesting the wall was becoming more flexible.

By tracking the motion of GFP-MreB filaments in these differing cell shapes, we found that motion is always oriented in rods, moving predominantly circumferentially at all induction levels above the rod/sphere transition. However, in round cells (those induced beneath the rod/sphere transition point or in *tagO* knockouts) MreB filaments continued to move directionally, but their motions were isotropic, moving in all directions (*Figure 2A*, Movie *Figure 2—video 1*). To quantify the relative alignment of MreB under each condition, we calculated the angle between trajectory pairs less than 1 μm apart (*Figure 2B*, *Figure 2—figure supplement 1A*). This analysis revealed that MreB motions are more aligned when cells are rods: above the rod/sphere transition, trajectories have a median angle difference of 26°; while at low TagO inductions, where cells are round, the angle difference increases to 42°, close to that of randomly oriented trajectories (45°).

To verify that the loss of oriented MreB motion was due to the changes in cell shape, and not from some other effect of reduced WTA levels, we created round cells by alternate means. Depletion of both elongation PG transpeptidases (Pbp2a and PbpH) causes rod-shaped cells to become wider over time as they convert to spheres (*Garner et al., 2011*). We used this gradual transition of rods into spheres to examine both the width and overall shape dependence of MreB motion. At

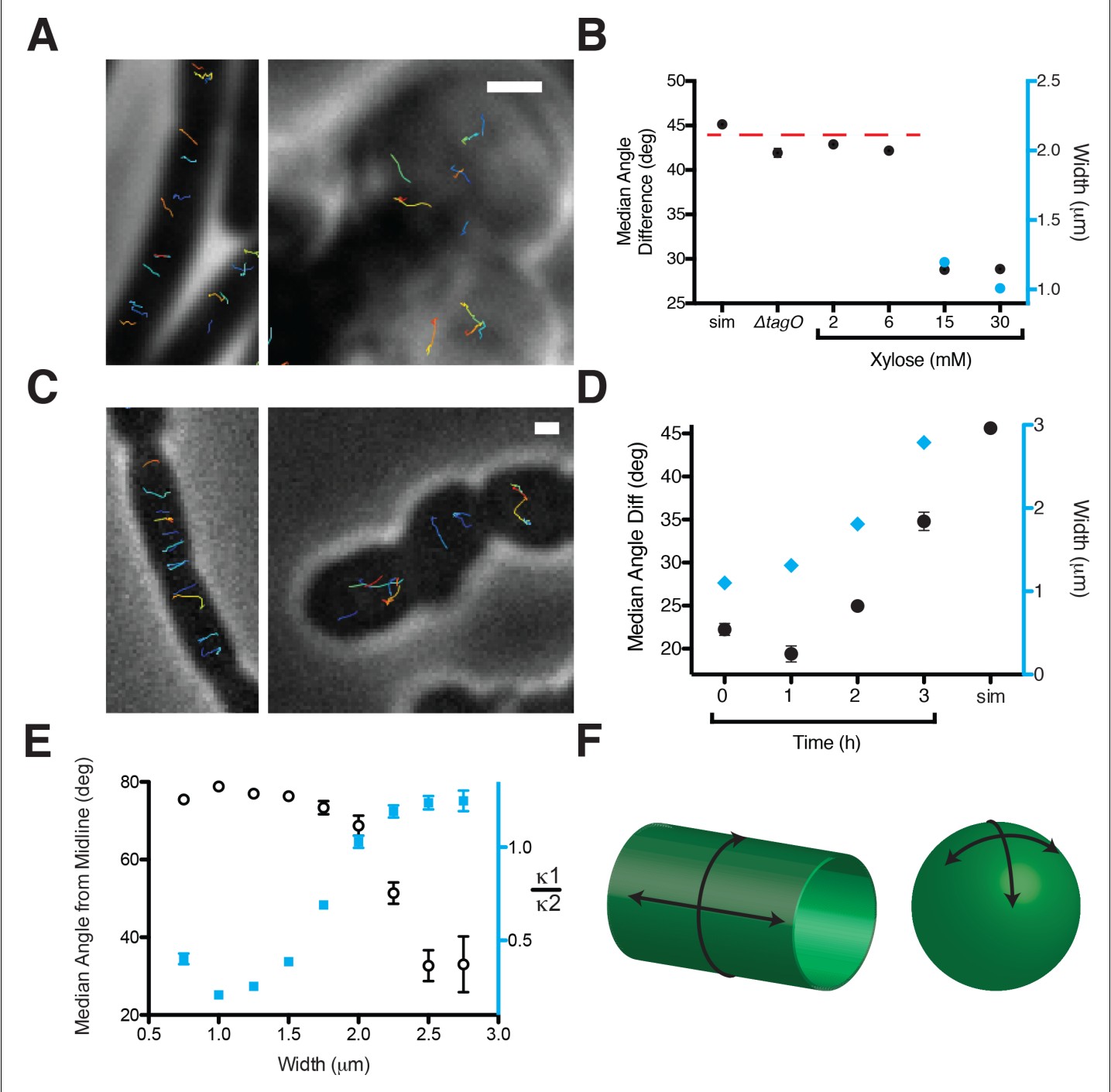

**Figure 2.** Oriented MreB motion correlates with rod shape. (A) BEG300 at maximum *tagO* induction (30 mM) is rod-shaped, and MreB tracks are largely oriented perpendicular to the midline of the cell (*left*). Δ*tagO* cells show round morphologies with unaligned MreB motion (*right*). (B) Median inter-track angle difference for track pairs ≤ 1 μm apart, plotted for BEG300 at several *tagO* induction levels, Δ*tagO* cells, and a simulation of randomly oriented angles (*sim*). n > 1300 for all data points. For spherical cells width is not measurable, indicated with a dashed red line. (C) Δ*pbpH* cells with *pbpA* under IPTG control display aligned MreB motion when *pbpA* is fully induced and cells are rods (*left*), but display unaligned MreB motion as Pbp2a levels reduce and cells become round (*right*). (D) Median inter-track angle difference for track pairs 1 μm apart during Pbp2a depletion with cell widths at each time point. Error bars are SEM, n > 600 for all data points. (E) Median angle from the midline (white circles) calculated for all rod-shaped cells from experiments in 2A-D plotted as a function of cell width. MreB filament alignment falls off rapidly beyond 2 μm, a point corresponding to where cells become round, as shown by the ratio of principal curvatures (blue squares) approaching 1. Error bars are SEM, n = 2993. See *Figure 2—figure supplement 1E* for further explanation. (F) Schematic showing the difference between the 2D surface curvature profile of rods and spheres. On *Figure 2 continued on next page*

*Figure 2 continued*
the inside surface of spheres, all points have negative, yet equal values for both principal curvatures. In rods, however, one principal curvature is negative (the radius), while the other is 0 (the flat axis along the rod). All scale bars are 1 μm. All error bars are SEM. See also *Figure 2—figure supplement 1*.

DOI: https://doi.org/10.7554/eLife.32471.009

The following video, source data, and figure supplement are available for figure 2:

**Source data 1.** *Figure 2B* – Median angle difference between track pairs at 2 mM, 6 mM, 15 mM and 30 mM xylose and in the *tagO* knockout, along with Standard Deviation (SD) values and the number of track pairs analyzed (N).

DOI: https://doi.org/10.7554/eLife.32471.011

**Source data 2.** *Figure 2—figure supplement 1A* – Angular correlation between track pairs binned as a function of the distance between the pair at various xylose concentrations and in the *tagO* knockout.

DOI: https://doi.org/10.7554/eLife.32471.012

**Figure supplement 1.** Relationships between cell width, MreB orientation, and cell shape.

DOI: https://doi.org/10.7554/eLife.32471.010

**Figure 2—video 1.** Timelapse showing circumferential motions of GFP-MreB in rod shaped cells with high TagO expression (BEG300 with 30 mM xylose, and GFP-MreB induced with 50 μM IPTG).

DOI: https://doi.org/10.7554/eLife.32471.013

**Figure 2—video 2.** Timelapse of GFP-Mbl trajectories occurring 2 hr after the initiation of Pbp2a depletion.

DOI: https://doi.org/10.7554/eLife.32471.014

initial points of depletion (1–2 hr) the rods widened but maintained circumferential MreB motion. At 2.5 hr of PbpA depletion, cells were a mix of spheres and rods of differing widths. These cells displayed the same pattern of MreB orientation observed with *tagO* depletions: round cells contained unoriented MreB, while nearby rod-shaped cells showed circumferential motion. Identical behavior was observed for GFP-Mbl during PbpA depletions (*Figure 2C*, Movie *Figure 2—video 2*). Quantitation of trajectories from all cells (both rods and spheres) at each time point of depletion indicated an increase in the median angle between trajectories as the population grew wider and rounder over time (*Figure 2D*, *Figure 2—figure supplement 1B*).

In *E. coli*, the angle of mutant MreB filaments relative to the long axis has been reported to increase with cell width (*Ouzounov et al., 2016*). To test if the angle of MreB movement changes with respect to cell width in *B. subtilis*, we calculated the angle of each trajectory to the long axis for all cells in our data with an identifiable width axis. At the same time, we also measured the curvature of each cell to determine how the overall shape of the cell affected the orientation of motion. This revealed that MreB motion in rods remained equivalently oriented over a wide range of rod widths, up to ~2 μm (*Figure 2E*, *Figure 2—figure supplement 1C–E*). Beyond a 2 μm width, cells began to lose their rod shape as they became more spherical, and the predominantly circumferential orientation of MreB motion was lost (*Figure 2E*, *Figure 2—figure supplement 1E*). This suggested that oriented MreB motion does not sense or rely on a specific cell radius; rather the orientation relies on differences between the two principal curvatures of the membrane. It appears that the motion of MreB filaments is oriented along the direction of greatest principal curvature: In rods, there is zero curvature along the rod length, and high curvature around the rod circumference, along which filaments orient. In contrast, in round cells where MreB motion is isotropic, the two principal curvatures are equal (*Figure 2F*).

## MreB aligns within round cells and protoplasts forced into rod shape

To further verify that MreB filaments orient in response to overall cell shape, we externally imposed rod shape on cells with unoriented MreB motion. We loaded TagO-induced cells into long 1.5 ×1.5 μm microfluidic chambers, then reduced TagO expression to levels insufficient to produce rods in liquid culture (*Figure 3A*, *Figure 3—figure supplement 1A*). After TagO depletion, cells expanded to fill the chamber indicating that WTA-depletion caused shape changes just as in bulk culture (*Figure 3A*, *Figure 3—figure supplement 1A*). Within these chambers, cells grew as rods, but at a wider width (1.5 μm) than wild-type cells, set by the chamber. When cells grew out of the chamber they swelled just as in bulk culture, showing confinement was required for rod shape at this induction level (*Figure 3B*, *Figure 3—figure supplement 1A*). In the TagO-depleted cells confined into rod shapes, MreB moved circumferentially (*Figure 3C*, *Figure 3—video 1*), confirming that MreB orients

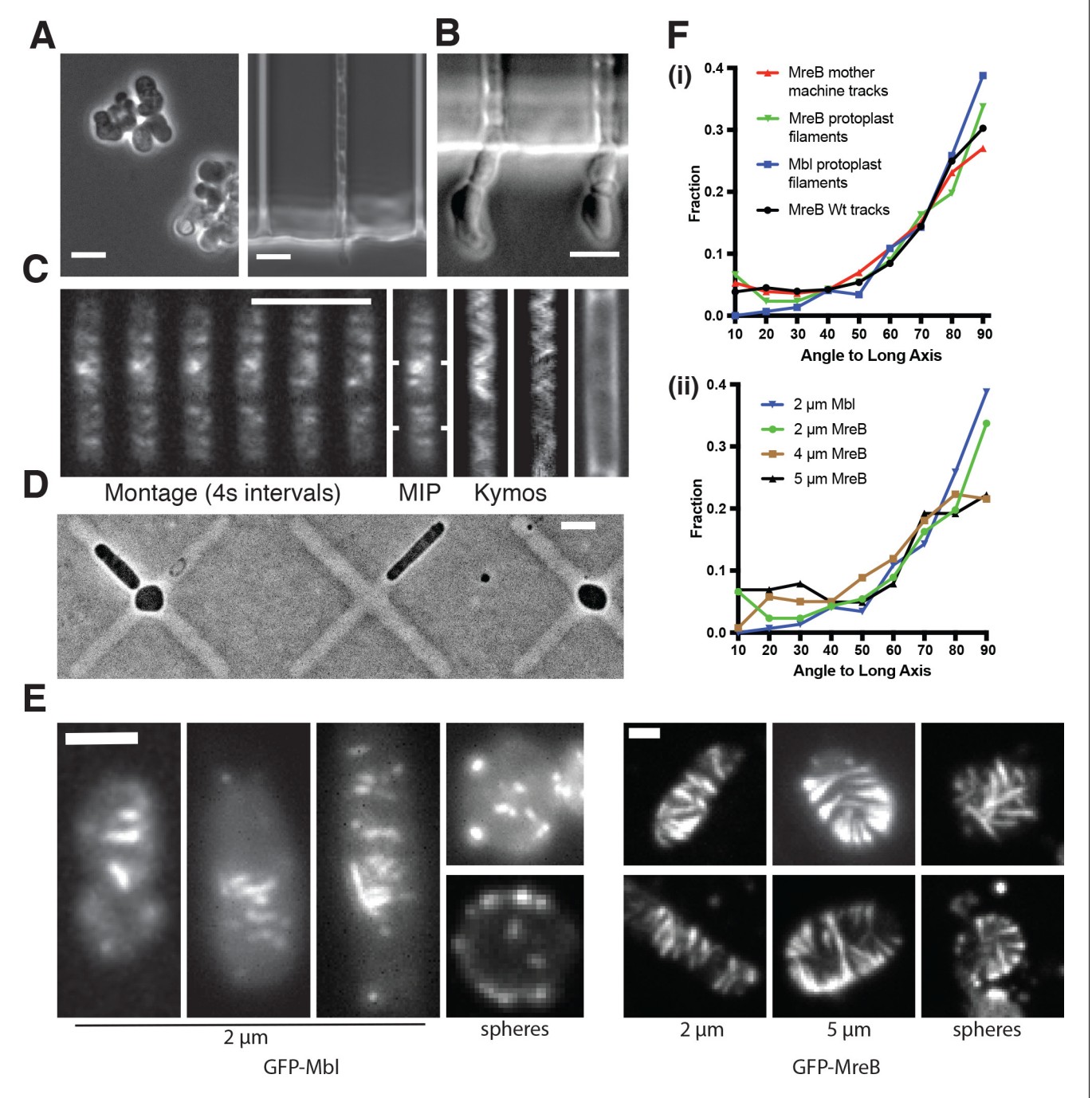

**Figure 3.** MreB filaments orient when rod shape is induced by external confinement. (**A**) Phase contrast images of BEG300 grown in LB supplemented with 2 mM xylose and 20 mM Mg$^{2+}$ in bulk culture (*left*) or confined into microfluidic channels of 1.5 ×1.5 µm (*right*). (**B**) Confined cells induced at 3 mM xylose in 20 mM Mg$^{2+}$ progressively swell upon escaping confinement into free culture. See also Figure *Figure 3—figure supplement 1A*. (**C**) (*Left*) Fluorescence microscopy montage of MreB filaments moving across a confined cell of BEG300 induced at 2 mM xylose in 20 mM Mg$^{2+}$. (*Right*) Maximal intensity projection of montage, kymographs of marked points and a phase contrast image of the cell. Scale bars for a-c = 5 µm. (**D**) Phase contrast images of protoplasts contained in agar crosses. Cells in the center grow to be round while cells in arms grow as elongated rods. (**E**) (*left*) Short GFP-Mbl filaments orient circumferentially in rod-shaped protoplasts (*2 µm*) but lack orientation in round protoplasts (*spheres*). (*right*) Long GFP-MreB filaments orient in rod-shaped protoplasts (*2 µm*); GFP-MreB filaments are still oriented in wider rod-shaped protoplasts (*5 µm*), but not to the same extent. In round protoplasts, GFP-MreB filaments are unoriented (*spheres*). Scale bar is 2 µm. (**F**) (i)The angular distribution of filaments within protoplasts (*Protoplast filaments*) is peaked at 90° (mean deviation = 34°, n = 147), similar to that of MreB motion in TagO-depleted, confined cells (*Mother machine tracks*) (mean deviation = 36°, n = 359) and MreB motion in wild-type cells (*Wt. tracks*) (mean deviation = 34°, n = 1041). (ii) In channels

*Figure 3 continued on next page*

*Figure 3 continued*

of varying widths (2, 4 and 5 µm), the orientation of GFP-MreB filaments remains circumferential, peaking at 90° but the filament angles deviate more from 90° as channel width increases (mean deviation = 34°, n = 258 at 2 µm), (mean meviation = 35°, n = 260 at 4 µm) and (Mean Deviation = 41°, n = 203 at 5 µm.). All mean deviation"values are calculated as the mean deviation from 90°.

DOI: https://doi.org/10.7554/eLife.32471.015

The following video, source data, and figure supplement are available for figure 3:

**Source data 1.** *Figure 3Fi* – Raw angle values from the cell midline of wildtype MreB tracks (tracks), MreB tracks in confined cells (mother machine) and MreB filaments in protoplasts (filaments), along with their associated frequency distributions.

DOI: https://doi.org/10.7554/eLife.32471.017

**Source data 2.** *Figure 3—figure supplement 1B* – Doubling times (min) and Standard Deviation (SD) of rod-shaped and spherical cells measured by taking bulk OD600 measurements (Bulk), using single cell measurements (Single Cell), single cell measurements in cells recovering rod shape (Recovery Single), single cell measurements normalized to the cell volume (Single/Volume) and single cell measurements in spherical cells confined to rod shape (Confined).

DOI: https://doi.org/10.7554/eLife.32471.018

**Figure supplement 1.** TagO levels and confinement.

DOI: https://doi.org/10.7554/eLife.32471.016

**Figure 3—video 1.** Timelapse showing circumferential motion of GFP-MreB in BEG300 induced at low TagO levels (2 mM xylose) when confined into long 1.5 × 1.5 µm channels.

DOI: https://doi.org/10.7554/eLife.32471.019

**Figure 3—video 2.** Timelapse showing GFP-MreB (top) and GFP-Mbl (bottom)in protoplasted cells showing Mbl does directionally.

DOI: https://doi.org/10.7554/eLife.32471.020

in response to the cells having rod shape. This experiment demonstrates that the isotropic MreB motion observed in round cells arises from the lack of rod shape, and not from some other effect of our genetic perturbations. This experiment also showed another unexpected result: the doubling time of free (unconstrained) cells induced at similar TagO levels is long (53 ± 10 min), but confining them into rod shape restored their doubling time (44 ± 4 min) toward wild-type times (39 ± 9 min) (*Figure 3—figure supplement 1B*).

We next attempted to minimize any contribution to MreB filament orientation from (A) the directional motion of filaments, and (B) any pre-existing order within the sacculus. To accomplish this, we examined filament orientation in protoplasts (cells that had their cell wall enzymatically removed) confined into different shapes, using highly expressed GFP-MreB to assay long filaments, and GFP-Mbl to assay short filaments. We protoplasted cells in osmotically stabilized media (*Wyrick and Rogers, 1973*), then grew them under agar pads containing micro-patterned cross shapes. Cells in the center of these crosses (~5 µm diameter) were forced to grow as spheres, whereas cells in the arms were constrained to grow into rods of various widths ranging from 2 to 5 µm (*Figure 3D*). As reported previously (*Domínguez-Escobar et al., 2011*), MreB filaments within protoplasts did not move directionally (*Figure 4—video 1*), likely because the cell wall provides the fixed surface along which the PG synthesis enzymes move. Within the protoplasts confined into the smallest rod shapes (2 µm), filaments oriented at a distribution of angles predominantly perpendicular to the cell length (*Figure 3E–F*). The angular distributions of short GFP-Mbl filaments and longer GFP-MreB filaments were similar to each other, and also similar to the distribution of 1) filament trajectories observed in intact, wild-type cells and 2) filament trajectories of TagO depleted cells confined into the mother machine (*Figure 3Fi*). As we increased the width of the imposed rod shape from 2 to 5 µm, filaments remained predominantly oriented in all cases (*Figure 3Fii*), but their mean deviation from 90° increased as the rod width increased (34° at 2 µm, 35° at 4 µm, and 41° at 5 µm). In contrast to confinement in rods, both short and long filaments in spherically confined protoplasts remained unoriented (*Figure 3E*). Together, these data demonstrate that MreB filaments orient to point around the rod width even in the absence of pre-existing cell wall or directional motion, as long as the cell has a rod shape. These experiments also demonstrate that MreB filaments will align even in wider rods, where the difference in principal curvatures is smaller than in wild-type cells, but that, as the difference in principal curvatures decreases, filament alignment becomes more disordered.

## MreB filaments orient around liposome tubes in vitro

To test if MreB filaments are themselves sufficient to align along the predominant direction of membrane curvature, we assembled purified *T. maritima* MreB within liposomes and visualized it using cryoelectron microscopy and tomography. While controlling the final concentration of protein encapsulated within liposomes ≤1 μm is difficult, we were able to assemble MreB inside liposomes at high concentrations. At these concentrations, MreB filaments tubulated liposomes, creating rod-like shapes (*Figure 4A*, *Figure 4—figure supplement 1A–B*, *Figure 4—video 1*). In tubulated regions, MreB filaments could be traced around the circumference of the liposome tube, while filaments in spherical regions were found in all possible orientations (*Figure 4A*). At the highest concentrations, tubulated liposomes contained closely packed filament bundles, allowing us to observe a regular patterning of the canonical double filaments of MreB (*Figure 4B*). Purified wild-type MreB did not bind to the outside surface of small liposomes contained within larger ones (*Figure 4A*), indicating that MreB filaments preferentially polymerize on inward (negative) curvatures, akin to the inner leaflet of the bacterial membrane. In the absence of MreB, liposomes are spherical, with no deformations (*Figure 4—figure supplement 1C*). Together, this data suggests that MreB filaments themselves are sufficient to align along the predominant direction of membrane curvature, as observed here with laterally associated filaments. We note that the experimental limitations of the liposomal system, combined with the tendency of MreB filaments to self-associate make it difficult for us to acquire and study the alignment of single filaments in vitro. Also, it remains to be determined if membrane-associated MreB filaments exist as bundles or isolated filaments in vivo.

## Biophysical modeling suggests highly bent MreB filaments orient along the greatest principal curvature to maximize membrane interactions, a prediction insensitive to large variations in parameters

The above observations demonstrate that MreB filaments sense and align along the direction of greatest principal curvature, that is, the more curved inner surface of the rod circumference. The ultrastructure of MreB filaments provides a possible mechanism: MreB filaments are bent (*Salje et al., 2011*), with the membrane-interacting surface on the outer face of the bend (*Figure 4C*). This bent conformation could cause filaments to preferentially orient along the curved rod circumference, rather than the flat rod length, to maximize the burial of hydrophobic moieties into the membrane, a mechanism suggested by previous theory (*Wang and Wingreen, 2013*).

As the curvature of MreB filaments bound to liposomes is much greater (~200 nm diameter [*van den Ent et al., 2014*]) than that of *B. subtilis* cells (~900 nm diameter), we performed analytical calculations to model how highly curved MreB filaments would align within a cell with a less curved surface (*Figure 4D–G*, Appendix 1). As many of the biochemical and physical parameters of MreB are still unknown, we first assumed a fixed set of parameters, and later verified that our results were robust over a large parameter range. We initially assumed a membrane interaction energy of 10 kT per monomer (calculated from residues involved in membrane associations [*Salje et al., 2011*]), and a similar Young's modulus to actin (2 GPa). We modeled filaments as elastic beams made of two protofilaments. In addition, we used the Helfrich free energy to model the energetics of membrane deformation, and accounted for the work done against osmotic pressure due to changes in volume (Appendix 1). These calculations indicate that the total energy is minimized when filaments orient along the direction of maximal curvature (*Figure 4F*) and that, importantly, the energy penalty for incorrectly-oriented filaments is much greater than the energy of thermal fluctuations. Interestingly, this modeling indicates a decrease in energetic preference for the preferred filament orientation as the radius of the cell is increased (*Figure 4F*), a prediction in qualitative agreement with our observations of alignment in protoplasts. Furthermore, our calculations indicate that orientation is robust over a large, biologically relevant range of parameters, including the membrane binding energy, filament length, and filament Young's modulus (*Figure 4G*).

These calculations predict that filaments should orient circumferentially both if the membrane deforms to the filament (at low osmotic pressures or if filaments are stiff) (*Salje et al., 2011*), or if filaments deform to the membrane (at high osmotic pressures or if filaments are flexible) (*Figure 4E*). Our experimental data demonstrates MreB filament alignment across a range of pressures: high within cells, low to none within liposomes, and a pressure between the two within osmotically-stabilized protoplasts. In the absence of osmotic pressure, MreB filaments deform liposomes since it is

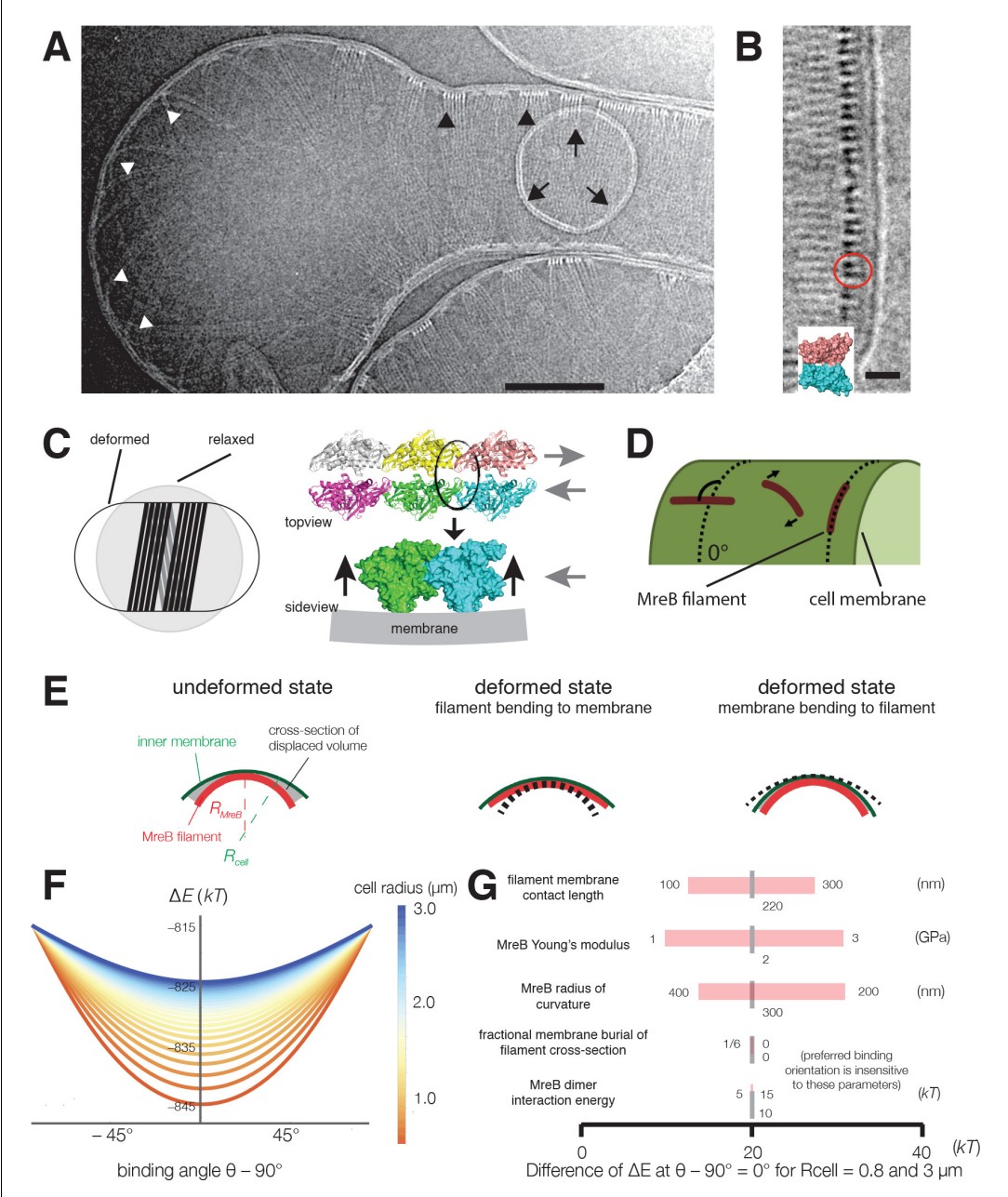

**Figure 4.** Cryoelectron tomography shows *T. maritima* MreB filaments assembled in liposomes align perpendicular to the rod axis. (A) Cryoelectron tomography of *T. maritima* MreB filaments assembled inside liposomes. Black arrowheads show aligned bundles of filaments in a tubulated liposome, white arrowheads show unaligned bundles in a spherical region of the same liposome. Arrows show a positively curved surface inside the liposome, to which no MreB filaments bind. Scale bar is 100 nm. See also *Figure 4—figure supplement 1*. (B) Zoomed in view of MreB filaments within a tubulated liposome shows MreB adopts a double-stranded antiparallel protofilament arrangement (dark lobes), consistent with previous models of MreB monomer interactions (colored structure in inset) (*van den Ent et al., 2014*). The red circle indicates the cross-section of an MreB filament, modelled in the inset. Scale bar is 50 nm. (C) (*left*) Schematic drawing depicting the cause of the shape change from spherical to rod-shaped liposomes: MreB wants to attain greater curvature and since there are many filaments, they are laterally stabilized. As the liposome is much more easily deformable than cells, the resulting energy minimum is a deformed liposome with an MreB helix on the inside. (*right*) Model showing why the unusual architecture of MreB filaments might have been selected during evolution: its juxtaposed subunits in the two antiparallel protofilaments produce putative hinges that could be the region of bending for these filaments. Canonical F-actin filament architectures, with staggered subunits, would need bending within the subunits, which is less easily achieved. Modeling of MreB – membrane interactions and filament orientation. (D, E) Hydrophobic residues are located on the outer edge of the antiparallel MreB double filament, which is here modeled as an elastic cylindrical rod. To achieve maximum hydrophobic burial, membrane deformation, MreB bending, or a combination of the two may occur. (F) A plot of the change in total energy (ΔE) caused by the MreB-

*Figure 4 continued on next page*

*Figure 4 continued*

membrane interaction against the binding angle θ for various cell radii shown in the color scheme on the right. Note that ΔE is minimal at θ = 90°, which agrees with the observed orientation of MreB binding and motion. At larger rod radii, the energetic well becomes flatter and MreB binding becomes more susceptible to thermal fluctuations and other sources of stochasticity, which would result in a broader angular distribution of filaments. (G) A sensitivity analysis of the model over a range of model parameters.

DOI: https://doi.org/10.7554/eLife.32471.021

The following video and figure supplement are available for figure 4:

**Figure supplement 1.** *T. maritima* MreB filaments assembled in liposomes support alignment around the rod axis.

DOI: https://doi.org/10.7554/eLife.32471.022

**Figure 4—video 1.** (first sequence) PyMOL volume rendering of an electron cryotomography 3D map *of T. maritima* MreB included in a liposome (corresponds to liposome depicted in *Figure 4E*.

DOI: https://doi.org/10.7554/eLife.32471.023

energetically more favorable to bend the membranes than to bend the filaments, as observed in our in vitro data (*Figure 4A*, *Figure 4—figure supplement 1*). However, in live cells, our modeling predicts that MreB filaments cannot deform the inner membrane due to the large osmotic pressure, and instead deform to match the greatest principal membrane curvature. Hence filaments create curvature in liposomes and sense it in cells.

## Rod-shape is lost in a global manner, but reforms locally

Together, the above data demonstrate that MreB filaments are sufficient to preferentially orient along the direction of greatest principal membrane curvature. In rod-shaped cells, this direction is along the rod circumference. As filaments move along their length, their orientation constrains the spatial activity of the PG synthetic enzymes such that new cell wall is inserted in a mostly circumferential direction (*Hayhurst et al., 2008*) to reinforce rod shape (*Chang and Huang, 2014*; *Yao et al., 1999*). While the ability of MreB filaments to orient in pre-existing rods can help explain how rod shape is maintained, we also wanted to understand how MreB filaments facilitate the de novo formation of rod shape. To explore this, we observed how cells interconvert between spheres and rods.

We first examined how rod shape fails, by growing our TagO-inducible strain at induction levels that produced rods and then reducing the $Mg^{2+}$ concentration to induce them to convert to spheres. This transition revealed that rods convert into round cells by continuous swelling: once a rod begins to widen, it continues to do so until reaching a fully spherical state with no reversion during the process (*Figure 5A*). Similar rod to sphere transitions could be attained by holding $Mg^{2+}$ constant while reducing TagO expression. Likewise, cells grown at intermediate TagO induction levels (8–12 mM) grew as steady state populations of interconnected rods and spheres, indicating that cells underwent repeated cycles of rod shape formation followed by reversion to spheres (*Figure 1D,F*). These results indicate that rod shape can be maintained only as long as the cell wall is sufficiently rigid to resist the internal osmotic pressure.

We next examined how rod shape forms from round cells. As the recovery of protoplasted *B. subtilis* is so infrequent that it has never been directly visualized (*Mercier et al., 2013*), we assayed how round cells with preexisting cell walls convert back into rods, using three systems: (1) re-inducing WTA expression within TagO-depleted, spherical cells, (2) holding TagO expression beneath the rod/sphere transition and increasing $Mg^{2+}$ levels, and (3) re-inducing Pbp2a expression in spherical, Pbp2a-depleted cells. In all three cases, rods reformed in a discrete, local manner; spheres did not form into rods by progressively shrinking along one axis, but rather, rods abruptly emerged from one point on the cell, growing more rapidly than the parent sphere (*Figure 5B*, *Figure 5—video 1* and *Figure 5—video 2*). This morphology is similar to the initial outgrowth of germinating *B. subtilis* spores (*Pandey et al., 2013*). We occasionally observed another mode of recovery, occurring when round cells were constrained, or divided into, ovoid or near-rod shapes. Once these near-rod shaped cells formed, they immediately began rapid, rod-like elongation along their long axis (*Figure 5—figure supplement 1A*).

## Rods form from local outward bulges, growing faster than non-rods

We focused on two salient features of the rod shape recoveries: (1) rod shape forms locally, most often at one point on the cell surface, and (2) once a rod-like region is formed, it appears self-

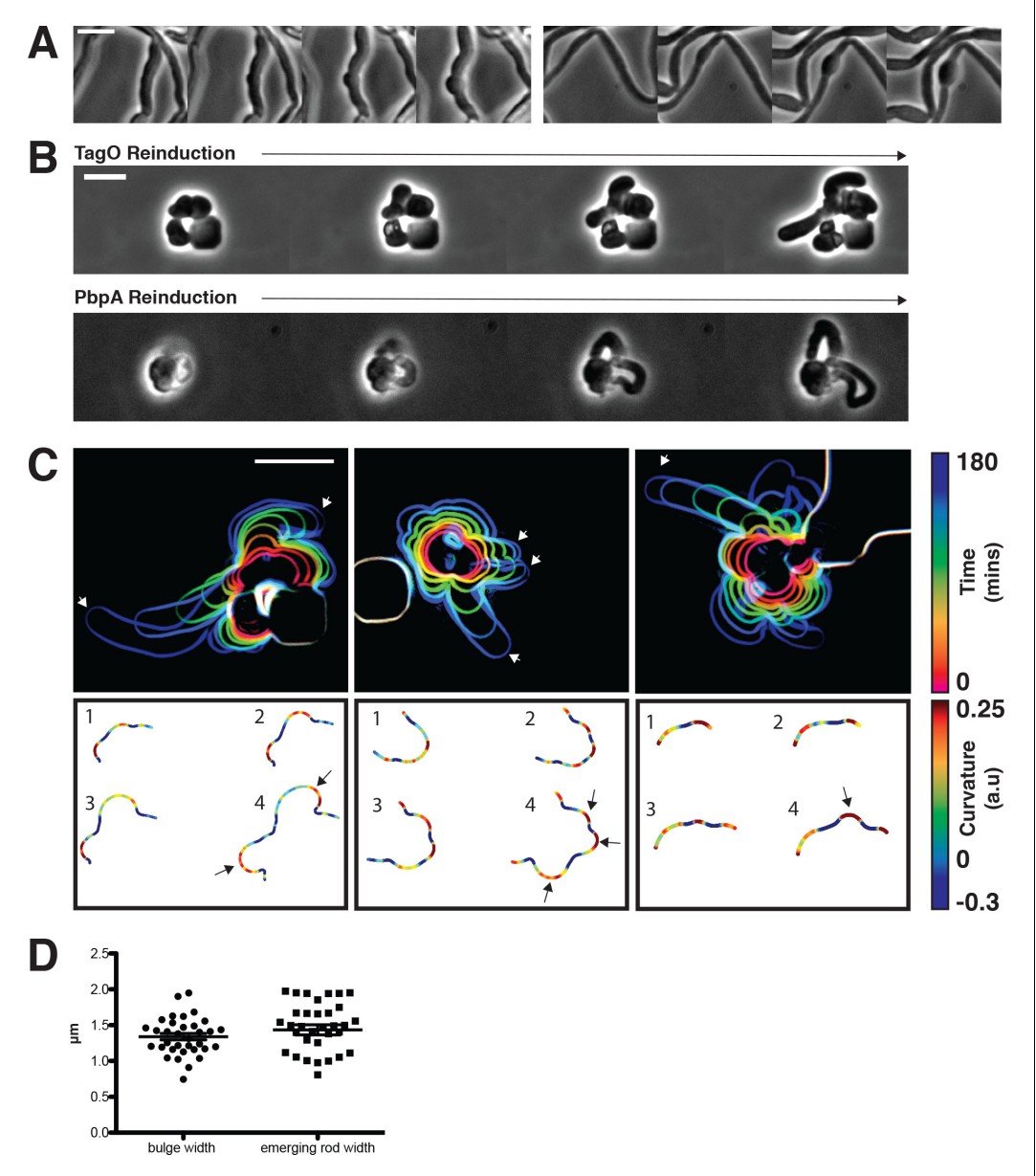

**Figure 5.** Sphere to rod transitions occur locally, then rapidly propagate. (**A**) Loss of rod shape proceeds continuously and without reversals, as shown by BEG300 cells grown in 12 mM xylose, shifted from 1 mM $Mg^{2+}$ to 100 μM $Mg^{2+}$ on a pad. Frames are 5 min apart. (**B**) Increases in expression of *tagO* or *pbpA* from depleted spherical cells causes cells to emit rapidly elongating rods from discrete points. (*Top*) BEG300 cells in 20 mM $Mg^{2+}$ were grown in 0 mM xylose for 4 hr, then transferred to a microfluidic chamber and grown in 0 mM xylose and 20 mM $Mg^{2+}$ for 1 hr. Following this, *tagO* expression was induced with 30 mM xylose at the first frame. (*Bottom*) BRB785 cells in 20 mM $Mg^{2+}$ were depleted of Pbp2a by growth in 0 mM IPTG for 4 hr. At the start of the frames, they were transferred to an agar pad containing 1 mM IPTG to induce *pbpA* expression. Frames are 30 min apart. (**C**) Plots of cell contours as cells recover from TagO depletion: (*top*) cell outlines are colored in time red to blue (0–180 min). White arrows indicate emerging rods; (*bottom*) heat maps of curvature show that rods emerge from small outward bulges (red) flanked by inward curvatures (blue). Black arrows indicate points where emerging rods form. (**D**) The width of initial bulges and the rods that emerge from them are highly similar, indicating the initial deformations may set the starting width of the rods. Error bars are SEM, n = 33. All scale bars are 5 μm.

DOI: https://doi.org/10.7554/eLife.32471.024

The following video, source data, and figure supplement are available for figure 5:

**Source data 1.** *Figure 5D* – Widths of the initial outward bulges (bulge) and eventual emerging rods (emerging rod) during multiple sphere to rod transitions by the re-induction of *tagO*.
DOI: https://doi.org/10.7554/eLife.32471.026

**Source data 2.** *Figure 5—figure supplement 1C* – 1/doubling time measured at various points in the xylose, magnesium phase space.

*Figure 5 continued on next page*

*Figure 5 continued*

DOI: https://doi.org/10.7554/eLife.32471.027

**Figure supplement 1.** Growth of rod-shaped and spherical cells measured by doubling times, and rod shape recovery.

DOI: https://doi.org/10.7554/eLife.32471.025

**Figure 5—video 1.** (top and middle) Timelapses showing the local recovery of rod shape upon TagO reinduction from depleted cells.

DOI: https://doi.org/10.7554/eLife.32471.028

**Figure 5—video 2.** Timelapse showing the loss and recovery of rod shape in cells with intermediate TagO levels, induced by removal and subsequent readdition of $MgCl_2$.

DOI: https://doi.org/10.7554/eLife.32471.029

reinforcing, both propagating rod shape and growing faster than adjacent or attached non-rod shaped cells.

We first wanted to understand how rod shape initiates de novo from spherical cell surfaces. By examining the initial time points of recoveries, we found that rods begin as small outward bulges: local regions of outward (positive Gaussian) curvature flanked by regions of inward (negative Gaussian) curvature (*Figure 5C*). These initial outward bulges showed a width distribution similar to that of the later emerging rods (*Figure 5D*). Once these bulges formed, they immediately began rapid elongation into nascent rods, which would then thin down to wild type width over time. Bulge formation and rod recovery were independent of cell division, as cells depleted of FtsZ still recovered rod shape (*Figure 5—figure supplement 1B*). Rather, these bulges appeared to arise randomly, evidenced by the fact that different cells produced rods at different times during WTA or Pbp2a repletion. We conclude that the appearance of a local outward bulge can act as the nucleating event of rod shape formation.

As emerging rods appeared to grow faster than adjacent round cells, we tested if the doubling times of rod-shaped cells were faster than those of non-rods by measuring the doubling times in our inducible TagO strain at different induction levels using both $OD_{600}$ measurements and single cell microscopy under steady state conditions (*Figure 3—figure supplement 1B*). This revealed a sharp transition in doubling time that matched the conditions of the rod/sphere transition: growth is slow when cells are spheres, yet greatly increases when cells are rods (*Figure 3—figure supplement 1B*, *Figure 5—figure supplement 1C*). Furthermore, the doubling times of recovering rods was similar to that of rods at steady state (*Figure 3—figure supplement 1B*).

We believe the lower doubling time of rods is likely due to cell shape and not another effect, such as the lack of WTAs, as (1) the doubling time of TagO-depleted cells confined in the microfluidic chambers matched that of wild type cells; and (2) both the doubling times and the boundary of the rod/sphere transition could be equivalently shifted by changing the $Mg^{2+}$ concentration (*Figure 1F*, *Figure 1—figure supplement 1*, *Figure 5—figure supplement 1C*, *Figure 5—video 2*). Combined, these results indicate that rod shape creates local, self-reinforcing regions that are poised for more rapid growth; once any small region of the cell approximates a rod shape, growth of the rod-like region is amplified, growing faster than other regions, and thereby outcompeting non-rod growth at the population level.

## Rod-shape formation correlates with aligned MreB motion

We next sought to determine what features distinguished rods from round cells. As the elongation of rod-shaped cells requires a sufficiently rigid cell wall (*Figures 1D–E* and *5A*), the localized, self-reinforcing formation of rods in our system could arise from either of the two major cell wall components, PG and WTAs: (1) the PG strands could be arranged such that they better reinforce rod shape (*Amir and Nelson, 2012*; *Chang and Huang, 2014*), or (2) WTAs could be preferentially incorporated into the emerging rods to stabilize them.

To assay the orientation of newly inserted cell wall, we imaged the motions of MreB as we induced TagO-depleted cells to recover into rods. This revealed that oriented MreB motion correlates with local shape: emerging rods displayed oriented MreB motion even at the initial points of their formation, while attached round parent cells displayed unaligned motion (*Figure 6A*, *Figure 6—video 1*). This demonstrates that oriented MreB motion correlates with local geometry and does not arise from a global, cell spanning change. We next examined the overall cellular distribution of MreB in recovering cells with confocal microscopy. This revealed that, immediately prior to

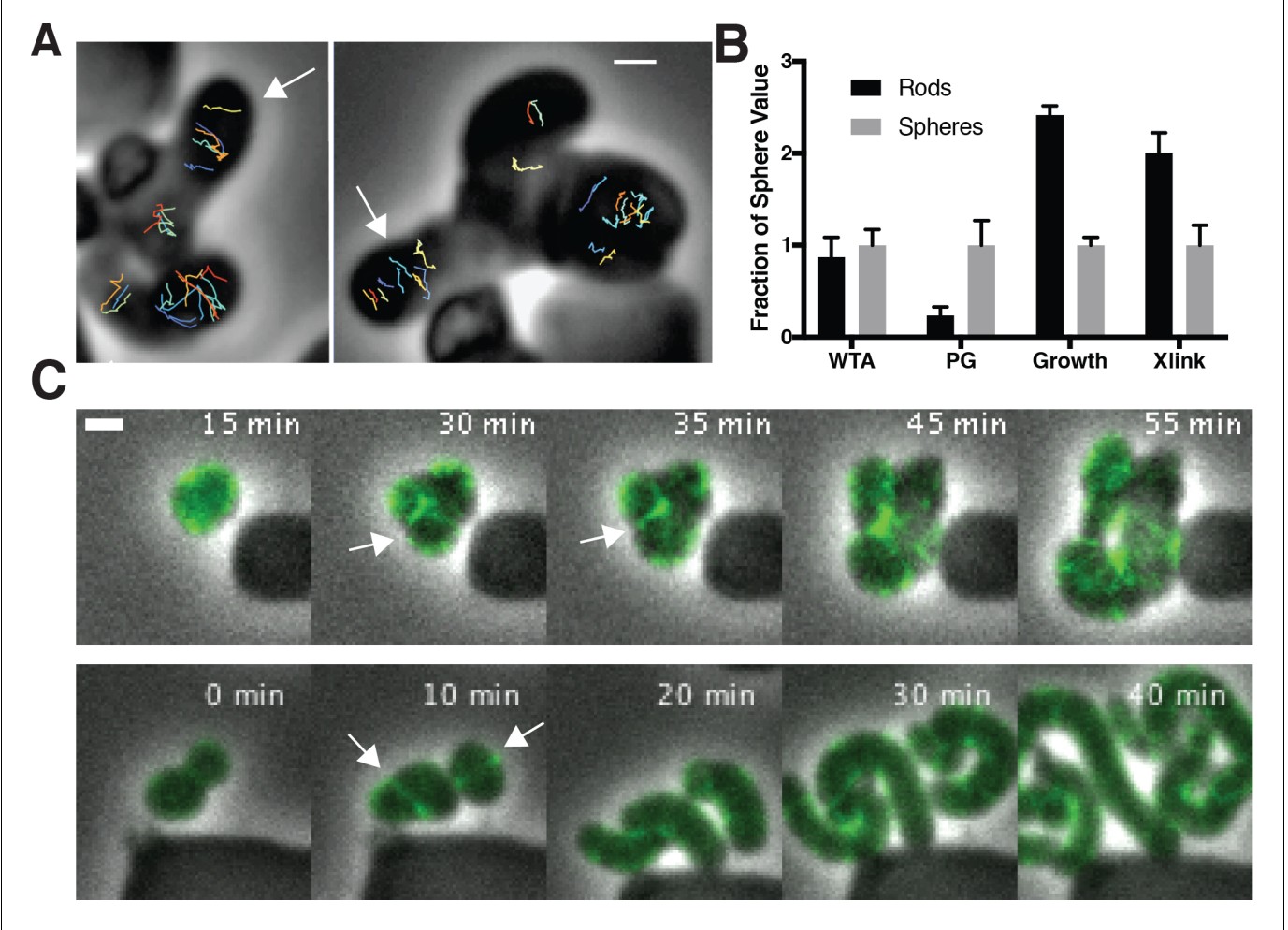

**Figure 6.** Oriented MreB motion is coupled to rod shape formation and preceded by transient MreB accumulation at the bulge neck. (**A**) (*left*) GFP-MreB trajectories during a sphere to rod transition. Emerging rods exhibit oriented MreB motion (white arrows) while attached round cells have unoriented motion. Scale bar is 1 μm. (**B**) During shape recoveries, immediately before rod emergence, MreB transiently accumulates in a bright ring where the bulge connects to the parent sphere. See also *Figure 5—figure supplement 1D*. Scale bar is 2 μm. (**C**) Fold change between spheres and rods in the teichoic acid incorporation and growth rate, assayed by $OD_{600}$ measurements (inducible TagO with 30 mM xylose in LB with 20 mM $Mg^{2+}$). Error bars are SD. See also *Figure 3—figure supplement 1B* and *Figure 6—figure supplement 1B*.
DOI: https://doi.org/10.7554/eLife.32471.030

The following video, source data, and figure supplements are available for figure 6:

**Source data 1.** *Figure 6C* – Ratio of wall teichoic acid localization (WTA) as measured by ConA staining, and single cell growth rates in rod-shaped and spherical cells.
DOI: https://doi.org/10.7554/eLife.32471.033

**Source data 2.** *Figure 6—figure supplement 1E* – Frequency distributions of the velocity and alpha values of TagT, TagU, TagV, MreB and Pbp2A tracks.
DOI: https://doi.org/10.7554/eLife.32471.034

**Figure supplement 1.** ConA staining of cells during rod shape recovery and localization of WTA ligases.
DOI: https://doi.org/10.7554/eLife.32471.031

**Figure supplement 2.** Single molecule tracking of WTA ligases.
DOI: https://doi.org/10.7554/eLife.32471.032

**Figure 6—video 1.** Timelapse of rod shape recoveries showing that circumferential MreB-GFP motion.
DOI: https://doi.org/10.7554/eLife.32471.035

**Figure 6—video 2.** Timelapse movies showing that msfGFP fusions to the teichoic acid ligases TagT, TagU, and TagV do not move circumferentially.
DOI: https://doi.org/10.7554/eLife.32471.036

**Figure 6—video 3.** Timelapse and tracking overlays of single molecule imaging of HaloTag-JF549 tagged MreB, Pbp2a, TagT, TagU, and TagV, all expressed from at the native promoter and locus, and sparsely labeled using low concentrations of HaloTag-JF549 dye.
DOI: https://doi.org/10.7554/eLife.32471.037

rod emergence, MreB transiently accumulated in a bright ring oriented perpendicular to the direction of rod emergence, most often occurring at the interface of the bulge and the round cells (*Figure 6B*, *Figure 5—figure supplement 1D*).

To observe whether both spheres and rods inserted new PG during the process of rod shape recovery, we used fluorescent D-amino-acids (FDAAs), which crosslink into newly inserted cell wall. We grew TagO-depleted cells in a microfluidic device in the presence of HADA, then switched the media to contain Cy3B-ADA as we re-induced TagO expression. During rod emergence, the old cell wall signal (HADA) remained in the sphere, while the emerging rod was almost entirely composed of new (Cy3B-ADA) material, confirming the discrete nature of rod shape recovery. However, the attached spheres also incorporated Cy3B-ADA, indicating PG synthesis occurs in both rods and spheres during recovery (*Figure 5—figure supplement 1E*).

The local reinforcement of rod shape in recovering cells could arise from preferential incorporation of the cell wall rigidifying WTAs. As the WTA ligases have been reported to interact with MreB (*Kawai et al., 2011*), we tested if rod shape correlated with increased WTA accumulation in emerging rods. To test this, we labeled recovering cells with fluorescently labeled lectins that specifically bind to WTAs (*Figure 6—figure supplement 1A*). Following TagO reinduction, WTAs in recovering cells were (*Figure 6—figure supplement 1B*), equally present in the cell walls of both rods and spheres (*Figure 6C*). To test if the WTA ligases move with MreB, we created GFP fusions to these proteins at their native locus and examined their dynamics with TIRFM. We were unable to observe any of the circumferential motions expected if the WTA ligases moved with MreB; instead they appeared to be rapidly diffusing on the membrane (*Figure 6—figure supplement 1C*, *Figure 6—video 2*, Appendix 2). For further confirmation, we tracked the single molecule motions of these ligases, using JF549 labeled HaloTag fusions expressed at the native locus and promoter. We did not observe any directional motions of the ligases that would indicate they move or associate with MreB and Pbp2A (*Figure 6—figure supplement 2D–E*, *Figure 6—video 3*).

In summary, these data gives new insights into what properties of the cell wall can be modulated to create and stabilize rod shape: rod shape is not formed by preferential localization of teichoic acids to rods, and both spheres and rods incorporate PG before and during rod shape recovery, in line with reports that PG synthesis is unchanged by the inhibition of WTA synthesis (*Pooley et al., 1993*). Rather, the only differences we detected between rod shaped and round cells were increased growth rates and oriented MreB motion. We note that as WTAs have been shown to affect hydrolase activity (*Kasahara et al., 2016*), their depletion may cause other rod-shape inhibiting PG abnormalities that we cannot observe.

## Discussion

The above experiments give new insights into possible mechanisms by which MreB guides cell wall synthesis to create rod shape. First, the curved ultrastructure of MreB filaments causes them to orient and move along the direction of greatest membrane curvature, inserting material in that direction. Second, both the formation and propagation of rod shape occurs by a local, self-reinforcing process: once a local region of rod shape forms, it propagates more rod shape. Finally, as far as we can determine, the primary differences between the growth of rods and non-rods is the circumferential orientation of MreB motion.

Combined, these findings indicate that MreB filaments function as curvature-sensing rudders, a property that allows them to organize cell wall synthesis so that it builds rod shape: MreB filaments orient along the greatest membrane principal curvature, thereby constraining the activity of the associated PG synthases so that, as they move via their synthetic activity, they deposit glycans oriented in the direction of that curvature, an arrangement of material that further reinforces rod shape. Even during the initial stages of rod shape formation, oriented MreB motion and rod shape always coincide, and the intrinsic curvature of MreB filaments suggests these properties cannot be uncoupled. This coupling appears to be an essential component of the Rod system: by linking filaments that orient along the greatest principal curvature to cell wall synthetic enzymes reinforcing that curvature, the Rod complex may function as a local, self-organizing system that allows bacteria to both maintain rod shape and also establish rod shape de novo.

In established rods, we propose that MreB maintains and propagates rod shape via a local feedback between existing shape, filament orientation, and subsequent shape-reinforcing PG synthesis.

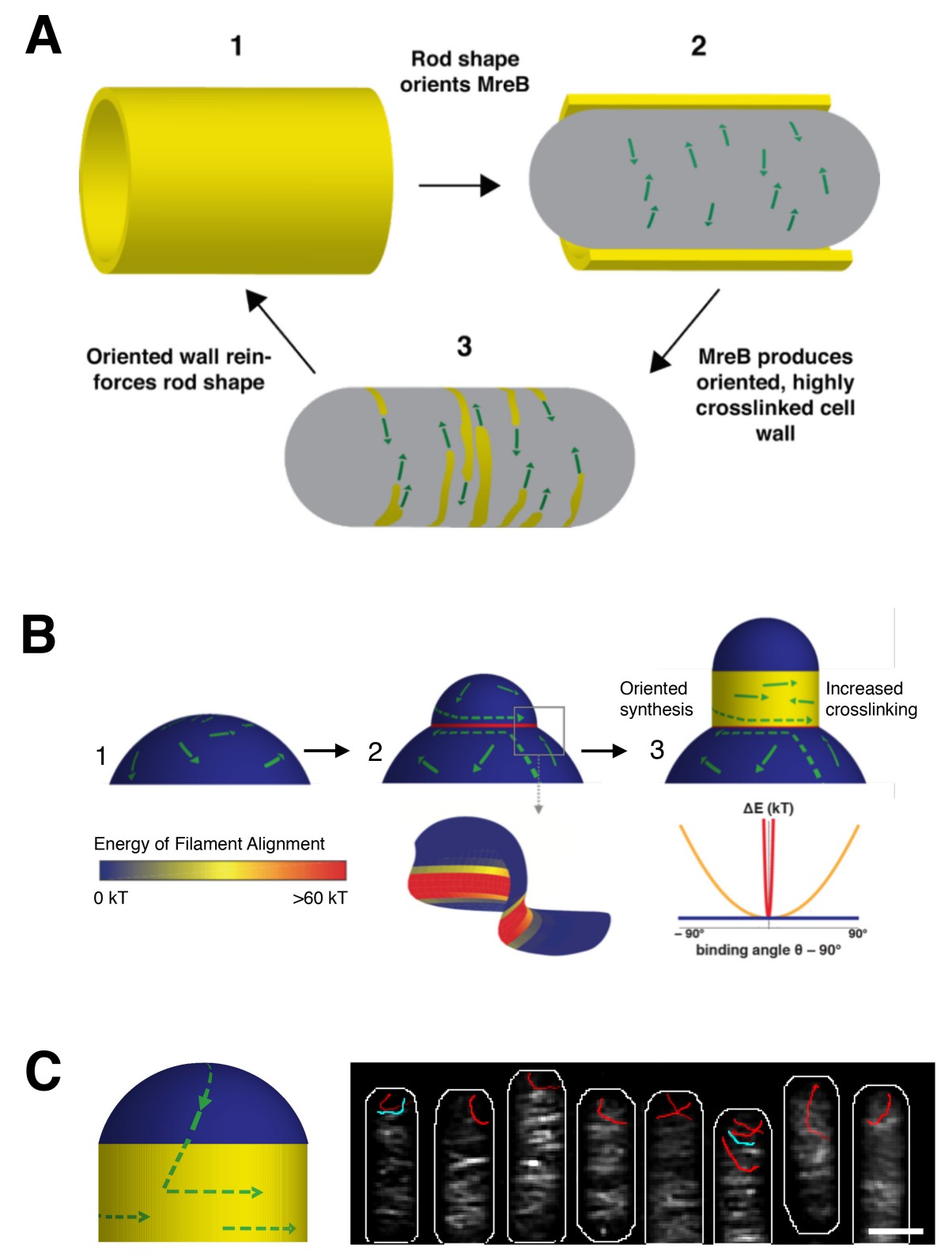

**Figure 7.** Model for how MreB filament orientation along the greatest curvature can both maintain and establish rod shape. (**A**) Rod-shaped cells present a single curved axis along which MreB filaments orient (1). This orientation determines the direction of MreB motion (2), thus orienting the insertion of new cell wall material around the rod (3). Thiscircumferential arrangement of cell wall material reinforces rod shape (1), leading to more aligned MreB filaments, thus creating a local feedback between the orientation of MreB filaments, oriented cell wall synthesis, structural integrity of the

*Figure 7 continued on next page*

*Figure 7 continued*

rod, and overall rod shape. (B) MreB motion in spherical cells is isotropic (1), but the introduction of an outward bulge (2, upper) creates a curved geometry (red) at the neck of the bulge that initiatesrod shape formation. Due to the high energy of alignment in this region, (two lower and chart), any filaments that encounter the neck of the bulge would prefer to align to point around the neck rather than cross it, creating a ring-shaped region of aligned MreB motion that nucleates rod formation.Repeated rounds of oriented synthesis around the ring could initiate the elongation of a rod from the initial bulge site (3), beyond which rod shaped elongation would be self-sustaining. Colors correspond to the difference of alignment energies along the two principal curvatures at the negatively-curved neck region (red), flat regions with one dimension of curvature (yellow), and the positively-curved sphere/bulge (blue). (C) Model for how filament orientation can explain the absence of MreB at poles. (*Left*) MreB filaments, predominantly oriented to move around the rod width, rarely enter the poles. As filaments move directionally, any filaments within the poles will eventually translocate out of them into the cell body, where the difference in curvatures would cause them to reorient to move around the circumference. (*Right*) Average intensity projection of 70 s of SIM TIRF imaging of mNeonGreen-MreB at the poles of *B. subtilis.* The intensity projection (white) shows that most filaments rarely enter the poles. However, a few filaments are observed entering (blue traces) or leaving (red traces) the poles. Notably, filaments that move out of the poles reorient their motion to move around the rod circumference when they enter the cell body. See corresponding Movie *Figure 7-video 1*. Scale bar is 1 μm.

DOI: https://doi.org/10.7554/eLife.32471.038

The following video, source data, and figure supplement are available for figure 7:

**Source data 1.** *Figure 7—figure supplement 1A* – Frequency distributions of the velocity and alpha values for tracks of wildtype *E. coli* MreB and of MreB(S14A) in Δ*rodZ* cells grown in (i) LB at 37° and (ii) M63 at 37°C.

DOI: https://doi.org/10.7554/eLife.32471.040

**Figure supplement 1.** MreB(S14A) moves directionally and processively in *E. coli* lacking RodZ.

DOI: https://doi.org/10.7554/eLife.32471.039

**Figure 7—video 1.** SIM TIRF timelapse of mNeonGreen-MreB at cell poles, showing that filaments moving out of the poles reorient when they move into the cell body.

DOI: https://doi.org/10.7554/eLife.32471.041

**Figure 7—video 2.** TIRF time lapse imaging of *E. coli* strain RM478 (Δ*rodZ, mreBS14A-msfGFP*).

DOI: https://doi.org/10.7554/eLife.32471.042

As rod-shaped cells grow (*Figure 7A1*), MreB filaments orient along the more curved axis around the bacterial width (*Figure 7A2*). Because MreB filaments always translocate along their length, filament orientation constrains the activity of the associated PG synthases such that new cell wall is inserted in bands predominantly oriented around the width of the rod (*Figure 7A3*). This circumferential insertion of glycan strands would yield an anisotropic arrangement of material that reinforces rod shape (*Figure 7A1*), allowing for continued MreB filament orientation. Once the rod is formed, this feedback loop allows robust shape maintenance, as long as the material within the rod sidewalls is sufficiently rigid to withstand the stresses arising from osmotic pressure.

While the MreB guided circumferential insertion of cell wall material is expected to reinforce rod shape (*Amir and Nelson, 2012*; *Chang and Huang, 2014*; *Baskin, 2005*), it must be noted that it is not sufficient, as other factors are also needed: WTA-depleted cells become round (*D'Elia et al., 2006*) (*Figure 1D*) even under conditions where they contain oriented MreB motions (*Figure 3—figure supplement 1A*). Based on both past work (*Matias and Beveridge, 2005*; *Kern et al., 2010*) and our data (*Figure 1D–F*, *Figure 1—figure supplement 1*, *Figure 5A*, *Figure 5—video 2*), we believe that WTAs, in combination with PG, work to coordinate $Mg^{2+}$ to increase the overall cell wall stiffness, thus allowing regions of oriented insertion to maintain and propagate their shape.

The coupling between the local sensing and reinforcement of differences in principal curvature could also facilitate the de novo formation of rod shape. In round cells, there is no difference in principal curvatures (*Figure 7B1*), so MreB motion is isotropic. Rods do not form by squeezing these round cells across one axis, rather we observe them forming by the amplification of local rod-like regions. Given the rapid timescale of our recoveries, the Rod system appears poised to propagate any shape variations that create curved regions favorable to oriented MreB motion: as soon as regions of oriented motion are established, they self-propagate and elongate, creating more rod shape and thus continued oriented MreB motion.

It remains to be determined how these local self-propagating regions form. In our recoveries, the most common shape variation we observe preceding rod emergence is small outward bulges flanked by regions of inward curvature (*Figure 5C*). What establishes these rod nucleating geometries remains to be determined: they could arise from stochastic fluctuations in membrane curvature, or they may form by the actions of some factor acting upstream of the Rod complex to create these

geometries. Thus far we have only been able to determine that cell division is not required to form these bulges (*Figure 5—figure supplement 1B*). Another possibility is that these bulges arise from local changes in cell wall stiffness, as local softening of the cell wall has been observed to induce the rod shaped outgrowth of germinating fission yeast spores (*Bonazzi et al., 2014*).

Whether these outward bulges arise from physical effects or by the action of upstream factors, the geometry at their interface may play a nucleating role in rod shape formation. In three dimensions, the intersection of the bulge and the sphere creates a geometry that can establish a zone of aligned filaments: while both the parent sphere and the outward bulge have principal curvatures in the same direction (positive Gaussian curvature), the intersection of the sphere and bulge creates an interface with large differences in principal curvatures, one inward, and one outward (negative Gaussian curvature). Upon entering these negatively curved regions it is energetically unfavorable for the inwardly curved MreB filaments to deviate from their preferred binding orientation, as our modeling indicates that this region presents a steep well in the energy profile for alignment (*Figure 7B2* and Appendix 1). Thus, filaments moving into this rim from either side would reorient to move along it, creating a concentrated band of filaments moving around the bulge neck. This concentrated ring of oriented MreB filaments may then construct a local region of rod shape that subsequently self-propagates into an emerging rod (*Figure 7B3*). In support of this hypothesis, immediately preceding rod shape formation, we observe concentrated bands of MreB transiently appearing at the neck of emerging bulges (*Figure 6B*, *Figure 5—figure supplement 1D*). Likewise, similar patterns of MreB accumulation at points of negative Gaussian curvatures have been observed in recovering *E. coli* L-forms (*Billings et al., 2014*)

The common observation of MreB accumulation at the necks of rod-producing bulges in both *E. coli* and *B. subtilis* hints at a solution to an outstanding discrepancy: Why do inwardly curved MreB filaments show an enriched localization at negative Gaussian curvatures (inward dimples or the more curved faces of bent cells) (*Billings et al., 2014*; *Renner et al., 2013*; *Ursell et al., 2014*), and how is this enrichment maintained as filaments are constantly moving around the cell, in and out of these curvatures? The finding that MreB filaments align along the greatest curvature poses a solution: If the sharpness of filament alignment changes in response to the difference in principal curvatures in each region they pass through, areas of negative Gaussian curvature may act as points that focus the subsequent motion of filaments so that, on average, more filaments pass through these regions. Thus, the enrichment or depletion of MreB at observed different curvatures may not arise from different binding affinities, but rather from filaments being steered more toward or away from given areas as they move through different curvatures.

Likewise, the tendency of MreB to align and move along the direction of greatest principal curvature may also explain the absence of MreB at cell poles. Consistent with our model for binding, we observed MreB filaments bound to the round poles of liposome tubes in vitro (*Figure 4—figure supplement 1A*). In the cell, however, MreB filaments move directionally, and thus filaments oriented around the rod are less likely to move into the poles. Filaments that do move into the symmetrically curved poles would, regardless of their orientation, quickly translocate back out of them into the cylindrical cell body where they would reorient along the single direction of curvature (*Figure 7C*). In support of this model, Structured Illumination Total Internal Reflection Microscopy (SIM-TIRF) of mNeonGreen-MreB revealed multiple examples of filaments moving out of the poles. As predicted, when these filaments entered the cell body, they reoriented their motion to move around the rod circumference (Movie *Figure 7-video 1*, *Figure 7C*).

We note that, while rod-shaped cells show both oriented MreB motion and an increased rate of growth relative to spheres, it is unlikely these phenomena are mechanistically linked. Rather, the decreased rate of growth of non-rods likely arises from a downstream effect of the lack of rod shape on cell physiology. Indeed, many spatial processes in *B. subtilis*, such as chromosome segregation and division site selection, read out and partition along the long axis established by rod shape (*Jain et al., 2012*). Thus, the slower doubling times observed in non-rod shaped cells may arise from the improper spatial organization of these processes, or stress responses to this spatial disarray.

As the curvature of membrane-bound MreB filaments (200 nm) observed in vitro (*van den Ent et al., 2014*) is much greater than the cell diameter (900 nm), these findings suggest that the curvature of MreB filaments does not define a specific cell radius; rather filament curvature acts to orient PG synthesis to maintain (*Harris et al., 2014*) or reduce cell diameter. If the curvature of MreB filaments reflects the smallest possible cell diameter, bacterial width may be specified by opposing

actions from the two spatially distinct classes of PG synthases: a decreasing, 'thinning' activity from the action of MreB and its associated SEDS family PG synthases, and an increasing 'fattening' activity from the non-MreB associated Class A PG synthases.

In contrast to our model, a previous study in *E. coli* concluded that directional MreB motion was not required for rod shape. This conclusion was based on the observation that cells lacking RodZ are still rod-shaped, even while they observed no directional motion of GFP-MreB(S14A) filaments (*Morgenstein et al., 2015*). We acquired this strain from the Gitai lab (Δ*rodZ*, *mreBS14A-msfGFP*), sequenced the mutations, and examined the motions of MreB with TIRFM. Surprisingly, we observed directional motions of mreBS14A-msfGFP filaments in almost every cell (*Figure 7—video 2*) and subsequent analysis indicated these movements were processive (*Figure 7—figure supplement 1*). We could observe directional motion across a range of growth conditions, although at lower temperatures, or in less rich media (the conditions used for imaging in Morgenstein et al), it appeared that smaller fraction of the mreBS14A-msfGFP moved directionally. Thus, in contrast to the conclusions of (*Morgenstein et al., 2015*): (1) the directional motion of MreB has not yet been uncoupled from rod shape, and (2) RodZ is not required for directional MreB motion. We suspect that the conclusions of Morgenstein et al. arose from growing the cells in rich media at high temperatures, then imaging the cells in less rich media at low temperatures, where only a small fraction of the MreB filaments move directionally.

## Conclusion

To construct regular, micron-spanning shapes made of covalently crosslinked material, nature must devise strategies for coordinating the activities of disperse, nanometer-scale protein complexes. This work reveals that the role of MreB in creating rod shape is to locally sense and subsequently reinforce differences in principal curvatures. The local, short-range feedback between differences in curvature, MreB orientation, and shape-reinforcing cell wall synthesis could provide a robust, self-organizing mechanism for the stable maintenance and rapid reestablishment of rod shape, allowing the local activity of short MreB filaments to guide the emergence of a shape many times their size.

## Materials and methods

**Key resources table**

| Reagent type (species) or resource | Designation | Source or reference | Identifiers |
|---|---|---|---|
| strain (*Bacillus subtilis*) | BCW51 | This work | *ycgO::Pxyl-tagO, tagO::erm, amyE::sfGFP-mreB, sinR::phleo* |
| strain (*B. subtilis*) | BCW61 | This work | *tagE::erm* |
| strain (*B. subtilis*) | BCW72 | This work | *yvhJ::PxylA-mazF (cat)* |
| strain (*B. subtilis*) | BCW77 | This work | *ywtF::PxylA-mazF (cat)* |
| strain (*B. subtilis*) | BCW78 | This work | *ywtF::msfGFP-ywtF* |
| strain (*B. subtilis*) | BCW79 | This work | *yvhJ::msfGFP-yvhJ* |
| strain (*B. subtilis*) | BCW80 | This work | *lytR::PxylA-mazF (cat)* |
| strain (*B. subtilis*) | BCW81 | This work | *lytR::msfGFP-lytR* |
| strain (*B. subtilis*) | BCW82 | This work | *tagO::erm, ycgO::PxylA-tagO, amyE::Pspac-gfp-mreB (spec), dacA::kan* |
| strain (*B. subtilis*) | BDR2061 | (*Carballido-López and Errington, 2003*) 10.1016/S1534-5807(02)00403–3 | *amyE::PxylA-gfp-mbl (spec), mbl⍭pMUTIN4 (erm)* |
| strain (*B. subtilis*) | BEB1451 | (*D'Elia et al., 2006*) 10.1128/JB.01336–06 | *hisA1 argC4 metC3 tagO::erm* |
| strain (*B. subtilis*) | BJS18 | (*Defeu Soufo and Graumann, 2004*) 10.1038/sj.embor.7400209 | *amyE::PxylA-gfp-mbl (spec)* |
| strain (*B. subtilis*) | BMD61 | This work | *mbl::mbl-msfGFP (spec)* |

*Continued on next page*

*Continued*

| Reagent type (species) or resource | Designation | Source or reference | Identifiers |
|---|---|---|---|
| strain (*B. subtilis*) | BRB785 | (*Garner et al., 2011*) 10.1126/science.1203285 | *yhdG::Pspank-pbpA (phleo), pbpH::spec, pbpA::erm, mblΩPxylA-gfp-mbl (cat)* |
| strain (*B. subtilis*) | BRB786 | (*Garner et al., 2011*) 10.1126/science.1203285 | *yhdG::PspanK-pbpA (phleo), pbpH::spec, pbpA::cat, yvbJ::PxylA-gfp-mreB (erm)* |
| strain (*B. subtilis*) | BEG202 | (*Schirner et al., 2015*) 10.1038/nchembio.1689 | *ΔtagO::erm amyE::Pxyl-gfp-mbl (spec)* |
| strain (*B. subtilis*) | BEG203 | (*Schirner et al., 2015*) 10.1038/nchembio.1689 | *ΔtagO::erm amyE::Pxyl-gfp-mreB (spec)* |
| strain (*B. subtilis*) | BEG281 | This work | *ycgO::PxylA-tagO* |
| strain (*B. subtilis*) | BEG291 | This work | *tagO::erm, ycgO::PxylA-tagO,* |
| strain (*B. subtilis*) | BEG275 | (*Meeske et al., 2016*) 10.1038/nature19331 | *amyE::Pspac-gfp-mreB (spec)* |
| strain (*B. subtilis*) | BEG300 | This work | *tagO::erm, ycgO::PxylA-tagO, amyE::Pspac-gfp-mreB (spec),* |
| strain (*B. subtilis*) | BRB4282 | (*D'Elia et al., 2006*) 10.1128/JB.01336–06 | *168 trpC2 ΔtagO::erm* |
| strain (*B. subtilis*) | bAB343 | This work | *ftsZ::mNeonGreen-15aa-ftsZ, amyE::spc-Pspank-mciZ, ycgO::cat-Pxyl-tagO, tagO::erm* |
| strain (*B. subtilis*) | bAB327 | This work | *ftsZ::mNeonGreen-15aa-ftsZ, amyE::Phyperspank-minCD, ycgO::Pxyl-tagO, tagO::erm* |
| strain (*B. subtilis*) | bAB388 | This work | *ftsZ::mNeonGreen-15aa-ftsZ, amyE::Physpank-ftsA ycgO::cat-Pxyl-tagO, tagO::erm* |
| strain (*B. subtilis*) | bYS09 | This work | *mreB::mreB-40aa-mNeonGreen* |
| strain (*B. subtilis*) | bYS40 | This work | *mreB::HaloTag–MreB* |
| strain (*B. subtilis*) | bYS201 | This work | *HaloTag-Pbp2A::cat* |
| strain (*B. subtilis*) | bAB198 | This work | *tagT::erm-Pxyl-HaloTag-15aa-tagT* |
| strain (*B. subtilis*) | bAB197 | This work | *tagV::erm-Pxyl-HaloTag-15aa-tagV* |
| strain (*B. subtilis*) | bAB196 | This work | *tagU::erm-Pxyl-HaloTag-15aa-tagU]* |
| strain (*Escherichia coli*) | NO50 | (*Ouzounov et al., 2016*) 10.1016/j.bpj.2016.07.017 | *E. coli - msfGFP-MreB(sw)* |
| strain (*E. coli*) | RM478 | (*Morgenstein et al., 2015*) 10.1073/pnas.1509610112 | *E. coli - ΔrodZ (cam), mreBS14A-msfGFPSW (kan)* |
| software, algorithm | uTrack | (*Jaqaman et al., 2008*) 10.1038/nmeth.1237 | |
| software, algorithm | Trackmate | (*Tinevez et al., 2017*) 10.1016/j.ymeth.2016.09.016 | |
| software, algorithm | Morphometrics | (*Ursell et al., 2017*) 10.1186/s12915-017-0348-8 | |

## Overnight culture growth

All *B. subtilis* strains were prepared for experimentation as follows: strains were streaked from −80°C freezer stocks onto lysogeny broth (LB) agar plates. Following >12 hr of growth at 37°C, single colonies were transferred to serially diluted overnight bulk liquid cultures in LB supplemented with 20 mM magnesium chloride, placed on a roller drum agitating at 60 rpm, and grown at 25°C. After >12 hr growth to $OD_{600}$ <0.6, these starter cultures were transferred to or inoculated into subsequent growth conditions. All strains with *tagO* under inducible control were grown overnight in the presence of 30 mM xylose unless otherwise noted.

## Single cell and bulk doubling time measurements

For the experiments in *Figure 6C* and *Figure 3—figure supplement 1B*, BEG300 cells were inoculated in the indicated medium (LB with 20 mM MgCl$_2$ unless otherwise stated) from logarithmic phase overnights; 'rods' were grown from a low dilution with 30 mM xylose, and 'spheres' were grown with 0 mM xylose.

For bulk culture doubling time measurements, doubling times were calculated from the slope of a graph of time vs. dilution for a succession of serial dilutions of a given strain. Time, the dependent variable, was taken as the time for a given dilution to pass the OD cutoff of OD$_{600}$ = 0.20. In place of technical replicates, a large number of replicates were performed on a continuous gradient of xylose induction, showing a consistent trend between the extreme values depicted in the figure.

Single cell measurements were made in three ways.

1. Spherical and rod-shaped cells were allowed to grow on agarose pads made with LB supplemented with 20 mM MgCl$_2$. 30 mM xylose was added to agarose pads for rod-shaped cells. Cells were imaged every 2 min for 4 hr with phase contrast microscopy as described in the section below. N = 32 for spherical cells and N = 51 for rod-shaped cells.
2. ii)Spherical and rod-shaped cells were grown in the CellASIC B04A plate in LB supplemented with 20 mM MgCl$_2$ for spherical cells and LB supplemented with 20 mM MgCl$_2$ and 30 mM xylose for rod-shaped cells. The CellASIC unit confined the cells in the Z dimension due to the fixed height of the ceiling. Cells were imaged every 10 min for 2 hr using phase contrast microscopy as described in the section below. N = 8 for spherical cells and N = 10 for rod-shaped cells.
3. For cells growing in the mother machine microfluidic device (see below), the expansion of the cell length along the channel was quantified using FIJI (*Schindelin et al., 2012*); only the cells closest to the mouth of the channel were counted. Since cells were always oriented along the length of the channel (see *Figure 3A*, *Figure 3—figure supplement 1A*), changes in expansion in this dimension accounted for all growth. Multiple cycles of expansion from a single cell were counted; for a given experiment, all such expansions during the observation time were averaged together. In the published experiment, there were four such expansions with at least seven data points each. The published experiment is representative independent observation periods.

## Imaging – phase contrast microscopy

Phase contrast images were collected on a Nikon (Tokyo, Japan) Ti microscope equipped with a 6.5 μm-pixel Hamamatsu (Hamamatsu City, Japan) CMOS camera and a Nikon 100x NA 1.45 objective. Cells were collected by centrifugation at 6000 x *g* for 2 min and re-suspended in the original growth medium. Unless otherwise specified, cells were then placed on No. 1.5 cover glass, 24 × 60 mm, under a 1 mm thick agar pad (2–3% agar) containing LB supplemented with 20 mM magnesium chloride. Unless otherwise noted, all cells were imaged at 37°C on a heated stage.

## Imaging – MreB particle tracking

Images were collected on a Nikon TI microscope with a 6.5 μm-pixel CMOS camera and a Nikon 100x NA 1.45 objective. Cells of strain BEG300 were grown overnight in LB supplemented with 30 mM xylose, 20 mM magnesium chloride, 1 μg/mL erythromycin, and 25 μg/mL lincomycin at 25°C at the specified xylose concentrations. 11 μM isopropyl β-D-1-thiogalactopyranoside (IPTG) was added to induce GFP-MreB and the cells were shifted to 37°C and allowed to grow for 2 hr before imaging. Cells of strain BEG202 (Δ*tagO*) with GFP-Mbl under a xylose-inducible promoter were grown overnight at 25°C in LB supplemented with 20 mM magnesium chloride and 0.125 mM xylose, and shifted to 37°C for 2 hr before imaging. Cells were placed on cleaned glass coverslips thickness No. 1.5, as described in the next section. 3–6% agar pads were prepared in LB supplemented with 20 mM magnesium chloride, 11 μM IPTG and the desired concentration of xylose. Images were collected for 3 min at 1 or 2 s intervals, as specified. 30–50 cells were imaged in a day, and the experiments were repeated on at least one other day to test for technical variation.

## Imaging – slide preparation

Coverslips were sonicated in 1 M KOH for 15 min, followed by five washes with water. Coverslips were washed twice with 100% ethanol, and then sonicated in 100% ethanol, followed by one more wash in 100% ethanol. They were stored in ethanol and dried for 10 min before use.

## Imaging – spinning disk confocal

Images were collected on a Nikon TI microscope with a Hamamatsu ImagEM (EM-CCD) camera (effective pixel size 160 nm) and Nikon 100x NA 1.45 TIRF objective. Z stacks were obtained at 0.2 μm slices. Total image depth was 3 μm. Only the top 3 slices of the cell were used in maximum intensity projections in *Figure 3E*.

## Imaging – structured illumination total internal reflection microscopy (Sim TIRF) of cell poles

Cells were prepared as in 'Imaging – MreB particle tracking' above. Cells were placed under an agar pad in a MatTek (Ashland, Massachusetts) dish for imaging. Images were collected on an DeltaVision OMX SR Blaze system in SIM TIRF mode, using an Edge 5.5 sCMOS camera (PCO) and a 60x objective. 75msec exposures from a 488 nm diode laser were used for each rotation. Spherical aberration was minimized using immersion oil matching. Raw images were reconstructed using Softworx (Applied Precision).

## Image processing

All image processing unless otherwise specified was performed in FIJI (*Schindelin et al., 2012*). Images used for particle tracking were unaltered, except for trimming five pixels from the edges of some videos to remove edge artifacts detected by the tracking software. Phase contrast images and fluorescent images of protoplasts were adjusted for contrast. Phase contrast images presented in the manuscript collected from cells in the custom microfluidic device, which did not undergo quantitative processing, were gamma-adjusted ($\gamma = 1.5$) to compensate for changes in brightness occurring at the device's feature borders; such processing was not used for growth quantification. Images were background-subtracted for viewing purposes; unaltered images were used for quantitative processing in all cases.

## Microfluidics

The custom microfluidic setup used to confine cells in *Figure 3A–C*, *Figure 3—figure supplement 1A*, and *Figure 3—video 1* was previously described in *Norman et al. (2013)*. Briefly, a polydimethylsiloxane slab with surface features was bonded to a 22 × 60 mm glass coverslip by oxygen plasma treatment followed by heating to 65°C for >1 hr. The features in our setup differed from those described in *Norman et al. (2013)*, particularly in the omission of a second, wider layer in the cell chambers, which enhanced growth at timescales beyond that of our experiments. Syringes containing growth medium were connected to the microfluidic features using Tygon tubing stainless steel dispensing needles (McMaster Carr Supply Company, Elmhurst, Illinois). Medium was supplied to cells at a constant rate of 2–5 μL/min using automatic syringe pumps. Imaging was carried out using phase contrast microscopy as described above. For the microfluidics experiments in *Figures 5* and *6* and *Figure 5—video 1* (top), *Figure 5—video 2*, and *Figure 6—video 1*, the CellASIC platform from Merck Millipore (Billerica, Massachusetts) was used with B04A plates.

## Cell confinement experiments

The cell confinement experiment in *Figure 3A–C* was conducted by first loading cells into the chamber: BEG300 cells were grown to stationary phase (OD$_{600}$3.0–5.0) in LB supplemented with 20 mM magnesium chloride, passed through a 5 μm filter, and concentrated 100-fold before loading in the custom-made microfluidic device. Both phase contrast and fluorescent imaging were performed as described in the 'Imaging' section above. For observing MreB movement, MreB-GFP expression was induced with 50 μM IPTG upon loading into the microfluidic chamber, and cells were imaged every 2 s with a camera exposure time of 300 ms. The biological phenomena depicted in *Figure 3A–C*, *Figure 3—figure supplement 1A*, and *Figure 3—video 1*. are representative data drawn from three experiments for phase contrast data and two experiments for fluorescent data. Biological replicates,

providing information on cell-to-cell clonal variation, for the purposes of this experiment should be considered individual channels, which are seeded by one or a few cells at the start of the experiment.

## MreB alignment within protoplasts

Cells of strains bJS18 (GFP-Mbl) and bEG300 (GFP-MreB) were grown overnight at 25°C in the osmoprotective SMM media (LB supplemented with 20 mM magnesium chloride, 17 mM maleic acid, 500 mM sucrose, brought to a pH of 7.0) with maximum xylose induction (30 mM); cells were shifted to 37°C in the morning. For strain bEG300, the SMM media was supplemented with 8 mM xylose (for intermediate TagO induction). Following 2 hr of growth, 10 mg/mL of freshly suspended lysozyme was added to the cultures with $OD_{600}$ >0.2. After growing for 1–2 hr in lysozyme, the cells were spun and concentrated. 6% agar pads made in LB-SMM were made using a polydimethylsilox-ane (PDMS) mold with crosses (2, 4 and 5 μm arms and 5 μm center). The cells were placed on the agar pad for 2 min, allowing the cells to settle in the crosses. The pad was then placed in a MatTek (Ashland, Massachusetts) dish for imaging. 10–20 biological replicates were imaged per day and the experiment was replicated on 3 days.

## Depletions in liquid culture

TagO depletions in *Figure 2A* were conducted using strain BEG300 in liquid culture. Cells were prepared as overnights, as described above, then grown at the specified xylose concentration at 37°C in LB with 20 mM magnesium chloride for 4 hr. The cells were then imaged as described above in the 'Imaging – MreB particle tracking' section.

Pbp2A depletions shown in *Figure 2* were conducted in liquid culture using strain BRB785 and BRB786 with an IPTG-inducible Pbp2A fusion at the native locus with the redundant transpeptidase PbpH deleted. This strain was grown overnight in the presence of 2 mM IPTG, and then inoculated into CH media containing 2 mM IPTG, 0.015% xylose, and 20 mM magnesium chloride to stabilize the cells against lysis. At an $OD_{600}$ of 0.6, cells were spun down in a tabletop centrifuge and washed three times in CH media lacking IPTG. Cells were placed under agar pads containing 20 mM magnesium chloride, and spinning disk confocal images were taken every 5 s on a Nikon Ti microscope with a 100 × 1.49 TIRF objective and a Hamamatsu ImagEM C9100-13 EM-CCD camera (effective pixel size of 160 nm).

## Depletions under solid state medium

Depletions shown in *Figure 5A* were conducted using strain BEG300. Cells were prepared as overnights in LB with 1 mM magnesium chloride and 12 mM xylose. In the morning, they were washed in LB with 12 mM xylose and no magnesium and placed under a 3% agar pad with the same medium. Phase contrast images were collected every 5 min using a Photometrics (Tucson, Arizona) CoolSNAP HQ2 CCD camera. The experiment was replicated on 2 days.

## Repletions

Repletions of TagO or Pbp2a on pads, as shown in *Figure 5B* and *Figure 5—video 1* (bottom), were performed with strains BEG300 and BRB785 respectively. Cells were grown as overnights, as described above, then depleted at 37°C for >4 hr in LB with 20 mM magnesium chloride and collected by centrifugation at 6000 x *g* for 2 min. The cells were re-suspended in LB supplemented with 20 mM magnesium chloride and 1 mM IPTG (BRB785) and 30 mM xylose (BEG300), placed under 5% agarose pads on coverslips with thickness No. 1.5 for imaging. Phase contrast images were collected every 5 min using a Photometrics CoolSNAP HQ2 CCD camera.

For the repletions shown in *Figures 5B–C* and *6A*, *Figure 5—figure supplement 1*, and *Figure 5—video 1* (top) and 2, performed in the CellASIC microfluidic device in a B04A plate, BCW82 and BEG300 cells were grown to $OD_{600}$1.2–1.5 in LB supplemented with 20 mM magnesium chloride, centrifuged to pellet large clumps for 3 min at <500 x *g*, and the supernatant loaded into the plate. Growth medium was supplied at 5–6 PSI. Cells were grown for at least an additional 30 min before the addition of inducer to the growth medium. Phase contrast images were collected every 10 min. Fluorescent images were collected on the imaging setup described in the 'Imaging – MreB Particle Tracking' section above: GFP-MreB was induced upon loading into the microfluidic chamber

with 1 mM IPTG, and MreB dynamics were observed for 3 min after every 10 min, using 300 ms camera exposures taken every 2 s. The experiment was repeated six times, with the number of biological replicates per experiment limited by chance variation in the loading procedure, but on the order of one to four per experiment.

For the repletions shown in *Figure 6B* and *Figure 5—figure supplement 1D*, the same procedure was used, but with imaging performed on the spinning disk confocal microscope described in 'Imaging – Spinning Disk Confocal'. Z-stacks were collected with a range of 3 µm around the focal plane and 0.2 µm steps. The MreB localization experiments were done using strain bEG300 with full induction of GFP-MreB (1 mM IPTG) and recovering cells were imaged using the spinning disk microscope, collecting Z-stacks as described before.

Where indicated, instead of visualizing MreB dynamics, fluorescent D-amino acids (*Kuru et al., 2012*) (7 µM) were added to the growth medium in the CellASIC device: HADA during depletions of TagO (0 mM xylose) and Cy3B-ADA during repletion of TagO (30 mM xylose). Cells were washed with LB supplemented with 20 mM magnesium chloride containing no D-amino acids for 1–2 min before imaging.

To test if rod shape recovery occurs in the absence of cell division, three strains were tested (BAB327, BAB343 and BAB388). Cells of BAB327 and BAB388 were grown in CH media with 25 mM magnesium chloride in the absence of xylose at 37°C until $OD_{600}$ ~0.5 and diluted 10-fold in fresh media. After 2 hr of growth, IPTG was added to a final concentration of 1 mM (MinCD and FtsA, respectively) and cells were incubated for an extra 1 hr. Cells were imaged on a spinning disk confocal under pads with 1 mM IPTG and 60 mM xylose (for TagO repletion). Phase-contrast and fluorescent images were acquired at 10 min intervals for a total of 8 hr. Cells of BAB343 were grown in LB supplemented with 20 mM, magnesium chloride in the absence of xylose at 25°C overnight. The next day, after 2 hr of growth in the same media at 37°C, IPTG was added to a final concentration of 1 mM (MciZ) and cells were incubated for an extra 1 hr. Cells were imaged on a spinning disk confocal under pads with 1 mM IPTG and 30 mM xylose (for TagO repletion). Phase-contrast and fluorescent images were acquired at 10 min intervals for a total of 4 hr.

## Depletion and repletion of magnesium in the CellASIC

For *Figure 5—video 2*, cells of BCW51 were grown overnight at 25°C in LB supplemented with 8 mM xylose, 20 mM magnesium chloride, 1 µg/ml erythromycin and 25 µg/ml lincomycin (MLS). Cells were shifted to 37°C for 2 hr and loaded into the CellASIC B04A plate at $OD_{600}$ ~ 0.6. At the start of imaging, magnesium was depleted by flowing in LB supplemented only with 8 mM xylose and MLS at 3 psi. Images were collected every 20 min over a 4 hr period. Magnesium was resupplied to the cells by changing to LB supplemented with 8 mM xylose, 20 mM magnesium chloride, and MLS. Imaging was continued every 20 min for an additional 4 hr. 5–10 cells were imaged and the experiment was done once.

## Measurements of cell shape at steady state growth

Cells were grown overnight at 25°C in LB supplemented with 30 mM xylose, 20 mM magnesium chloride, 1 µg/mL erythromycin and 25 µg/mL lincomycin. In the morning they were collected at $OD_{600}$ ~ 0.2, spun in a tabletop centrifuge at 9000 rpm for 3 min and washed in LB supplemented with various xylose (0–30 mM) and magnesium chloride (0–20 mM) levels. 25-fold serial dilutions into LB supplemented with the same xylose and magnesium chloride concentrations were made and allowed to grow at 37°C for 4 hr. Cells at $OD_{600}$ ~ 0.2 were concentrated by spinning in a tabletop centrifuge at 9000 rpm for 3 min. They were placed on a coverslip thickness No. 1.5 under 3% agarose pads made in LB supplemented with the same concentrations of xylose and magnesium chloride. Images were collected using the imaging setup described in the 'Imaging – phase contrast microscopy' section above, as well as with a Photometrics CoolSNAP HQ2 CCD camera. The magnification and pixel size were the same in both setups. The experiment was replicated on two days and the data pooled together.

## Particle tracking

The MATLAB based software uTrack was used for particle tracking (*Jaqaman et al., 2008*). We used the comet detection algorithm to detect filaments (difference of Gaussian: one pixel low-pass to 4–6

pixels high pass, watershed segmentation parameters: minimum threshold 3–5 standard deviations with a step size of 1 pixel) which, at our MreB induction levels gave better localization of the resultant asymmetric particles over algorithms that search for symmetric Gaussians. Visual inspection of detected particles confirmed that most of the particles and none of the noise were being detected. A minimum Brownian search radius of 0.1–0.2 pixels and a maximum of 1–2 pixels was applied to link particles with at least five successive frames. Directed motion propagation was applied, with no joins between gaps allowed. Tracks were visualized using the FIJI plug-in TrackMate (*Tinevez et al., 2017*). For sphere to rod transitions and cells confined in microfluidic channels, movies were processed by subtracting every 8th frame from each frame to remove stationary spots using the FIJI plugin StackDifference before tracking. The tracking was done as described earlier in this section.

## Microscopic analysis of GFP-TagTUV

Strains containing fluorescent fusions to TagT, TagU, and TagV were grown as described in the 'Overnight culture growth' section but in CH medium instead of LB. Cells were grown for 3 hr at 37°C before imaging, then collected by centrifugation at 6000 x *g* for 2 min and re-suspended in CH. Cells were then placed on a glass coverslip thickness No. 1.5 under an agar pad thickness 1 mm made from CH and 1.5% agarose. Timelapse images were collected with TIRF illumination, using continuous 100 ms 488 nm exposures. Epifluorescent illuminated images were collected from a single exposure, while maximal intensity projections were formed from a series of continuous 100 ms TIRF exposures. Technical replicates were not collected, but each strain was imaged under a variety of appropriate imaging conditions to establish that the phenomena observed were not an artifact of the experimental setup.

## Single-Molecule imaging with HaloTag-JF549 ligand

For single-molecule experiments imaging HaloTag-15aa-TagU (bAB196), HaloTag-15aa-TagV (bAB197), HaloTag-15aa-TagT (bAB198), Pbp2A-30aa-HaloTag (BSY201) and MreB-30aa-HaloTag (BYS40), cells were grown at 37°C in CH medium from fresh colonies until reaching mid-exponential phase ($OD_{600}$ = 0.5), ten-fold diluted back in fresh CH medium and grown one more round until $OD_{600}$ = 0.5. Cultures were then incubated for 15 min with 25 nM (HaloTag-15aa-TagU/V/T and Pbp2A-30aa-HaloTag) and 25 pM (MreB-30aa-HaloTag) of HaloTag-JF549 ligand. Cells were spun for 1 min at 4000 xg, 10-fold concentrated and imaged under 2% agarose pads. Images were collected on a Nikon TI microscope equipped with an EMCCD camera, together with a Nikon 100x NA 1.45 objective. Exposure times were 0.25 s, and illumination was accomplished using a 561 nm laser.

## Teichoic acid labeling with concanavalin A

BEG300 cells were grown from overnights as described in 'Overnight culture growth' but in CH medium instead of LB. Cells were then grown at 37°C for 4 hr without xylose to deplete WTAs, then induced with 30 mM xylose for 1.5 hr to re-induce WTA expression. Cells were then moved to 25°C for at least 30 min and incubated with 25 µg/mL Concanavalin A conjugated to Alexa Fluor 647. Cells were collected by centrifugation at 6000 x *g* for 2 min, washed with CH medium, then re-suspended in fresh CH medium. Cells were then placed on a glass coverslip thickness No. 1.5 under an agar pad thickness 1 mm made from CH medium and 1.5% agarose. For PY79 and BCW61 controls, lectin-Alexa Fluor conjugate concentration was 200 µg/mL. Separate technical replicates were collected with the microscopes described in 'Imaging – MreB Particle Tracking' and 'Imaging – Spinning Disk Confocal'. For quantitative analysis, the latter setup was used. Quantification was performed in FIJI. Five individual cells were selected arbitrarily and pooled for quantitative analysis. The contours of each cell were manually traced, and intensity along these contours measured, then corrected for the mean fluorescent background. The per cell average was calculated from the mean of each pixel in the contour less background (calculated from an empty field). The per strain average was taken as the mean of the per cell average.

## Data analysis – selecting directional tracks

The output of uTrack is the position coordinates of tracks over frames. We fit a line through these coordinates using orthogonal least squares regression to minimize the perpendicular distance of the points from the line of best fit. We used principal component analysis for orthogonal regression

using custom written MATLAB code. The R$^2$ values we obtain range from 0.5 to 1. We calculated mean track positions, angles and displacement using the line of best fit for all tracks. We also calculated the mean square displacement versus time of individual tracks and fit these curves to the quadratic equation $MSD(t) = 4Dt + (Vt)^2$, using nonlinear least squares fitting. As later times have fewer points and are noisier, we fit the first 80% of the data for each track. We determined $\alpha$ by fitting a straight line to the $\log(MSD(t))$ vs. $\log(t)$ curve. The goodness of fit was evaluated by determining the R$^2$ value. We selected tracks for linearity and directional motion, based on the following cutoffs: R$^2$ >0.9, displacement >0.2 μm,, velocity >1e$^{-9}$ μm/s, and R$^2$ of the linear fit of log(MSD(t)) vs. log(t) >0.6.

## Data analysis – cell segmentation

The MATLAB-based software Morphometrics (*Ursell et al., 2017*) was used to segment phase contrast images of cells. We used the phase contrast setting for rod-shaped and intermediate states and the peripheral fluorescence setting for spherical states, because in this latter condition, peripheral fluorescence empirically did a better job of fitting cell outlines. The cell contours obtained were visually inspected and any erroneous contours were removed by custom written MATLAB code.

## Data analysis – track angles with respect to the long axis of the cell

Track angles were calculated with respect to the cell midline as defined by the Morphometrics 'Calculate Pill Mesh' feature, which identifies the midline based on a unique discretization of the cell shape determined from its Voronoi diagram. The difference between the track angle and midline angle was then calculated. Since the track angles $\theta_t$ and midline angles $\theta_m$ both ranged from −90° to 90°, the range of angle differences $\theta = \theta_t - \theta_m$ was −180° to 180°. We changed the range to 0 to 180° by the transformation: $\Delta\theta = 180 + \Delta\theta$ *if* $\Delta\theta$<0, and 0 to 90° by the transformation: $\Delta\theta = 180 - \Delta\theta$ *if* $\Delta\theta$>90. The mean deviation from 90° ($\sigma_{90}$) for each distribution was calculated using the following formula, where x$_i$ is each angle in the distribution and N is the total number of angles:

$$\sigma_{90} = \sqrt{\frac{\sum_{i=1}^{N}(x_i - 90)^2}{N}}$$

## Data analysis – mean dot product of tracks

Custom written MATLAB code was used to calculate the normalized dot product (*DP*) of track pairs along with the distance (*d*) between their mean positions $\bar{x}$ and $\bar{y}$ as follows:

$$DP_{ij} = \cos(\theta_i - \theta_j), \, d_{ij} = \sqrt{(\bar{x}_i - \bar{x}_j)^2 + (\bar{y}_i - \bar{y}_j)^2}$$

To eliminate out-of-cell tracks we only considered those that had three other tracks within a 5 μm radius of their mean position. The dot product of track pairs (DP) and distance (d) between them was stored in data files, along with all the previous information for each individual track (R$^2$, velocities, angles, mean positions, displacement etc). The files were then parsed using the cutoffs described in the 'Data analysis – selecting directional tracks' section. The tracks were binned based on the distance and the mean dot product calculated for each distance range as follows:

$$\bar{DP} = \frac{1}{N}\sum_{i\>j}^{N}\cos(\theta_i - \theta_j)$$

A cutoff of 3 μm was chosen as the maximum binning distance, which is the average length of a cell.

## Data analysis – simulation of random angles

A data file containing simulated tracks was created by a custom written MATLAB script, which generates random angles distributed randomly on a 100 × 100 μm area. Each track has R$^2$ = 0.95, velocity = 25 nm/s and displacement = 1 μm. The same analysis code was run on these simulated tracks to generate track pairs with dot product and distance stored in a new data file. The data file was parsed using the same cutoffs as the real data and the mean dot product for each distance range

calculated. The total numbers of trajectories within the simulation were much higher than the actual data (2–10 times higher).

## Data analysis – cell width

Pill meshes were created using Morphometrics (*Ursell et al., 2017*), which calculates the coordinates of line segments perpendicular to the cell long axis. For cell widths at various steady state TagO and $Mg^{2+}$ levels, the distance of these line segments was calculated using a custom written MATLAB script (*Hussain, 2017*; copy archived at https://github.com/elifesciences-publications/hussain-2017-elife) and the maximum width along the length of the cell was taken as the cell width. When measuring cell width nearest to a track (for calculating track angle as a function of cell width), the mean width of the 10 nearest contour points from the track was calculated using a custom written MATLAB script. Cell widths of emerging bulges and rods from round cells were measured manually in FIJI. Our ability to segment individual spherical cells was limited by their nonuniform contrast, perhaps arising from the nonuniform thickness of these cells in the Z dimension; consequently, Morphometrics-based width measurements in these cells was limited, especially in cells exceeding 2 µm in diameter.

## Data analysis – cell curvature

Sidewall curvature of cells was extracted from the pill mesh obtained from Morphometrics. The curvature values are calculated from three successive contour points and smoothed over two pixels. The mean curvature of 3 nearest points to each track were calculated from both sides of the cell contour and called the mean curvature. Principal curvature ratio was calculated by dividing the sidewall curvature with the curvature in the radial direction (calculated from cell width assuming the cell is radially symmetric). For radial curvature we used the following expression, where $r_{cell}$ is half the cell width:

$$\kappa_2 = \frac{1}{r_{cell}}$$

A value close to one indicates the two principal curvatures are similar and the cells are round.

## Data analysis – time and curvature plots of rod shape recovery

Phase contrast images were used to show rod shape emergence from local bulges. Edges were enhanced in FIJI and contrast adjusted to give bright cell outlines in the images. The stack was then colored in time using temporal color code function in FIJI. To create the curvature plot, the phase contrast images were run through Morphometrics which calculates the curvature at each contour point along the cell outline. The contour points of interest were selected and plotted using a custom written MATLAB script, which colored each point according to its local curvature as calculated by Morphometrics. To provide a good resolution for positive curvatures, we rescaled the color map such that negative curvatures were colored blue and positive curvatures were scaled by their curvature value.

## Data analysis – single cell doubling times

Data from agar pads experiments was analyzed using custom written MATLAB code. Data from cellASIC experiments was analyzed in Morphometrics to get areas for each cell. For doubling times during sphere to rod transitions, the data was collected by manually measuring the areas of the sphere and rod regions of the same cell in FIJI. In all cases, the area of each cell per frame was calculated and the log plot of area vs time was fit to a line. The doubling time was calculated using the slope of this line. N = 15 cells (transitioning from spheres to rods).

## Data analysis – Tangential correlation of cell contours

Cell contours (*Figure 1—figure supplement 1*) were used to calculate tangent angles using the equation: $\theta_i = \tan^{-1} \frac{y_{i+1} - y_i}{x_{i+1} - x_i}$. The correlation between angles was calculated using the cosine of the angle difference binned as a function of number of points (n) between the

angles: $G(n) = \frac{1}{N}\sum_{i=1}^{N}\cos(\theta_{i+n} - \theta_i)$. The number of points was converted to contour length using the

pixel size of the camera to get the final correlation function: $G(l) = \frac{1}{N}\sum_{i=1}^{N}\cos(\theta_{i+l} - \theta_i)$.

## Data analysis – Analysis of HaloTag and MreB trajectories

Tracks were generated using the TrackMate plugin in Fiji (2017). Particles were detected with the Laplacian of Gaussians (LoG) detector, with a 0.4 µm spot diameter. Tracks were generated using the Simple LAP Tracker, with a 0.1 µm linking max distance and no frame gaps allowed. Tracks were exported into MATLAB for further processing. Mean squared displacement (MSD) was calculated for all tracks as a function of time delay (t).

For visualization of HaloTag tracks overlaid on phase images, we analyzed all tracks 10 frames or longer. For each track, the scaling exponent (α) was calculated by fitting MSD(t)=C(t$^\wedge$ α)+4(σ$^\wedge$2) using nonlinear least-squares fitting with constant C and where σ is the localization error (*Monnier et al., 2012*).

For analysis of alpha value and velocity frequencies tracks were first filtered using masks generated from phase images acquired after single-molecule imaging. We then included all tracks between 10 and 120 frames in length. **α** values were calculated by linear fitting log(MSD) versus log (t). Velocity (v) was calculated by fitting MSD(t)=4D(t) + (v*t)$^\wedge$2 + 4(σ$^\wedge$2) using nonlinear least-squares fitting, where D is the diffusion constant (*Monnier et al., 2012*). For the graphs in *Figure 6—figure supplement 2*, only plotted are molecules that moved in a consistent manner during their lifetime [>0.95 r2 fit to log(MSD) versus log(t)]. Analysis of alpha value and velocity frequencies for MreB filaments was the same as for HaloTag, except we included all tracks between 10 and 60 frames in length and filtration by phase image masks was not necessary.

## *T. maritima* MreB protein purification

Full length, un-tagged *Thermotoga maritima* MreB was purified as described previously (*Salje et al., 2011*).

## In vitro reconstitution of *T. maritima* MreB filaments inside liposomes

The protein was encapsulated inside unilamellar liposomes following a previously published protocol (*Szwedziak et al., 2014*). For this, 50 µL of *E. coli* total lipid extract, dissolved in chloroform at 10 mg/mL, was dried in a glass vial under a stream of nitrogen gas and left overnight under vacuum to remove traces of the solvent. The resulting thin lipid film was hydrated with 50 µL of TEN100 8.0 (50 mM Tris/HCl, 100 mM NaCl, 1 mM EDTA, 1 mM NaN$_3$, pH 8.0), supplemented with 20 mM CHAPS (Anatrace, Maumee, Ohio), and shaken vigorously at 800 rpm using a benchtop micro centrifuge tube shaker for 2 hr. The lipid-detergent solution was then sonicated for 1 min in a water bath sonicator. Subsequently, 50 µL of MreB protein solution at 30 µM, supplemented with 0.5 mM magnesium ATP (Jena Bioscience, Germany) was added and left for 30 min at room temperature. Next, the mixture was gradually diluted within 10–20 min to 600 µL with TEN100 8.0 plus 0.5 mM magnesium ATP (without detergent) to trigger spontaneous liposome formation. 2.5 µL of the solution was mixed with 0.2 µL 10 nm IgG immunogold conjugate (TAAB, UK) and plunge-frozen onto Quantifoil R2/2 carbon grid, using a Vitrobot automated freeze plunger (FEI Company, Hillsboro, Oregon) into liquid ethane.

## Electron cryomicroscopy and cryotomography

2D electron cryomicroscopy images were taken on an FEI Polara TEM (FEI Company) operating at 300 kV with a 4k × 4 k Falcon II direct electron detector (FEI Company) at a pixel size of 1.8 Å. For electron cryotomography, samples were imaged using an FEI Titan Krios TEM (FEI Company) operating at 300 kV, equipped with a Gatan imaging filter set at zero-loss peak with a slit-width of 20 eV. A 4k × 4 k post-GIF K2 Summit direct electron detector (Gatan, a subsidiary of Roper Technologies, Lakewood Ranch, Florida) was used for data acquisition with SerialEM software (2005) at a pixel size of 3.8 Å at the specimen level. Specimens were tilted from −60° to +60° with uniform 1° increments. The defocus was set to between 8 and 10 µm, and the total dose for each tilt series was around 120–150 e/Å$^2$.

## Image processing

Tomographic reconstructions from tilt series were calculated using RAPTOR (2008) and the IMOD tomography reconstruction package followed by SIRT reconstruction with the TOMO3D package (1996; 2011a). Movies showing liposomes were prepared with Chimera and PyMOL (2002; 2004).

## Imaging of *E. coli* strain RM478

Imaging of *E. coli* strain RM478 was conducted as in 'Overnight culture growth', save the cells were grown in LB or M63-Glucose. When indicated, cells were grown and imaged under 2% agar pads made in the same media. For the media down shift (LB at 37°C to M63-Glucose at 25°C), cells were first grown in LB at 37°C in a drum roller, then placed under agar pads made of M63-Glucose at 25°C and allowed to equilibrate 15 min prior to imaging. Imaging was conducted as in 'Imaging – MreB particle tracking' using a Nikon TI-E with TIRF illumination.

## Strain construction

All strains used in this study are available in *Supplementary file 1*.

All primers used in this study are available in *Supplementary file 2*.

**BCW51** [*ycgO::Pxyl-tagO, tagO::erm, amyE::sfGFP-mreB, sinR::phleo*] was generated by transforming BEG300 with a Gibson assembly consisting of three fragments: (1) PCR with primers Sinr_up_F and Sinr_up_R and template PY79 genomic DNA; (2) PCR with primers oJM028 and oJM029 and template plasmid pWX478a (containing *phleo*); (3) PCR with primers Sinr_DOWN_R and Sinr_DOWN_F and template genomic DNA.

**BCW61** [*tagE::erm*] was generated by transforming PY79 with a Gibson assembly consisting of three fragments: (1) PCR with primers oCW054 and oCW055 and template PY79 genomic DNA; (2) PCR with primers oJM028 and oCW057 and template plasmid pWX467a containing *cat*; (3) PCR with primers oCW058 and oCW059 and template PY79 genomic DNA.

**BCW72** [*yvhJ::PxylA-mazF (cat)*] was generated by transforming PY79 with a Gibson assembly consisting of three fragments: (1) PCR with primers oCW139 and oCW141 and template PY79 genomic DNA; (2) PCR with primers oJM029 and oMK047 and template DNA consisting of a fusion of *cat* and the *mazF* counterselectable marker from pGDREF (*Yu et al., 2010*); (3) PCR with primers oCW142 and oCW143 and template PY79 genomic DNA.

**BCW77** [*ywtF::PxylA-mazF (cat)*] was generated by transforming PY79 with a Gibson assembly consisting of three fragments: (1) PCR with primers oCW159 and oCW161 and template PY79 genomic DNA; (2) PCR with primers oJM029 and oMK047 and template DNA consisting of a fusion of *cat* and the *mazF* counterselectable marker from pGDREF (*Yu et al., 2010*); (3) PCR with primers oCW164 and oCW165 and template PY79 genomic DNA.

**BCW78** [*ywtF::msfGFP-ywtF*] was generated by transforming BCW77 with a Gibson assembly consisting of three fragments: (1) PCR with primers oCW160 and oCW161 and template PY79 genomic DNA; (2) PCR with primers oCW072 and oCW073 and BMD61 genomic DNA; (3) PCR with primers oCW163 and oCW165 and template PY79 genomic DNA.

**BCW79** [*yvhJ::msfGFP-yvhJ*] was generated by transforming BCW72 with a Gibson assembly consisting of three fragments: (1) PCR with primers oCW139 and oCW146 and template PY79 genomic DNA; (2) PCR with primers oCW072 and oCW073 and BMD61 genomic DNA; (3) PCR with primers oCW143 and oCW145 and template PY79 genomic DNA.

**BCW80** [*lytR::PxylA-mazF (cat)*] was generated by transforming PY79 with a Gibson assembly consisting of three fragments: (1) PCR with primers oCW101 and oCW109 and template PY79 genomic DNA; (2) PCR with primers oJM029 and oMK047 and template DNA consisting of a fusion of *cat* and the *mazF* counterselectable marker from pGDREF (*Yu et al., 2010*); (3) PCR with primers oCW100 and oCW125 and template PY79 genomic DNA.

**BCW81** [*lytR::msfGFP-lytR*] was generated by transforming BCW72 with a Gibson assembly consisting of three fragments: (1) PCR with primers oCW101 and oCW137 and template PY79 genomic DNA; (2) PCR with primers oCW072 and oCW073 and BMD61 genomic DNA; (3) PCR with primers oCW100 and oCW138 and template PY79 genomic DNA.

**BCW82** [*tagO::erm, ycgO::PxylA-tagO, amyE::Pspac-gfp-mreB (spec), dacA::kan*] was generated by transforming BEG300 with genomic DNA from BGL19.

**BEG202** [*tagO::erm amyE::Pxyl-gfp-mbl* (*spec*)] was generated by transforming BEB1451 with genomic DNA from BJS18.

**BEG281** [*ycgO::PxylA-tagO*] was generated by transforming with a plasmid created via ligating a Gibson assembly into pKM077. pKM77 was digested with EcoRI and XhoI. The assembly was created with two fragments: (1) PCR with primers oEG85 and oEG86 and template py79 genomic DNA; (2) PCR with primers oEG87 and oEG88.

**BEG291** [*tagO::erm, ycgO::PxylA-tagO*] was generated by transforming BEG281 with genomic DNA from BRB4282.

**BEG300** [*tagO::erm, ycgO::PxylA-tagO, amyE::Pspac-gfp-mreB* (*spec*)] was generated by transforming BEG291 with genomic DNA from BEG275.

**BMD61** [*mbl::mbl-msfGFP* (*spec*)] was generated by transforming py79 with a Gibson assembly consisting of four fragments: (1) PCR with primers oMD44 and oMD90 and template PY79 genomic DNA; (2) PCR with primers oMD47 and oMD56 and template synthetic, codon-optimized *msfGFP*; (3) PCR with primers oJM028 and oJM029 and template plasmid pWX466a (containing *spec*); (4) PCR with primers oMD48 and oMD50 and template genomic DNA.

**bSW99** [*amyE::spc-Pspac-mciZ*] was generated by transforming PY79 with a Gibson assembly consisting of five fragments: (1) PCR with primers oMD191 and oMD108 and template PY79 genomic DNA (containing upstream region of amyE); (2) PCR with primers oJM29 and oJM28 and template plasmid pWX466a (containing *spec*); (3) PCR with primers oMD234 and oSW76 and template plasmid pBOSE1400 (a gift from Dr. Briana Burton, containing *spec*); (4) PCR with primers oAB307 and oAB291 and template PY79 genomic DNA (containing mciZ); (5) PCR with primers oMD196 and oMD197 and template PY79 genomic DNA (containing downstream region of amyE).

**bAB343** [*tagO::erm, ycgO::cat-PxylA-tagO, amyE::spc-Pspac-mciZ, ftsAZ::ftsA-mNeonGreen-ftsZ*] was generated by transforming bAB185 (*Bisson-Filho et al., 2017*) with genomic DNA from bSW99. *The resultant strain was then transformed with the genomic DNA from BEG291 and selected for Cm resistance. Subsequently, the resultant strain was transformed again with genomic DNA from BEG291, but colonies were selected for MLS resistance in the presence of 30 mM of xylose and 25 mM MgCl2.*

**bAB327** [*tagO::erm, ycgO::cat-PxylA-tagO, amyE::spc-Physpank-minCD, ftsAZ::ftsA-mNeon-Green-ftsZ*] was generated by transforming bAB185 (*Bisson-Filho et al., 2017*) with genomic DNA from JB60 (a gift from Dr. Frederico Gueiros-Filho). *The resultant strain was then transformed with the genomic DNA from BEG291 and selected for Cm resistance. Subsequently, the resultant strain was transformed again with genomic DNA from BEG291, but colonies were selected for MLS resistance in the presence of 60 mM xylose and 25 mM MgCl2.*

**bAB388** [*tagO::erm, ycgO::cat-PxylA-tagO, amyE::spc-Physpank-ftsA, ftsAZ::ftsA-mNeonGreen-ftsZ*] was generated by transforming bAB199 (*Bisson-Filho et al., 2017*) with genomic DNA from *BEG291 and selected for Cm resistance. Subsequently, the resultant strain was transformed again with genomic DNA from BEG291, but colonies were selected for MLS resistance in the presence of 60 mM xylose and 25 mM MgCl2.*

**bAB196** [*tagU::erm-Pxyl-HaloTag-15aa-tagU*] was generated by transforming PY79 with a Gibson assembly consisting of three fragments: (1) PCR amplifying the upstream region from *tagU* with primers oCW155 and oCW109 and template PY79 genomic DNA; (2) PCR amplifying the erm-Pxyl-HaloTag-15aa fragment with primers oJM029 and oCW73 and template bGS62 genomic; (3) PCR amplifying the downstream region from *tagU* with primers oCW138 and oCW156 and PY79 template genomic DNA.

**bAB197** [*tagV::erm-Pxyl-HaloTag-15aa-tagV*] was generated by transforming PY79 with a Gibson assembly consisting of three fragments: (1) PCR amplifying the upstream region from *tagV* with primers oCW140 and oCW141 and template PY79 genomic DNA; (2) PCR amplifying the erm-Pxyl-HaloTag-15aa fragment with primers oJM029 and oCW73 and template bGS62 genomic; (3) PCR amplifying the downstream region from *tagV* with primers oCW144 and oCW145 and PY79 template genomic DNA.

**bAB198** [*tagT::erm-Pxyl-HaloTag-15aa-tagT*] was generated by transforming PY79 with a Gibson assembly consisting of three fragments: (1) PCR amplifying the upstream region from *tagT* with primers oCW162 and oCW159 and template PY79 genomic DNA; (2) PCR amplifying the erm-Pxyl-HaloTag-15aa fragment with primers oJM029 and oCW73 and template bGS62 genomic; (3) PCR

amplifying the downstream region from *tagT* with primers oCW163 and oCW166 and PY79 template genomic DNA.

**bYS09** [*mreB::mreB-mNeonGreen*] was generated by transforming BMD88 (*Schirner et al., 2015*) with a Gibson assembly consisting of three fragments: (1) PCR with primers oMD134 and oMD262 and bMD135 template genomic DNA (containing the upstream of mreB and 40aa linker); (2) PCR with primers oYS007 and oYS008 and gBlocks gene fragment containing mNeonGreen; (3) PCR with primers oMD92 and oMD116 and template PY79 genomic DNA.

**bYS40** [*mreB::mreB-HaloTag*] was generated by transforming BMD88 with a Gibson assembly consisting of three fragments: (1) PCR with primers oMD134 and oYS602 and bMD135 template genomic DNA; (2) PCR with primers oYS603 and oYS604 and template plasmid cdr1086(containing HALO); (3) PCR with primers oMD92 and oMD116 and template PY79 genomic DNA.

**bYS201** [*HaloTag-Pbp2A::cat*] was generated by transforming PY79 with a Gibson assembly consisting of three fragments: (1) PCR with primers oMD083 and oYS136 and bMD98 sfGFP-Pbp2A::cat template genomic DNA (2) PCR with primers oYS599 and oYS598 and template plasmid cdr1086 (containing HALO); (3) PCR with primers oMD069 and oMD082 and template PY79 genomic DNA.

## Acknowledgements

We thank T Norman, N Lord, and M Cabeen for help with PDMS molds and microfluidics; M Dion and M Kapoor for strains; L Lavis for JF dyes; E Kuru, Y Brun, and M VanNieuwenhze for FDAAs; G Billings, S Van Teeffelen , and T Ursell for helpful discussion; Z Gitai for working cooperatively to figure out the discrepancy in MreB motion in E. coli RodZ MreB S14A cells, T Bharat for help with electron microscopy, and S Layer for advice and inspiration. This work was funded by National Institutes of Health Grant DP2AI117923-01, a Smith Family Award, a Searle Scholar Fellowship (to EG), a Sloan Foundation Award (to AA), an NSF GFRP to (FW), the Medical Research Council (U105184326 to JL) and the Wellcome Trust (095514/Z/11/Z to JL). SW was funded by NIH R01 GM076710. AA, LDR and EG acknowledge support from the Volkswagen Foundation. This work was performed in part at the Center for Nanoscale Systems at Harvard University, supported by NSF ECS-0335765. SH is a HHMI International Student Research Fellow.

## Additional information

### Funding

| Funder | Grant reference number | Author |
|---|---|---|
| National Institutes of Health | DP2AI117923-01 | Ethan C Garner |
| Wellcome Trust | 095514/Z/11/Z | Jan Löwe |
| National Science Foundation | GFRP | Felix Wong |
| Medical Research Council | U105184326 | Jan Löwe |
| Howard Hughes Medical Institute | International Student Research Fellow | Saman Hussain |
| National Institutes of Health | R01 GM076710 | Suzanne Walker |
| Searle Scholar Fellowship | | Ethan C Garner |
| Alfred P. Sloan Foundation | | Ariel Amir |
| Smith Family Award | | Ethan C Garner |
| Volkswagen Foundation | | Lars D Renner Ariel Amir Ethan C Garner |

The funders had no role in study design, data collection and interpretation, or the decision to submit the work for publication.

## Author contributions

Saman Hussain, Conceptualization, Data curation, Formal analysis, Supervision, Validation, Investigation, Visualization, Methodology, Writing—original draft, Project administration, Writing—review and editing; Carl N Wivagg, Conceptualization, Data curation, Software, Formal analysis, Validation, Investigation, Visualization, Methodology, Writing—original draft, Writing—review and editing; Piotr Szwedziak, Software, Validation, Investigation; Felix Wong, Formal analysis, Validation, Investigation, Visualization, Writing—original draft, Writing—review and editing; Kaitlin Schaefer, Formal analysis, Validation, Methodology, Writing—review and editing; Thierry Izoré, Alexandre W Bisson-Filho, Validation, Investigation; Lars D Renner, Formal analysis, Methodology; Matthew J Holmes, Data curation, Formal analysis, Writing—review and editing; Yingjie Sun, Software, Validation; Suzanne Walker, Supervision, Methodology; Ariel Amir, Formal analysis, Writing—original draft, Project administration, Writing—review and editing; Jan Löwe, Software, Formal analysis, Funding acquisition, Validation, Investigation, Writing—original draft, Writing—review and editing; Ethan C Garner, Conceptualization, Resources, Data curation, Formal analysis, Supervision, Funding acquisition, Validation, Methodology, Writing—original draft, Project administration, Writing—review and editing

## Author ORCIDs

Piotr Szwedziak (iD) http://orcid.org/0000-0002-5766-0873
Felix Wong (iD) http://orcid.org/0000-0002-2309-8835
Ariel Amir (iD) http://orcid.org/0000-0003-2611-0139
Jan Löwe (iD) http://orcid.org/0000-0002-5218-6615
Ethan C Garner (iD) http://orcid.org/0000-0003-0141-3555

## Decision letter and Author response

Decision letter https://doi.org/10.7554/eLife.32471.051
Author response https://doi.org/10.7554/eLife.32471.052

# Additional files

## Supplementary files

• Supplementary file 1. Strains used in this study.
DOI: https://doi.org/10.7554/eLife.32471.043

• Supplementary file 2. Oligonucleotides used in this study.
DOI: https://doi.org/10.7554/eLife.32471.044

• Supplementary file 3. Model Parameters. Related to *Figure 4* and Appendix 1.
DOI: https://doi.org/10.7554/eLife.32471.045

• Transparent reporting form
DOI: https://doi.org/10.7554/eLife.32471.046

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

## Appendix 1

DOI: https://doi.org/10.7554/eLife.32471.047

### MreB modeling

All parameters for the analytical model are available in *Supplementary file 3*.

### Modeling predicts preferred MreB orientation and a typical cell width for losing binding orientation

Here we show that energetic modeling of an MreB filament directly binding to the inner membrane predicts the existence of both a preferred orientation of binding and a typical cell width for losing binding orientation. MreB monomers assemble into higher-order oligomers and bind directly to the inner membrane. When an MreB filament binds to the inner membrane, the combined MreB-membrane system requires an energy of deformation $E_{def}(l_b)$ for the membrane to deviate from an equilibrium position and gains an energy of interaction $E_{int}(l_b)$ from the hydrophobic binding. Both the deformation and interaction energies are expressed as functions of the bound MreB length, $l_b$. Note that the rigid cell wall imposes a boundary constraint on the cell membrane and that the equilibrium membrane configuration arises from a balance of membrane bending, osmotic pressure, and cell wall confinement. If the MreB filament were to bind, the change in the total energy $E$ of the membrane-MreB system is:

$$\Delta E = E_{def} - E_{int}. \tag{1}$$

The binding configuration that minimizes $\Delta E$ corresponds to the one that is observed physically. We therefore wish to minimize $\Delta E$.

### Estimate of the hydrophobic interaction energy $E_{int}$

We assume that the biochemistry of MreB is conserved in prokaryotes so that, like *C. crescentus* and *E. coli* MreB (*van den Ent et al., 2014*), *B. subtilis* MreB is assembled into antiparallel double protofilaments consisting of many monomeric units. Consider an MreB filament containing $N_{int}$ interaction sites with a membrane, each with some independent and additive interaction energy $E_{int}^0$. Due to the antiparallel arrangement of the protofilaments (*Salje et al., 2011*; *van den Ent et al., 2014*), there are two binding sites per monomeric unit of MreB. We therefore estimate the number of binding sites per MreB binding length $l_b$ as:

$$N_{int} = \frac{2l_b}{l_{\mathrm{MreB}}}, \tag{2}$$

where $l_{\mathrm{MreB}} \approx 5.1$ nm is the length of a monomeric unit. The energy of burying the amino acids relevant to the binding is approximately:

$$E_{int}^0 \approx 6.04 \frac{\mathrm{kcal}}{\mathrm{mol}} = 10\,kT, \tag{3}$$

where $k$ denotes Boltzmann's constant and $T$ denotes the ambient temperature; the energies of burying individual amino acids were derived from water/octanol partitioning. At a room temperature of $T = 25°\mathrm{C}$, the interaction energy per MreB binding length $l_b$ is therefore:

$$\varepsilon_{int} = \frac{2 \times 10kT}{l_{\mathrm{MreB}}} = 1.8 \times 10^{-11}\ \mathrm{J/m} \tag{4}$$

and the hydrophobic interaction energy is: $E_{int}(l_b) = \varepsilon_{int} l_b$.

### Estimate of the membrane deformation energy

The membrane deformation energy $E_{def}(l_b)$ can be decomposed as:

$$E_{def}(l_b) = E_{bend}^{\mathrm{MreB}}(l_b) + E_{int}^{\mathrm{membrane}}(l_b) \tag{5}$$

where $E_{bend}^{\mathrm{MreB}}$ denotes the bending energy of the MreB filament and $E_{int}^{\mathrm{membrane}}$ denotes the indentation energy of the membrane. We wish to find the MreB-membrane configuration that minimizes the sum of these terms. We prescribe the forms of these terms as follows.

## The bending energy of an MreB filament

We model an MreB filament as a cylindrical rod, with circular cross-sections of radius $r_{\mathrm{MreB}}$ and an intrinsic curvature $1/R_{\mathrm{MreB}}$. The elastic energy density per unit length of bending a cylindrical rod of cross-sectional radius $r_{\mathrm{MreB}}$ from a curvature of $1/R_{\mathrm{MreB}}$ to a curvature of $1/R$ is given by:

$$\varepsilon_{bend} = \frac{\pi Y_{MreB} r_{MreB}^4}{8}\left(\frac{1}{R} - \frac{1}{R_{MreB}}\right)^2 = \frac{B}{2}\left(\frac{1}{R} - \frac{1}{R_{MreB}}\right)^2, \tag{6}$$

where $Y_{MreB}$ is the Young's modulus of an MreB filament and $B$ is its flexural rigidity (**Landau, 1970**). Assuming the Young's modulus of actin, we note that $B = 1.65 \times 10^{-25}$ J·m, which is two orders of magnitude smaller than that previously assumed by Wang and Wingreen for an MreB bundle of cross-sectional radius 10 nm (**Wang and Wingreen, 2013**). In particular, we assume that MreB binds to the inner membrane as pairs of protofilaments and does not bundle. For a uniform flattening of the MreB filament corresponding to $R = \infty, \varepsilon_{bend} \leq 8.2 \times 10^{-13}$ J/m, which is less than the MreB-membrane interaction energy $\varepsilon_{int}$ computed above. This suggests that an MreB filament may be susceptible to bending at our energy scale of interest. How much the MreB bends is determined by a trade-off between the polymer bending energy and the indentation energy of the membrane, which we discuss next.

## The membrane Hamiltonian

We model the inner membrane as an isotropic, fluid membrane composed of a phospholipid bilayer, where there is no in-plane shear modulus and the only in-plane deformations are compressions and expansions. The membrane indentation energy can be expressed as the minimum of an energy functional over the indented states of the membrane. This functional is given by the Helfrich Hamiltonian:

$$F[S] = \int_S \left[\frac{k_b}{2}(2H - H_s)^2 + \frac{k_t}{2}K + \gamma\right] dA + p\int_S dV, \tag{7}$$

where $k_b$ is the bending rigidity of the membrane, $k_t$ is the saddle-splay modulus of the membrane, $H_s$ is the spontaneous curvature of the bilayer, $\gamma$ is the membrane surface tension, $p$ is the pressure differential at the membrane interface, and $H$ and $K$ are the mean and Gaussian curvatures of the surface $S$, respectively (**Zhong-can and Helfrich, 1989**; **Safran, 2003**). The bending rigidity $k_b$, which depends on membrane composition, is typically 10 to 20 kT for lipid bilayers (**Phillips and Kondev, 2012**). Assuming that phospholipids are in excess in the bulk and rearrange themselves on the membrane surface to accommodate areal changes (**Safran, 2003**), we take the membrane surface tension $\gamma = 0$. For large deformations of the inner membrane such as those induced by cell wall lysis (**Deng et al., 2011**), the assumption that the phospholipids are in excess in the bulk may fail to hold and result in a nonzero surface tension. A nonzero surface tension would only enhance the energetic preference of the correct binding orientation; hence, taking a finite surface tension would not change our conclusions. The mechanical energy needed to deform the membrane is the difference between the free energies in the deformed $S$ and undeformed $S_0$ states:

$$E_{ind}^{\mathrm{membrane}} = F[S] - F[S_0]. \tag{8}$$

The surface integrals of the Gaussian curvature are topological invariants by the Gauss-Bonnet theorem and therefore cancel in the difference, hence:

$$E_{ind}^{\text{membrane}} = \left[ \frac{k_b}{2} \int_S (2H - H_s)^2 dA + p(\text{vol}[S]) \right] - \left[ \frac{k_b}{2} \int_S (2H_0 - H_s)^2 dA + p(\text{vol}[S_0]) \right]. \tag{9}$$

Here $H$ denotes the mean curvature of the state $S$, and $H_0$ denotes the mean curvature of the state $S_0$. For simplicity, we set the spontaneous curvature $H_s = 0$; the case of nonvanishing spontaneous curvature can be considered in a similar manner. We therefore write:

$$E_{ind}^{\text{membrane}} = \min_S \left[ 2k_b \int_S \left( H^2 - H_0^2 \right) dA + p \int dV \right], \tag{10}$$

where the volume integral is understood to be the difference of the volumes in the deformed and undeformed states and the areal change accompanying the membrane deformation is small, i.e. $dA_0 \approx dA$. We define the membrane bending energy $E_{bend}^{\text{membrane}}[S; S_0]$ for a conformation $S$ to be the former term and the membrane $pV$ energy $E_{pV}^{\text{membrane}}[S; S_0]$ to be the latter term in the right hand side of *Equation 10*.

As $k_b$ is typically 10 to 20 kT, we take $k_b = 10$ kT and $p$ as a parameter of the model. Note that $p$ denotes the pressure difference acting on the membrane. The value of $p$ is important for determining the tradeoff between the membrane indentation energy $E_{ind}^{\text{membrane}}$ and the MreB bending energy $E_{bend}^{\text{MreB}}$ accurately, but we will show that the preferred orientation of MreB binding is robust over a broad range of $p$.

## Mechanical equilibrium of the undeformed membrane

Consider the balance of forces on the inner membrane in the undeformed state. Assuming that the undeformed membrane is a cylinder with radius $r$ and length $L$ and that sufficient phospholipids exist in the bulk so that $\gamma = 0$, the membrane free energy is

$$E(r) = 2k_b \int_S H_0^2 \, dA + p \int dV = \frac{\pi k_b L}{r} - \pi p L r^2, \tag{11}$$

which is monotonically decreasing in $r$. This implies that the membrane radius should be maximal at equilibrium. If $p = 0$, then the membrane should press against the cell wall and squeeze out the periplasmic space due to minimization of the bending energy. A model in which the periplasm and cytoplasm are isosmotic (*Sochacki et al., 2011*) with no mechanical force exerted by the periplasm is therefore inconsistent with the existence of a periplasm. For the periplasm to exist at equilibrium, it must contribute an additional energy term $E_{peri}$ to the total energy, so that the total energy $F = E + E_{peri}$ as a function of $r$ has a stable fixed point at $r_0 = r^* + \delta r^*$. Here we define $r^* = R_{cell} - h_{peri}$, where $R_{cell}$ is the radius of the cell and $h_{peri}$ is the thickness of the periplasm, and $\delta r^*$ as the initial deformed height where $F'(r) = 0|_{r=r_0}$. We consider expansions of $E_{peri}$ and $F(r)$ around $r_0$. As a function of the deviation in membrane height $\delta r = r - r_0$, we take $E_{peri} \approx \kappa L (\delta r)^2$ and

$$F(\delta r) \approx \kappa^* L (\delta r)^2, \tag{12}$$

where $\kappa^* = \kappa - p\pi$ is the effective membrane pinning modulus, which has been examined before in Wang and Wingreen's work (*Wang and Wingreen, 2013*). For the stability of the fixed point at $r_0$, the condition that the second derivative $F''(\delta r)$ is positive at $\delta r = 0$ implies that $\kappa^* \geq 0$, or $\kappa \geq p\pi$. However, the validity of the expansion $E_{peri}(\delta r) = \kappa L (\delta r)^2$ may be questionable when the deformed height due to polymer binding is larger than or comparable to $\delta r^* \sim pr/\kappa$. Hence, the pinning model may be invalid when $p$ is vanishingly small. For various combinations of $\kappa$ and the polymer bending rigidity where this double-bind is avoided, such as that assumed by Wang and Wingreen's model, the periplasm is effectively a rigid

body. In this case, although a pinning potential can self-consistently penalize deviations in membrane height, it is more intuitive to take the formal limit $\kappa \to \infty$ and treat the periplasm as undeformable. We therefore model the periplasm as a rigid, undeformable body that mechanically supports the cell membrane and imposes a boundary condition on the membrane shape. Any deviation from the equilibrium membrane shape induced by MreB binding is then resisted by the full effect of osmotic pressure. For this reason, in the following analysis we take $p = p_{cell}$, stipulate that the MreB cannot indent the inner membrane outwards, and do not consider the energetic contribution of $E_{peri}$.

## Configuration with a uniformly bent MreB filament: first-order approximation

With the membrane Hamiltonian as defined in the section above, we now see that the total membrane deformation energy is given by the sum of the MreB bending energy and the membrane indentation energy:

$$E_{def}(l_b) = E_{bend}^{\mathrm{MreB}}(l_b) + E_{ind}^{\mathrm{membrane}}(l_b) = \min_{R,S}$$
$$\left[ \frac{B}{2} \int \left( \frac{1}{R} - \frac{1}{R_{MreB}} \right)^2 ds + 2k_b \int_S \left( H^2 - H_0^2 \right) dA + p \int dV \right]. \tag{13}$$

The minimization of **Equation (13)** over all surfaces $S$ and MreB curvatures $R = R(s)$ is generally difficult since minimization of the MreB bending energy determines the preferred conformation of MreB, which in turn restricts the set of surfaces $S$ that **Equation (13)** must be minimized over. In their work, Wang and Wingreen undertook an elegant approach to minimizing a similar combination of energies by writing the membrane indentation energy in Fourier space. Unlike a membrane pinning term, the pressure-volume energy in **Equation (13)** does not admit a simple Fourier space representation. Nevertheless, considerable insight can be obtained by assuming that MreB bends uniformly. In this case, MreB deforms from a bent cylinder with a native curvature $\frac{1}{R_{MreB}}$, to a bent cylinder with a constant, membrane-bound curvature $\frac{1}{R}$. In the following, we will take the radius $R$ of the bend to be a parameter in estimating the corresponding membrane indentation energy $E_{ind}^{\mathrm{membrane}}(l_b)$; $R$ will be determined later. We will also assume that MreB binds perpendicular to the cell's long axis, so that the curvature of the membrane in its undeformed state is simply $\frac{1}{R_{cell}}$, and determine the corrections due to a deviatory binding angle later.

To estimate $E_{ind}^{\mathrm{membrane}}$, we first examine the energetic contribution of the region $C$ of $S$ directly involved in the MreB-membrane interaction. Note that the biochemical conformation of MreB, particularly, the antiparallel orientation of its protofilaments, constrains the geometry of the MreB-membrane binding interface. Since we describe the interface $C$ as the surface of a bent cylinder with principal radii of curvature $\frac{1}{r_{MreB}}$ and $\frac{1}{R}$, we have:

$$H \approx \frac{1}{2}\left( \frac{1}{r_{MreB}} + \frac{1}{R} \right), \quad A = 2\pi b r_{MreB} l_b \tag{14}$$

where $r_{MrB}$ is the cross-sectional radius of MreB and $b$ is the fraction of interaction along a cross-section of the MreB filament. Thus:

$$E_{bend}^{\mathrm{membrane}}[C] \approx \pi b k_b r_{MreB} l_b \left( \frac{1}{r_{MreB}} + \frac{1}{R} \right)^2 - \pi b k_b r_{MreB} l_b \left( \frac{1}{R_{cell}} \right)^2, \tag{15}$$

where $H_0 = 1/2R_{cell}$ and $R_{cell}$ denoting the cell radius, is the mean curvature of the undeformed surface. The contribution of the $pV$ energy over $C$ can be similarly approximated by finding the area between two circles, one being the MreB filament and the other being the cross-section of a cell with radius $R_{cell}$, with $R_{cell} \geq R$ as follows:

$$E_{pV}^{\text{membrane}}[C] \approx 2pr_{MreB} \int_{-R\sin\left(\frac{l_b}{2R}\right)}^{R\sin\left(\frac{l_b}{2R}\right)} \left( \sqrt{R_{cell}^2 - x^2} - \sqrt{R^2 - x^2} + R - R_{cell} \right) dx \approx \frac{pr_{MreB}l_b^3}{12} \left( \frac{1}{R} - \frac{1}{R_{cell}} \right), \quad (16)$$

where we have assumed that $l_b \ll R$ and $\frac{1}{R} \geq \frac{1}{R_{cell}}$. This means that the approximation above is only valid for cases where the MreB filament can only bend up to a curvature $\frac{1}{R_{cell}}$. Now, since $\frac{1}{R} \ll \frac{1}{r_{MreB}}$, we deduce that the principal bending energy contribution over $C$ arises from having the inner membrane tightly wrapped around an MreB filament. For $k_b = 10kT$ and $b = 1/6$, $E_{bend}^{\text{membrane}}[C]$ takes on a value of:

$$E_{bend}^{\text{membrane}}[C] \approx \pi b k_b \, l_b \left( \frac{1}{r_{MreB}} \right) \equiv \varepsilon l_b, \quad \varepsilon = 1.0 \times 10^{-11} \,\text{J/m} \quad (17)$$

which is smaller than, but comparable in scale to, the interaction energy $E^{\text{int}}$ computed above. Writing out only the energetics of the binding region $C$ under the uniform bending assumption, we therefore see that:

$$E_{def}(l_b) \approx \min_R \left[ \frac{B}{2} \left( \frac{1}{R} - \frac{1}{R_{MreB}} \right)^2 l_b + \frac{\pi b k_b \, l_b}{r_{MreB}} + \frac{pr_{MreB}l_b^3}{12} \left( \frac{1}{R} - \frac{1}{R_{cell}} \right) + \min_S F[\boldsymbol{H}] \right], \quad (18)$$

where the last term is the energetic contribution of the falloff region $\boldsymbol{H} = S - C$. In the case that $\frac{1}{R} \to 0$, estimates of the values of the first three terms in **Equation (18)**, for the parameter values summarized in **Supplementary file 1**, are $10^{-19}\,\text{J}$, $10^{-18}\,\text{J}$ and $10^{-17}\,\text{J}$, respectively. This means that, as MreB binds to the inner membrane, the resulting deformation will tend to minimize volumetric changes at the cost of inducing membrane curvature and filament bending. The energetic contribution of the falloff region $\boldsymbol{H}$ can only be quantitatively accounted for by explicitly finding the membrane shape, which encompasses a tradeoff between the membrane bending energy and the $pV$ energy: the former term favors a gradual decay of the indentation, while the latter term prefers a steep decay as to minimize volume. Below, we find that it suffices to consider the case where MreB bends to match the cell curvature: $R = R_{cell}$. In this case, the energetic contribution of the falloff region $\boldsymbol{H}$ is vanishingly small compared to that of the binding region, since the membrane can heal in a manner in which its mean curvature is small compared to the mean curvature of the binding region. The energetic contribution of $\boldsymbol{H}$ can therefore be neglected, and we quantify it in future work.

## $\Delta E$ for the pure bending of an MreB filament

By examining the form of the energetics just over the region $C$, we note that the inclusion of a large pressure $p$ increases the energetic preference of an MreB filament binding perpendicular to the cellular long axis.

Consider a case where $p \geq p^*$ for some $p^*$ to be determined, so that it is energetically unfavorable to displace the membrane volume as opposed to bending the MreB filament. In this case, as discussed above, the energetic contribution of the falloff region $\boldsymbol{H}$ can be neglected, and an estimate for the minimal value of such a pressure can be obtained by requiring that:

$$E_{def}(l_b) \approx \min_R \left[ \frac{B}{2} \left( \frac{1}{R} - \frac{1}{R_{MreB}} \right)^2 l_b + \frac{\pi b k_b \, l_b}{r_{MreB}} + \frac{pr_{MreB}l_b^3}{12} \left( \frac{1}{R} - \frac{1}{R_{cell}} \right) \right] \quad (19)$$

as a function of $R$, be minimal at $R_{cell}$. For the numerical values relevant to MreB above and summarized in **Supplementary file 1**, this indicates that:

$$p^* \approx \frac{12\,B}{r_{MreB}l_b^2} \left( \frac{1}{R_{MreB}} - \frac{1}{R_{cell}} \right) \approx 20 \,\text{kPa}, \quad (20)$$

which is 1/100[th] of the osmotic pressure of *B. subtilis*. In this case, assuming $R = R_{cell}$,

$$\Delta E(l_b) \approx \left[ \frac{B}{2} \left( \frac{1}{R} - \frac{1}{R_{MreB}} \right)^2 + \frac{\pi b k_b}{r_{MreB}} - \varepsilon_{int} \right] l_b,$$ (21)

The energetic dependence on $R_{cell}$ is then manifested through the pure bending of MreB when binding to the inner membrane: in particular, we may assume that the MreB filament will always bend to attain a curvature matching that of the cell's (although small deviations in the membrane height may lead to an even lower energy conformation), and the energetic contribution of the falloff region $H$ can be neglected.

If $p<p^*$, note that both the membrane and the MreB filament can deform each other in a manner that minimizes the total energy, with the membrane shape determined by the geometry of the falloff region $H$. For vesicles with a pressure gradient $p \approx 0$, the fact that an MreB filament grossly deforms the membrane and generates membrane curvature (*Salje et al., 2011*) is predicted by the shape of $H$. For $p \approx 0$, it can also be shown that the energetic difference between MreB binding at $R_{cell} = 500$ nm and $R_{cell} = 3000$ nm is on the order of several $kT$, and so a perpendicular alignment of MreB filaments may also be energetically favorable. For simplicity, in the following discussion we shall consider the wild-type cell scenario, where $p = p_{cell} > p^*$, so that only the pure bending of MreB and the associated membrane bending energy need to be considered in $E$. We will therefore assume the form of *Equation (21)* for $E$ in the discussion that follows.

## Preferred orientation of MreB binding

*Equations (19) and (21)* describe the change in free energy due to MreB binding at a perpendicular angle and bending completely to match the curvature $\frac{1}{R_{cell}}$ of the cell membrane at this angle. Our modeling then predicts that MreB filaments tend to bind at an angle of $\theta = 90°$ relative to the long axis of *B. subtilis*: at any deviatory angle $|\theta - 90°|>0°$, the leading-order correction to the cell radius $R_{cell}$ in the MreB bending energy is a multiplicative factor of $1/\cos\theta$, which monotonically increases the MreB bending energy. Since the MreB bending energy is minimal when the principal axis of curvature of the MreB filament matches that of the cell and the curvature of the cell is maximal at a perpendicular binding angle, the angle distribution is symmetric about a minimum centered at $\theta = 90°$. This reasoning suggests the existence of a potential well centered at $\theta = 90°$, as shown in *Figure 4F* under the simplifying assumption that $b = 0$. The depth of this potential well is on the order of tens of $kT$, which appears to be a large enough energetic preference as to be robust to sources of stochasticity such as thermal fluctuations. Furthermore, our discussion shows that membrane binding energetics, and in particular the pure bending of an MreB filament, may complement the conjecture in Salje et al.'s work that the membrane insertion loop and amphipathic helix help ensure orthogonal membrane binding (*Salje et al., 2011*).

A sensitivity analysis shows that the energetic difference of an MreB filament perpendicularly binding to a region of ambient curvature $R_{cell} = 0.5$ $\mu$m and $R_{cell} = 1.5$ $\mu$m is still on the order of tens of $kT$ over a wide range of binding energy and flexural rigidity values, as shown in *Figure 4G*. Thus, we anticipate the alignment to be robust to changes in these two parameters.

## A typical cell radius for losing orientation

Our modeling shows that the depth of the potential well in $\theta$ is inversely related to $R_{cell}$, so that at larger cell diameters the angle distribution becomes more uniform. Varying the cell radius $R_{cell}$ from 0.5 microns to 3.0 microns results in a reduction of the depth of the potential well, as illustrated in *Figure 4F*. We therefore anticipate MreB filaments to bind with more variance in angle for higher values of $R_{cell}$, consistent with the existence of a typical radius at which the binding angle becomes less robust and affected by factors such as thermal fluctuations or other sources of stochasticity.

## Binding orientation at regions of different Gaussian curvatures

Our modeling predicts that, in live cells, MreB filaments will bend to conform to the shape of the inner membrane. Since the binding sites are located at the outer edge of a curved MreB filament, our modeling also predicts that the binding angle distribution becomes more narrow at regions of negative Gaussian curvature: to bind in a conformation that deviates significantly from the preferred binding orientation, in which the filament's deformed curvature remains of the same sign as its intrinsic curvature, an MreB filament must bend to the extent that its curvature flips sign. Similarly, at regions of positive Gaussian curvature, the binding angle distribution will be less narrow. Representative binding angle distributions are shown in three cases of positive, vanishing, and negative Gaussian curvatures in *Figure 7B* of the main text.

## Appendix 2

DOI: https://doi.org/10.7554/eLife.32471.048

### Localization of WTA ligases and WTAs

In order to understand how wall teichoic acid (WTA) depletion and recovery affects cell shape, we observed the spatial localization of WTAs and the extracellular ligases (the genes *lytR*, *yvhJ*, and *ywtF* [*Kawai et al., 2011*]) that determine their attachment to the cell wall. Previous work has suggested that YwtF and LytR localize in MreB-like patterns (*Kawai et al., 2011*) and associate with MreB (assayed by in vivo crosslinking and tandem affinity purification). We reasoned that if the synthesis or insertion of WTAs is MreB associated, then the emergence of discrete rod-shaped cells upon *tagO* repletion might correlate with preferential WTA insertion at the emerging rod, where MreB shows oriented motion. To test this, we examined (1) the localization and dynamics of fluorescent tagged WTA ligases, and (2) the spatial localization of WTAs in cells recovering from spheres into rods. We constructed sfGFP fusions to each of the WTA ligases under their native promoters, and examined their localization using TIRF and epifluorescent illumination. Although we did observe variation in ligase intensity around the cell periphery under epifluorescent illumination, we did not see any characteristic banding across the cell surface as is seen with MreB using TIRF microscopy (*Figure 6—figure supplement 1C*).

Furthermore, TIRF imaging of these fusions at different frame rates did not show any directional motions, indicating they are not moving along with MreB filaments; rather these motions suggested the WTA ligases were diffusing on the cell membrane (Figure 6—video 2).

To verify this result, we imaged single molecules of HaloTag fusions to these three ligases at their native locus and promoter. We also constructed HaloTag-MreB and HaloTag-Pbp2A strains as controls for directional motion. We sparsely labeled cells with JF549-HaloTag, then tracked their single molecule motions using 250msec exposures, a frame rate at which directional motion is oversampled and easily tracked, but most diffusing particles cannot be tracked for more than a few frames. As shown in *Figure 6—figure supplement 2D*, HaloTag-MreB showed directional motion, as expected. Similarly, we could observe the directional motion of HaloTag-Pbp2A, which has been shown display a mixed population of directional and membrane diffusive molecules (*Garner et al., 2011*; *Cho et al., 2016*). In contrast, we could not observe any directional motions in any of the ligases; rather, the molecules appeared to be diffusing on the membrane, and occasionally, staying in one spot for a few seconds. To quantitatively compare these motions, we calculated the velocity and $\alpha$ for all molecules over 10 frames in length (*Figure 6—figure supplement 2E*). As a stringent cutoff for particles that moved in a consistent manner during their lifetime, we only analyzed particles with an >0.95 $r^2$ fit to log(MSD) versus log(t). Even though most of the ligases molecules were excluded from analysis (95%), as their diffusive motion made it so they could not be tracked more than 10 frames, the $\alpha$ value distributions showed a clear distinction between the remaining trajectories: using an $\alpha > 1.2$ as a cutoff for directional motion, most (54%) of the MreB molecules moved directionally, as did a smaller, but substantial amount of Pbp2A molecules (22%). In contrast, only a small subfraction of ligase molecules longer than 10 frames in length (1.4–4.8%) exceeded $\alpha > 1.2$. To test if this small fraction of apparently directional ligases moved with MreB, we examined their velocity. This revealed that the velocities of this small fraction of ligase molecules is inconsistent with what would be expected if they moved with MreB. Rather, it is more likely this small number of apparently directional trajectories, which occur at a frequency of, at most, less than 1% of the total molecules observed, is likely due to chance, as a small fraction of apparently directional trajectories often arise from the analysis of a large numbers of particles undergoing random walks.

We next explored the localization of WTAs, using fluorescently labeled Concanavalin A (ConA). ConA is a sugar-binding protein with specific affinity for $\alpha$-D-glucose, which decorates WTA polymers. ConA has previously been used to localize WTAs (1975b; 1975a). The gene *tagE* encodes the glycosylase that adds $\alpha$-D-glucose to WTA molecules (2011c), which is

recognized by ConA. ConA staining of cells deleted for *tagE* shows no staining (**Figure 6—figure supplement 1A**), verifying the specificity of ConA for WTAs over other surface sugars. We then used this probe to examine WTA localizations during recoveries. Addition of ConA to WTA-depleted cells in bulk culture shows very little staining at the cell periphery, consistent with a basal level of WTA expression; induction of *tagO* results in a dramatically increased intensity of staining. Even at early time points, when rod-shaped cells are just starting to appear in the population, WTA staining is relatively uniform, with no patterns reminiscent of the patchy distribution of MreB (**Figure 6—figure supplement 1B**). Together, this data indicates that WTA ligases and WTA incorporation occur uniformly around the cell in both wild type cells as well as in TagO depleted spheres recovering into rods. These findings suggest that the changes in activity of TagO that cause the loss of rod shape (or the reformation thereof) occur uniformly around the cell wall. Furthermore, these findings also suggest that the WTA ligases do not consistently localize to MreB within *B. subtilis*.

