## [Decision Letter]

Thank you for submitting your article "MreB Filaments Create Rod Shape by Aligning Along Principal Membrane Curvature" for consideration by *eLife*. Your article has been reviewed by two peer reviewers, and the evaluation has been overseen by a Reviewing Editor and Gisela Storz as the Senior Editor. One of the reviewers, Arnaud Chastanet, has agreed to reveal his identity.

The reviewers have now discussed the reviews with one another and the Reviewing Editor has drafted this decision to help you prepare a revised submission.

The manuscript by Hussain et al. reports data on the dynamics of MreB filaments in *Bacillus subtilis* cells with altered cell shape. Using an impressive array of in vivo and in vitro approaches, they convincingly show that MreB moves in the direction of the largest membrane curvature, i.e. along the short axis in rod-shape cells and with random orientation in spherical cells. This movement somehow directs cell wall synthesis and guides/reinforces rod shape formation. Interestingly, when spherical cells convert to rods, elongative growth starts at a ring at the base of a bulge that grows to form a new rod-shaped cell body. The authors propose a model according to which MreB movement functions to strengthen the cross-linkage of the cell wall peptidoglycan.

Although the work is an important addition to the field several points made by the authors should be toned down before the paper is published.

- While it is possible that MreB indeed affects PG crosslinks, the peptidoglycan (PG) analysis data have major flaws because:i) The authors do not use the appropriate strains and conditions to test if oriented MreB motion results in more cross-linked PG.ii) They did not use the appropriate methodology to determine the muropeptide composition (see specific comments below). Therefore, the authors cannot draw any conclusions from the data presented and they should remove the flawed PG analysis data and all conclusions/models based on these from the manuscript.

- Although MreB could be sufficient to generate rod shape, the data does not prove it and thus the title is an overstatement (as well as any issuing statement). The whole set of data gathered present very clear evidence that MreB aligned along the highest curvature and even bend liposomes because it prefers a sharp ~200nm ring structure to align with. But that does not mean it creates the shape, only that once there, it aligns with it. In fact, the experiments do not rule that there is not another factor in charge of creating the rod and that the resulting structure in turn recruits MreB. Perhaps, the absence of TagO, leading to the loss of rod shape despite an MreB-dependent oriented synthesis, suggests MreB may not be sufficient. Similarly, the authors do not refer to a paper from the Gitai group showing that rodZ mutants of *E. coli* are able to grow as rods without MreB motion. How does this work relate to their model? Is MreB-motion required in Bacillus for rod-shape maintenance?

- The assumption that MreB and Mbl are interchangeable is a big stretch. It is still unclear why *B. subtilis* have several paralogs and if they carry specific functions. Since the authors are bringing new insights on a critical feature of MreBs, it is important to show that this can be extend to both MreB and Mbl, without starting with such assumption. At the very least, they should stick to one and not go back and forth between both. For example, in Figure 2—video 1, it is not acceptable to show GFP-MreB as a control experiment but use Mbl-GFP in the tagO knockout. This is not the only example. Many experiments are performed with one or the other protein, the authors should either "fill the gaps" or justify rigorously the choice of a given fusion for a given experiment.

Specific comments:

1) Peptidoglycan:

1) Does the oriented MreB motion help to incorporate PG with normal composition/cross-linkage, and is therefore the PG composition/cross-linkage altered in spherical cells (Figure 6B, Supplementary Figure 6D)? They aimed to answer this question by analysing the total PG of cells harvested from an over-night culture (subsection “Muropeptide analysis”), which is a poor experiment for two reasons. First, it is known that newly incorporated PG changes its structure over time (PG maturation) resulting in significant changes in the muropeptide composition; not all of the later PG maturation steps may be affected by the motion of MreB. Hence, to answer the question requires to determine the composition of newly made PG just after its incorporation into the cell wall near the membrane (and guided by MreB motion), perhaps by analysing the structure of new PG made during a short pulse after the addition of a radioactive compound incorporating into PG. Second, an overnight culture of *B. subtilis* contains a complex and unknown mixture of growing and non-growing cells, lysing cells, lysed cells, sporulating cells and spores, and over-night cultures of rod-shaped or spherical cells might well differ in the degree of lysis, sporulation etc. The overall PG composition of such mixtures may be related to MreB motion only indirectly or not at all. In my view to answer the question would require to analyse the composition of nascent PG upon short labelling in exponentially growing cells.

Strain chosen for determining the composition of the PG. They used the tagO depletion strain called BEG300 (which also expressed a Gfp-MreB protein; why would they need this?). The strain is a poor choice because the depleted cells lack wall teichoic acid and are therefore more prone to autolysis (endogenous PG hydrolases are known to be affected by the presence of teichoic acids). Ideally, one would use round cells that do not differ in the teichoic acid content, to prevent uncontrolled cleavage by hydrolases which alters the muropeptide profile. That the round cells of the tagO depleted cells (from the overnight-cultures) suffered from uncontrolled PG hydrolase activity is readily seen from the data presented in Figure 6B and Supplementary Figure 6D: I am not aware of any muropeptide profile of a strain/mutant from a Gram-positive species with a cross-linkage of as low as 11%, as was determined here. This low cross-linkage must come from uncontrolled activity of autolysins because of the presence of the disaccharide tetrapeptide muropeptide (peak 3, Supplementary Figure 6) that is not normally present in PG from *Bacillus subtilis*. This muropeptide originates from uncontrolled endopeptidase activity, degrading tetra-tri dimers to tetra and tri monomers, explaining both the low cross-linkage and the presence of tetra monomers. In my view this uncontrolled endopeptidase activity has nothing to do with the lack of orientation of the MreB motion, but is just a sign of extensive lysis of tagO depleted cells during over-night cultivation.

The PG isolation method used does not allow quantitative analysis of muropeptide composition. Remarkably, they wrote: "To prepare muropeptides from strain BEG300, a similar protocol was performed as reported previously (Atrih et al., 1999; Kuhner et al., 2014)". This statement is incorrect because both cited papers describe two very different PG isolation methods, and what they have used was the quick method by Kuhner et al., 2014 and not the method by Atrih et al., 1999. Importantly, only the Atrih method allows to isolate muropeptides for quantitative analysis. The Kuhner et al., 2014 reports a quicker and simpler method to isolate muropeptides to determine their structure, but for many reasons this method is not appropriate for quantitative determination of muropeptide composition. Indeed, the Kuhner paper does not claim that the new method is to determine the composition; they rather provided mass spectrometry data for the muropeptides isolated (from *E. coli* and S. aureus) but, rightly so, the Kuhner paper does not provide any quantitative data on the profiles they obtained. Hence, Hussain et al. should have used the Atrih method to isolate muropeptides for quantitative analysis.

Supplementary Figure 6 illustrates major problems of the data presented. The structures and many masses determined (as mentioned in the legend of Supplementary Figure 6) are obviously erroneous: previous publications showed that the majority of the meso-Dap residues in the PG from *B. subtilis* are amidated, due to amidation of the lipid II precursor, but this modification is not seen in their muropeptide structures. The tetrasaccharide (with tetra-tri peptide) is split into 4 peaks (No. 4, 5, 6 and 7) with the same mass, with retention times that differ by almost 10 minutes, and the determined mass of 1795.8 is 3 Da off the theoretical mass of 1792.8 Da (H^+^ versions) of the *B. subtilis* tetra-tri muropeptide. The presence of 4 peaks and the discrepancy to the theoretical mass (which is too high considering the accuracy of current mass spectrometers) are not explained, but I conclude that the muropeptide patterns and masses presented in Supplementary Figure 6 and its legend are not consistent with what is known about PG from *B. subtilis*.

Another obvious mistake in the manuscript is the sentence: "Similar muropeptides were observed as previously reported for other *Bacillus subtilis* strains (Kuhner et al., 2014)". Although they cite Kuhner et al., 2014 in this way, they seem to be unaware that Kuhner et al., 2014 does not report any data on *Bacillus subtilis* muropeptides! In fact, the Kuhner et al. 2014 paper only reports data for muropeptides from *E. coli* and S. aureus.

2) Protoplasts:

Absence of CW synthesis in protoplast (subsection “MreB Aligns Within Round Cells and Protoplasts Forced into Rod Shape”, last paragraph): This experiment using WGA tagging to show the absence of PG is useless at two levels. The control is useless as it is, because it does not prove there is no CW, but fortunately, this is not required for the demonstration. Since the cells have been protoplasted, we know that most of the PG is gone, by definition. The purpose of this control is to check if residual PG, too thin or non-continuous (thus insufficient to counteract osmotic pressure and maintain shape) could still be present and serve as tracks. The problem is if there were residual PG strands, would the WGA staining be sensitive enough to detect traces of PG? In fairness, we do not know and it would be interesting to know what can be detected by this method. However, as it is, this experiment is not a good control and would require to first prove than it can detect such traces by checking cells containing decreasing quantities of CW in parallel to another more sensitive approach (HPLC?). Now, this control is not critical because if there are traces of PG, they will be present both in the rod and round sections of the protoplast and will not be sufficient to organize the movement of MreBs in round areas anyway. This control should be removed and the text should be toned down, since there is no certainty that "any structure within the cell wall" has been removed.

Average angular distributions compared between wt and protoplasts of various width (see the aforementioned paragraph). The authors report a broadening of the distribution with the increase of the imposed width. But Figure 3 legend says that the standard deviation SD +/- 35° at 4µ, thus not significantly broader than in wt cells (SD +/-34°). Also, it is noticeable that the average angles are not significantly different between the cells tested: 86° observed with 5µ protoplasts and 88° observed in the Wt (with SD > 30°). Thus, the numbers do not agree with the authors' description in text. Could this discrepancy come from the fact that averages may not fully reflects the changes and that a graphical representation of the distribution would be more appropriate? The >90° angles reported in the text and legend are puzzling. For example, the authors report 93° and 94° for 2µ width protoplasts with MreB and Mbl. The angle distribution so far (Figure 1 and Figure 3) presented distributions ranging between 10 and 90°, thus not discriminating the direction of the pole that the patch is leaning to. It is therefore puzzling that the average numbers are here above 90°, hence now considering this direction. This is a bit misleading the values are very different from that observed in protoplast (81 and 86° at 4 and 5µ width) with wider shape, while in fact they are quite close.

3) WTA labeling:

WTA is uniformly distributed (Figure 6). It does not seem completely uniform on the presented images, with accumulation in inward bending of the membrane. In fact, the increase in WTA is more convincing than the MreB accumulation presented in Figure 6; Also, the assertion that the ligases move homogeneously in the membrane is not fully convincing. First, the cells in Figure 6—figure supplement 1 are too small for the reader to assert a lack of tracks for the Ligases (MIP). In fact, if a fraction of the population of ligases were interacting with MreB and move directionally, this could be easily missed with the MIP because of the bad S/N ratio. If the authors want to make this point (not critical for the whole story), single particle tracking would here be more appropriate to check for the absence of such a subpopulation.

---

## [Author Response]

[…] Although the work is an important addition to the field several points made by the authors should be toned down before the paper is published.

As explained below, we have worked to tone down the stronger claims previously made in the paper, as well as the title.

- While it is possible that MreB indeed affects PG crosslinks, the peptidoglycan (PG) analysis data have major flaws because:i) The authors do not use the appropriate strains and conditions to test if oriented MreB motion results in more cross-linked PG.ii) They did not use the appropriate methodology to determine the muropeptide composition (see specific comments below). Therefore, the authors cannot draw any conclusions from the data presented and they should remove the flawed PG analysis data and all conclusions/models based on these from the manuscript.

We appreciate the reviewers’ feedback on the PG crosslinking experiment and our associated analysis. Per the reviewers’ request we have removed this data from the manuscript. Even with this data removed, we respond to the points raised regarding this topic later in this response.

- Although MreB could be sufficient to generate rod shape, the data does not prove it and thus the title is an overstatement (as well as any issuing statement). The whole set of data gathered present very clear evidence that MreB aligned along the highest curvature and even bend liposomes because it prefers a sharp ~200nm ring structure to align with. But that does not mean it creates the shape, only that once there, it aligns with it. In fact, the experiments do not rule that there is not another factor in charge of creating the rod and that the resulting structure in turn recruits MreB. Perhaps, the absence of TagO, leading to the loss of rod shape despite an MreB-dependent oriented synthesis, suggests MreB may not be sufficient.

We respond to the points within the above section separately below.

Although MreB could be sufficient to generate rod shape, the data does not prove it and thus the title is an overstatement (as well as any issuing statement).

We agree with the reviewers, we cannot prove MreB is sufficient to generate rod shape, and such an experiment would be incredibly difficult. We did not mean to state or imply that MreB (or the Rod complex) was sufficient for rod shape. We realize that this impression may have arisen from our overly strong title. To fix this, we have:

1) Significantly softened the title. It is now: “MreB Filaments Align Along Greatest Principal Membrane Curvature to Orient Cell Wall Synthesis.”

2) Reworked the text to remove any issuing statements and made it clear which statements are models and which are direct interpretations of our data.

3) We also point out in the text that MreB is not sufficient for rod shape, as other factors are also required. This is directly demonstrated by our data, showing that WTA depletions create round cells.

In fact, the experiments do not rule that there is not another factor in charge of creating the rod and that the resulting structure in turn recruits MreB.

We completely agree with the reviewers, there remains a possibility that some other cellular factor first determines a region of rod shape, and the Rod complex subsequently reinforces that region. In fact, we do not favor either of the two possibilities: 1) Local regions of rod shape form on their own by some physical phenomena, or 2) Local regions of rod shape form by the actions of upstream factors. We had, in fact, looked for such factors, attempting to see if FtsZ acts upstream of rod shape formation (Figure 5—figure supplement 1).

We thought this point was clear in the initial submission. But as the reviewers bring this up as a main point, it is evident we failed at communicating it clearly. To remedy this, we have rewritten and expanded on this point in the Discussion, specifically noting that regions of rod shape could form via the action of some other cellular factor that acts upstream of MreB and the Rod Complex. This point is then repeated later in the text for further clarity. We also point to the experiments conducted looking for such factors. We include these paragraphs below.

“It remains to be determined how these local self-propagating regions form. In our recoveries, the most common shape variation we observe preceding rod emergence is small outward bulges flanked by regions of inward curvature (Figure 5). […] Thus far we have only been able to determine that cell division is not required (Figure 5—figure supplement 1).”

“Whether these outward bulges arise from physical effects or by the action of upstream factors, the geometry at their interface may play a nucleating role in rod shape formation.”

Although MreB could be sufficient to generate rod shape….. Perhaps, the absence of TagO, leading to the loss of rod shape despite an MreB-dependent oriented synthesis, suggests MreB may not be sufficient.

Again, we completely agree with the reviewers: our *tagO* depletion data directly demonstrates that MreB (and the Rod Complex) cannot be sufficient for rod shape. Given our *tagO* depletions made spherical cells we had assumed this was obvious, but we now realize that we should state have stated this more explicitly. This comment also made us realize we should explicitly state our model for the relationship between MreB, the WTAs, and rod shape. To remedy this, we now include a paragraph in the Discussion addressing this point, explaining the data demonstrating that both WTAs (TagO) and MreB are required for rod shape, and our model for how each of these systems functions to reinforce the rod sidewalls.

“While the MreB guided circumferential insertion of cell wall material is expected to reinforce rod shape (Amir and Nelson, 2012; Chang and Huang, 2014; Baskin, 2005), it must be noted that it is not sufficient, as other factors are also needed: WTA-depleted cells become round (D'Elia et al., 2006) (Figure 1) even under conditions where they contain oriented MreB motions (Figure 3—figure supplement 1). Based on both past work (Matias and Beveridge, 2005; Kern et al., 2010) and our data (Figure 1, Figure 1—figure supplement 1, Figure 5, Figure 5—video 2), we believe that WTAs, in combination with PG, work to coordinate Mg^2+^ to increase the overall cell wall stiffness, thus allowing regions of oriented insertion to maintain and propagate their shape.”

Similarly, the authors do not refer to a paper from the Gitai group showing that rodZ mutants of E. coli are able to grow as rods without MreB motion. How does this work relate to their model?

We appreciate the reviewers pushing us to address this point, as we were curious about this finding as well. Given the rather low α values reported for wild type MreB motion (~1.2) in Morgenstein et al. (Amir and Nelson, 2012; Chang and Huang, 2014), we initially suspected their conclusions might have arisen from a subfraction of filaments moving directionally. However, this was hard to ascertain from their paper, as their analysis was based on an average RMSD of overall filaments at each t (aka TARSMD). To examine this ourselves, we contacted the Gitai lab and requested the strain that was the basis of their conclusion (RM478, *ΔrodZ, mreBS14A-msfGFP*), as well as a “wild type” strain (NO50, *msfGFP-MreB(sw)*).

When we imaged the ΔrodZ, mreBS14A-msfGFP strain, we were surprised to find that, in contrast to the findings of Morgenstein, MreB does indeed move directionally in their strain. We next imaged this strain in different media and different temperatures, directional MreB motion was present in all cases. We did observe some differences: at higher temperatures or richer media, far more filaments appeared to move, whereas at lower temperatures or slower media, less filaments moved. Furthermore, by replicating the growth and imaging conditions used in (Amir and Nelson, 2012; Chang and Huang, 2014) – first growing cells at 37C LB, then placing them under a pad of M63Glucose 25C, then imaging – we noted that while the large majority of filaments were immobile, a small fraction of filaments still moved directionally. TIRFM movies of different growth conditions are now included as Figure 7—video 2.

To verify this result, we first checked this was the correct strain by sequencing, confirming the ΔrodZ and mreBS14A mutations. Next, we reached out to Tom Bernhardt’s lab to independently confirm our observations, who using TIRF microscopy also observed robust directional MreB motion – many filaments moving at 37C, and a lesser number of filaments moving at 30C. We next contacted the Gitai lab, shared our findings and data, and worked with them over the last few weeks on this issue. They could reproduce our findings, observing directional MreB motion by 37C, and at 25C (the conditions they imaged these cells in their previous paper), they see much less, (if any) filaments move (“little or no motion”). While we are still having discussions as to how many moving filaments are required to make a rod, currently all groups agree that MreB can move directionally in the “rod shaped” *ΔrodZ* cells. Thus, two conclusions within Morgenstein 2016 no longer hold: 1) MreB motion requires RodZ, and 2) Rod shape is independent of directional MreB motion.

We note that, in our opinion, even very small number of filaments moving at 25C is not an argument for the decoupling of MreB motion and rod shape, as:

1) Even though only a few filaments move directionally at 25C, this is not unexpected, as work from the Carballido-López lab have shown that under similar conditions wild type *E. coli* in only contains a few directionally moving MreB filaments (Billaudeau et al., 2017).

2) In Morgenstein 2016, before imaging the cells under the conditions that show greatly reduced MreB motion, the rod shape of the cells was formed in during the “pre-imaging” condition (before they were put under pads at 25C). These pre-growth conditions (LB at 37C) show extensive directional motion.

We now include a paragraph discussing this point at the end of the paper, along with supplemental video showing the directional motion of MreB filaments this strain under 4 different growth conditions (Figure 7—video 2). We also include a panel analyzing trajectories of these filaments showing that motion is directional (Figure 7—figure supplement 1). We would really like to thank the Gitai lab for working with us to resolve this issue in such an open fashion, and we include an acknowledgement.

- The assumption that MreB and Mbl are interchangeable is a big stretch. It is still unclear why B. subtilis have several paralogs and if they carry specific functions. Since the authors are bringing new insights on a critical feature of MreBs, it is important to show that this can be extend to both MreB and Mbl, without starting with such assumption. At the very least, they should stick to one and not go back and forth between both. For example, in Figure 2—video 1, it is not acceptable to show GFP-MreB as a control experiment but use Mbl-GFP in the tagO knockout. This is not the only example. Many experiments are performed with one or the other protein, the authors should either "fill the gaps" or justify rigorously the choice of a given fusion for a given experiment.

We did not mean to assume MreB and Mbl are interchangeable, rather that they always copolymerize and thus are within the same filaments. We based this assumption on not only data from within our lab, but also multiple publications: 1) they colocalize with each other in vivo (Baskin, 2005; Chang and Huang, 2014), and 2) have been shown to directly interact by both FRET and BiFC (Yao et al., 1999). We note the copolymerization of these two proteins is so established, it has been summarized by A. Chastanet in a review: “These studies conclusively showed that all three MreBs of *B. subtilis* belong to the same structures in the cell” (Baskin, 2005).

However, we do understand the reviewers’ concerns, especially given the different genetic effects that have been observed in the different homologs (Paredez et al., 2006) (Wachi et al., 1989). Per their request, we now include GFP-MreB data in all figures and movies that previously assayed GFP-Mbl. In agreement with the multiple reports that these proteins function as copolymers, we see no difference between the behavior of either Mbl and MreB filaments in any of our assays.

Specifically:

– Figure 1 – we note that the distribution in Figure 1 was mislabeled as Mbl, but was actually MreB, (discovered when we looked at a previous draft of the paper and confirmed in our source data.) We apologize for this error and have corrected the label. We now show both MreB and Mbl distributions in this figure. As previously observed, (Alyahya et al., 2009; Bendezú et al., 2009) these distributions overlap.

– Figure 1 and Figure 1—video 1 – have been updated to show intersection of both MreB and Mbl tracks.

– Figure 2 and Figure 2—video 1 – have been updated to show the circumferential motion of Mbl and MreB in rod shaped cells, and also isotropic motion of Mbl and MreB in *tagO* knockout cells.

– Figure 2—video 2 – now shows examples of both oriented motion of Mbl and MreB in rods, as well as isotropic motion of both filaments at later points of Pbp2a depletion as cells become round.

– Figure 3—video 2 – we now include timelapse movies of both GFP-MreB and GFP-Mbl in protoplasts.

Specific comments:1) Peptidoglycan:1) Does the oriented MreB motion help to incorporate PG with normal composition/cross-linkage, and is therefore the PG composition/cross-linkage altered in spherical cells (Figure 6B, Supplementary Figure 6D)? They aimed to answer this question by analysing the total PG of cells harvested from an over-night culture (subsection “Muropeptide analysis”), which is a poor experiment for two reasons. First, it is known that newly incorporated PG changes its structure over time (PG maturation) resulting in significant changes in the muropeptide composition; not all of the later PG maturation steps may be affected by the motion of MreB. Hence, to answer the question requires to determine the composition of newly made PG just after its incorporation into the cell wall near the membrane (and guided by MreB motion), perhaps by analysing the structure of new PG made during a short pulse after the addition of a radioactive compound incorporating into PG. Second, an overnight culture of B. subtilis contains a complex and unknown mixture of growing and non-growing cells, lysing cells, lysed cells, sporulating cells and spores, and over-night cultures of rod-shaped or spherical cells might well differ in the degree of lysis, sporulation etc. The overall PG composition of such mixtures may be related to MreB motion only indirectly or not at all. In my view to answer the question would require to analyse the composition of nascent PG upon short labelling in exponentially growing cells.Strain chosen for determining the composition of the PG. They used the tagO depletion strain called BEG300 (which also expressed a Gfp-MreB protein; why would they need this?). The strain is a poor choice because the depleted cells lack wall teichoic acid and are therefore more prone to autolysis (endogenous PG hydrolases are known to be affected by the presence of teichoic acids). Ideally, one would use round cells that do not differ in the teichoic acid content, to prevent uncontrolled cleavage by hydrolases which alters the muropeptide profile. That the round cells of the tagO depleted cells (from the overnight-cultures) suffered from uncontrolled PG hydrolase activity is readily seen from the data presented in Figure 6B and Supplementary Figure 6D: I am not aware of any muropeptide profile of a strain/mutant from a Gram-positive species with a cross-linkage of as low as 11%, as was determined here. This low cross-linkage must come from uncontrolled activity of autolysins because of the presence of the disaccharide tetrapeptide muropeptide (peak 3, Supplementary Figure 6) that is not normally present in PG from Bacillus subtilis. This muropeptide originates from uncontrolled endopeptidase activity, degrading tetra-tri dimers to tetra and tri monomers, explaining both the low cross-linkage and the presence of tetra monomers. In my view this uncontrolled endopeptidase activity has nothing to do with the lack of orientation of the MreB motion, but is just a sign of extensive lysis of tagO depleted cells during over-night cultivation.

We appreciate the reviewers’ insight and comments. We begin by stating that, upon further reflection, we completely agree with the reviewer that our PG crosslinking analysis of *tagO* null cells vs wild type strains is not a well-controlled experiment. As they note, it would be more dominated by effects arising from the absence of WTAs on the hydrolase activity than those from changes in cell shape. As requested, we have completely removed the crosslinking data and analysis from the text, as well as any references to or interpretation thereof. Even though this is now removed, we still respond to the reviewers’ points.

Strain chosen for determining the composition of the PG. They used the tagO depletion strain called BEG300 (which also expressed a Gfp-MreB protein; why would they need this?).

We used bEG300 as we wished to conduct our comparison of crosslinking in the same strain in which we did the rest of our assays, so all genetic conditions would be comparable across experiments.

The reviewer notes – They aimed to answer this question by analysing the total PG of cells harvested from an over-night culture, which is a poor experiment for two reasons, First, it is known that newly incorporated PG changes its structure over time (PG maturation) resulting in significant changes in the muropeptide composition; not all of the later PG maturation steps may be affected by the motion of MreB.

We apologize for the confusion raised by our methods. While this culture was indeed grown overnight, it did not reach anywhere near stationary phase. i.e. the culture was grown overnight because cells with low TagO expression grow incredibly slowly. We never allowed the OD of the cells to exceed 0.8, and cells for this were harvested at an OD600 of 0.7-0.8. Likewise, the inocula for the large flask cultures from which the cells were harvested were themselves drawn from standard, log phase “overnights”, created from serial dilutions that never exceeded 0.6. In fact, the reviewer’s confusion is justified, as we erroneously reported in the initial methods section that we drew the cells from a “2 mL overnight” culture. We apologize for this error.

The reviewer notes – Ideally, one would use round cells that do not differ in the teichoic acid content, to prevent uncontrolled cleavage by hydrolases which alters the muropeptide profile.

This is one point we pondered deeply when conducting these experiments. The ideal experiment, as the reviewer notes, is to use round cells without altered TagO expression. Unfortunately, there is no way to make round cells without affecting some component of the cell wall as *B. subtilis*, unlike *E. coli,* is extremely resistant to mechanical shape perturbations. The only means known to make *B. subtilis* round is to: 1) interfere with Wall teichoic acid synthesis, or 2) inhibit or deplete a component of the Rod complex (such as MreB or Pbp2a). We opted for assaying WTA depletions, as inhibition or depletion of the Rod complex, would directly affect the process we were attempting to study (synthesis and crosslinking by the Rod complex.)

The PG isolation method used does not allow quantitative analysis of muropeptide composition. Remarkably, they wrote: "To prepare muropeptides from strain BEG300, a similar protocol was performed as reported previously (Atrih et al., 1999; Kuhner et al., 2014)". This statement is incorrect because both cited papers describe two very different PG isolation methods, and what they have used was the quick method by Kuhner et al., 2014 and not the method by Atrih et al., 1999.

We note that the referencing was incorrectly placed in this section, and the sacculi preparation should have cited Kuhner et al. 2014. Several manuscript edits were circulated, leading to this misplacement of referencing that is also mentioned in the next comment.

Importantly, only the Atrih method allows to isolate muropeptides for quantitative analysis. The Kuhner et al., 2014 reports a quicker and simpler method to isolate muropeptides to determine their structure, but for many reasons this method is not appropriate for quantitative determination of muropeptide composition. Indeed, the Kuhner paper does not claim that the new method is to determine the composition; they rather provided mass spectrometry data for the muropeptides isolated (from E. coli and S. aureus) but, rightly so, the Kuhner paper does not provide any quantitative data on the profiles they obtained. Hence, Hussain et al. should have used the Atrih method to isolate muropeptides for quantitative analysis.

There were significant differences in the ratios of muropeptide species as determined by integrated peaks using extracted ion chromatograms and this was also reflected in the total ion chromatogram traces. It seems likely that similar trends in muropeptide abundances would have been observed using Atrih et al.1999.

Supplementary Figure 6 illustrates major problems of the data presented. The structures and many masses determined (as mentioned in the legend of Supplementary Figure 6) are obviously erroneous: previous publications showed that the majority of the meso-Dap residues in the PG from B. subtilis are amidated, due to amidation of the lipid II precursor, but this modification is not seen in their muropeptide structures. The tetrasaccharide (with tetra-tri peptide) is split into 4 peaks (No. 4, 5, 6 and 7) with the same mass, with retention times that differ by almost 10 minutes, and the determined mass of 1795.8 is 3 Da off the theoretical mass of 1792.8 Da (H^+^ versions) of the B. subtilis tetra-tri muropeptide.

For the mass of the tetra-tri muropeptide (M+H=1795.8) was consistent with Atrih et al. 1999 for species 20 (also a tetra-tri) muropeptide. It may be possible that loss of amidation of this species could explain this mass difference.

The presence of 4 peaks and the discrepancy to the theoretical mass (which is too high considering the accuracy of current mass spectrometers) are not explained, but I conclude that the muropeptide patterns and masses presented in Supplementary Figure 6 and its legend are not consistent with what is known about PG from B. subtilis.

We note that in Atrih et al. 1999 analysis, muropeptides corresponding to the same composition were also observed split between several peaks of different retention times. It is possible that the muropeptides species associate with different salts during the preparation of the sacculi and could therefore migrate differently.

Another obvious mistake in the manuscript is the sentence: "Similar muropeptides were observed as previously reported for other Bacillus subtilis strains (Kuhner et al., 2014)". Although they cite Kuhner et al., 2014 in this way, they seem to be unaware that Kuhner et al., 2014 does not report any data on Bacillus subtilis muropeptides! In fact, the Kuhner et al. 2014 paper only reports data for muropeptides from E. coli and S. aureus.

We note that the references were also incorrect in this description, and that we used Atrih et al., 1999 to compare the muropeptides observed to previously reported *B. subtilis* sacculi analysis.

2) Protoplasts:Absence of CW synthesis in protoplast (subsection “MreB Aligns Within Round Cells and Protoplasts Forced into Rod Shape”, last paragraph): This experiment using WGA tagging to show the absence of PG is useless at two levels. The control is useless as it is, because it does not prove there is no CW, but fortunately, this is not required for the demonstration. Since the cells have been protoplasted, we know that most of the PG is gone, by definition. The purpose of this control is to check if residual PG, too thin or non-continuous (thus insufficient to counteract osmotic pressure and maintain shape) could still be present and serve as tracks. The problem is if there were residual PG strands, would the WGA staining be sensitive enough to detect traces of PG? In fairness, we do not know and it would be interesting to know what can be detected by this method. However, as it is, this experiment is not a good control and would require to first prove than it can detect such traces by checking cells containing decreasing quantities of CW in parallel to another more sensitive approach (HPLC?). Now, this control is not critical because if there are traces of PG, they will be present both in the rod and round sections of the protoplast and will not be sufficient to organize the movement of MreBs in round areas anyway. This control should be removed and the text should be toned down, since there is no certainty that "any structure within the cell wall" has been removed.

We appreciate the reviewers feedback on this point. As suggested, we have removed the control. We have also toned down the text – it now reads:

“We next attempted to minimize any contribution to MreB filament orientation from A) the directional motion of filaments, and B) any pre-existing order within the sacculus.”

Average angular distributions compared between wt and protoplasts of various width (see the aforementioned paragraph). The authors report a broadening of the distribution with the increase of the imposed width. But Figure 3 legend says that the standard deviation SD +/- 35° at 4µ, thus not significantly broader than in wt cells (SD +/-34°). Also, it is noticeable that the average angles are not significantly different between the cells tested: 86° observed with 5µ protoplasts and 88° observed in the Wt (with SD > 30°). Thus, the numbers do not agree with the authors' description in text. Could this discrepancy come from the fact that averages may not fully reflects the changes and that a graphical representation of the distribution would be more appropriate? The >90° angles reported in the text and legend are puzzling. For example, the authors report 93° and 94° for 2µ width protoplasts with MreB and Mbl. The angle distribution so far (Figure 1 and Figure 3) presented distributions ranging between 10 and 90°, thus not discriminating the direction of the pole that the patch is leaning to. It is therefore puzzling that the average numbers are here above 90°, hence now considering this direction. This is a bit misleading the values are very different from that observed in protoplast (81 and 86° at 4 and 5µ width) with wider shape, while in fact they are quite close.

We apologize for the confusion regarding the negative angles and the standard deviations. The distributions shown originated from plots of the angles between 0-180 degrees, and the SD values are from these original distributions, shown in Author response image 1. We later switched the range to 0-90, due to the fact that the direction of motion (or orientation) is not meaningful, as no biased angle to MreB motion has been observed in *B. subtilis* by 3 different groups (as noted by the reviewer) (Dempwolff et al., 2011; Domínguez-Escobar et al., 2011; Garner et al., 2011).

Rather than SD, we now report mean deviation from 90°, calculated with the following formula:

σ90=∑i=1N(xi-90)2N

The point of this figure was to test if MreB filaments remain aligned at various rod widths, and as expected, the averages are not significantly different. However, we agree with the reviewer that, given these highly asymmetric 0-90° distributions, SD is not a good measure of variability and that a graphical representation is better. We think 3F(ii) gives a much better picture than the SD values, and apologize for the confusion caused.

3) WTA labeling:WTA is uniformly distributed (Figure 6). It does not seem completely uniform on the presented images, with accumulation in inward bending of the membrane. In fact, the increase in WTA is more convincing than the MreB accumulation presented in Figure 6; Also, the assertion that the ligases move homogeneously in the membrane is not fully convincing. First, the cells in Figure 6—figure supplement 1 are too small for the reader to assert a lack of tracks for the Ligases (MIP). In fact, if a fraction of the population of ligases were interacting with MreB and move directionally, this could be easily missed with the MIP because of the bad S/N ratio. If the authors want to make this point (not critical for the whole story), single particle tracking would here be more appropriate to check for the absence of such a subpopulation.It does not seem completely uniform on the presented images, with accumulation in inward bending of the membrane

In regard to the WTA accumulation as measured by ConA, we realize our wording was inaccurate, and the phrase “*uniformly distributed*” in the legend is incorrect. We intended to state that WTAs appear “*equally distributed”* in rods and spheres. We have updated the figure legend of Figure 6—figure supplement 1 to read:

“Concanavalin A conjugated to Alexa Fluor 647 specifically stains wall teichoic acids and localizes equally to both rods and spheres during sphere to rod transitions.”

We would like to remind the reviewers that the point of this experiment was not to look at subcellular localization, but rather to test if, during *tagO* reinduction, WTAs preferentially incorporated into rods, perhaps giving an explanation as to why only one part of the cell reformed rod shape. Figure 6—figure supplement 1 shows both rods and spheres display ConA staining, and thus WTAs are incorporated into both spheres and the rods during *tagO* reinduction. This is quantitated in Figure 6, where the average intensity per cell was determined by tracing cell outlines and measuring the intensity.

First, the cells in FigS6C are too small for the reader to assert a lack of tracks for the Ligases (MIP).

We agree that the images of cells were too small. We have updated this figure, zooming in so that the cells are more visible.

If the authors want to make this point (not critical for the whole story), single particle tracking would here be more appropriate to check for the absence of such a subpopulation

We thank the reviewers for suggestion, we followed up on it by conducting single molecule tracking of all 3 ligases. We created native HaloTag fusions to *tagT tagU and tagV*. By sparsely labeling these strains with JF549 and imaging with TIRF we have gained much better descriptions of their dynamics. This data is now included as figures Figure 6—figure supplement 2 and Figure 6—video 3. As a control for directional motion, we also imaged HaloTag-mreB and HaloTag-pbp2a. For this experiment, we imaged all strains at 250msec, a frame rate that oversamples directional motion, but diffusive particles cannot be tracked for more than a few frames. As shown in Figure 6—figure supplement 2, at this frame rate we see clear directional motion for MreB and PBP2A, but we never observe any clear directional motions in TagT, TagU, or TagV.